# Polygenic and developmental profiles of autism differ by age at diagnosis

Xinhe Zhang[1,2 ✉], Jakob Grove[3,4,5,6,7], Yuanjun Gu[1,2], Cornelia K. Buus[5,7], Lea K. Nielsen[5,7], Sharon A. S. Neufeld[1], Mahmoud Koko[8], Daniel S. Malawsky[8], Emma M. Wade[8], Ellen Verhoef[9], Anna Gui[10,11], Laura Hegemann[12,13], APEX Consortium*, iPSYCH Autism Consortium*, PGC-PTSD Consortium*, Daniel H. Geschwind[14,15,16,17], Naomi R. Wray[18,19], Alexandra Havdahl[12,13,20], Angelica Ronald[11,21], Beate St Pourcain[9,22,23], Elise B. Robinson[24,25], Thomas Bourgeron[26], Simon Baron-Cohen[1,2,27] & Anders D. Børglum[3,4,5,7], Hilary C. Martin[8,28], Varun Warrier[1,2,27 ✉]

Although autism has historically been conceptualized as a condition that emerges in early childhood[1,2], many autistic people are diagnosed later in life[3–5]. It is unknown whether earlier- and later-diagnosed autism have different developmental trajectories and genetic profiles. Using longitudinal data from four independent birth cohorts, we demonstrate that two different socioemotional and behavioural trajectories are associated with age at diagnosis. In independent cohorts of autistic individuals, common genetic variants account for approximately 11% of the variance in age at autism diagnosis, similar to the contribution of individual sociodemographic and clinical factors, which typically explain less than 15% of this variance. We further demonstrate that the polygenic architecture of autism can be broken down into two modestly genetically correlated ($r_g = 0.38$, s.e. = 0.07) autism polygenic factors. One of these factors is associated with earlier autism diagnosis and lower social and communication abilities in early childhood, but is only moderately genetically correlated with attention deficit–hyperactivity disorder (ADHD) and mental-health conditions. Conversely, the second factor is associated with later autism diagnosis and increased socioemotional and behavioural difficulties in adolescence, and has moderate to high positive genetic correlations with ADHD and mental-health conditions. These findings indicate that earlier- and later-diagnosed autism have different developmental trajectories and genetic profiles. Our findings have important implications for how we conceptualize autism and provide a model to explain some of the diversity found in autism.

Ever since its earliest descriptions in the 1940s[1,2], autism has been thought of as a condition that emerges in early childhood. However, a greater proportion of autistic individuals are now receiving an autism diagnosis from mid-childhood onwards than in early childhood[3–5]. One factor that may explain these findings is a shift in the conceptualization of the condition over time, including the recognition that the behavioural signs of autism may not manifest clearly in the first three years of life[6–9]. Supporting this, some studies have demonstrated that a subset of children who do not initially meet the criteria for an autism diagnosis receive a diagnosis later[7,10–14]. Later autism diagnosis is associated with elevated co-occurring mental-health conditions[15,16], highlighting the need to understand why some autistic people are not diagnosed until later in life.

Several social, demographic and clinical factors have been linked to age at autism diagnosis[17]. However, previous studies have shown that individual clinical and sociodemographic factors explain only a small proportion (typically less than 15%) of the variance in age at autism diagnosis (Extended Data Fig. 1 and Supplementary Table 1). This indicates that other factors contribute to age at autism diagnosis. One of these additional factors could be genetic differences between autistic individuals. Despite the relatively high heritability of autism[18], the role of genetics in age at autism diagnosis has not to our knowledge been previously studied.

Two theoretical models can explain how genetics affect age at autism diagnosis. In the first model, autism has a single polygenic aetiology, with the same set of genetic variants underlying autism, regardless of age at diagnosis (the 'unitary model'; Extended Data Fig. 2). In this model, later-diagnosed autism may have subtle clinical features that are harder to recognize early in life, so individuals do not cross a diagnostic threshold earlier in life, perhaps because they have a lower genetic predisposition to autism. As these people get older, environmental factors may alter their clinical features, eventually bringing individuals above the clinical threshold to receive an autism diagnosis later in life.

---

A list of affiliations appears at the end of the paper. *Lists of authors and their affiliations appear online. ✉e-mail: xz452@cam.ac.uk; vw260@cam.ac.uk

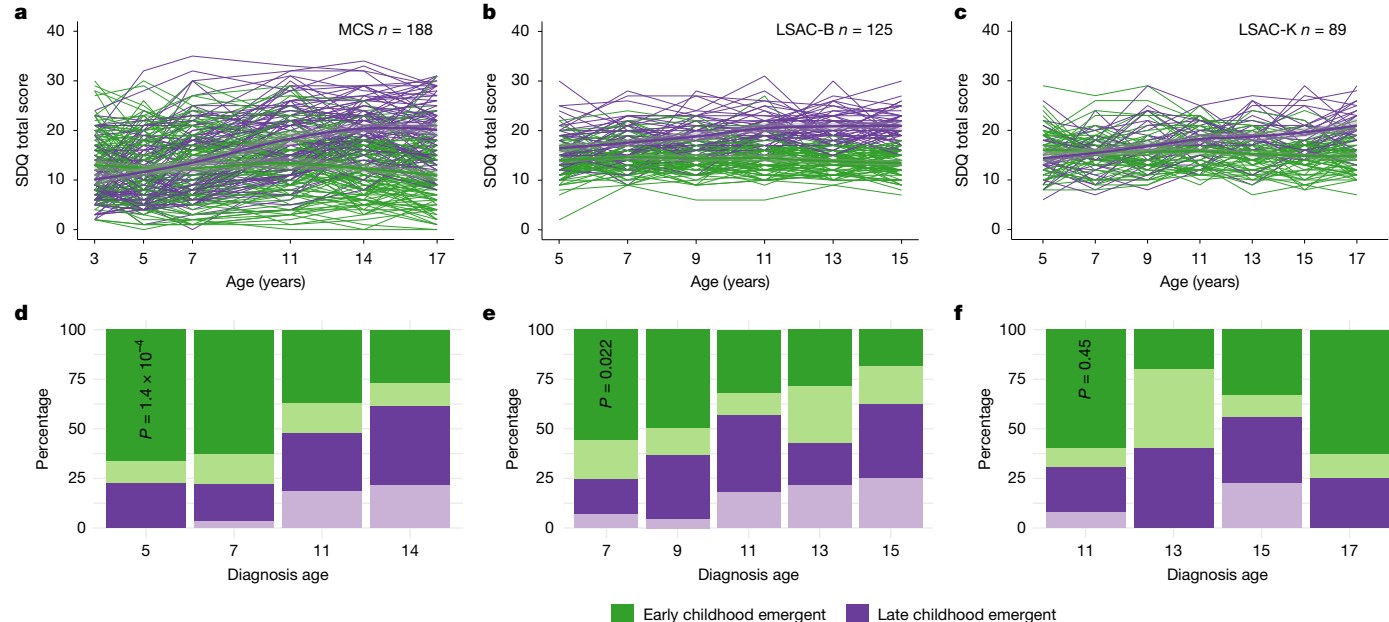

**Fig. 1 | Trajectory analyses in three of the four birth cohorts. a–c**, Longitudinal growth mixture models of SDQ total scores in autistic individuals, demonstrating the presence of two groups in the MCS (**a**), LSAC-B (**b**) and LSAC-K (**c**) cohorts. Shaded areas indicate 95% confidence intervals of the line of best fit. **d–f**, Stacked bar charts show the proportion of individuals who had been diagnosed as autistic at specific ages, categorized by membership in the latent trajectories identified from the growth mixture models in MCS (**d**), LSAC-B (**e**), and LSAC-K (**f**) cohorts. Darker colours indicate male individuals and lighter colours indicate female individuals. *P*-values are from χ² tests (two-sided) comparing the distribution of age at autism diagnosis between the two latent trajectories (pooling the two sexes).

An alternative model is that earlier- and later-diagnosed autism have different underlying developmental trajectories and polygenic aetiologies (the 'developmental model'; Extended Data Fig. 2). This model aligns with existing evidence that the genetic influences on traits related to autism vary across development[19–21]. This model does not preclude a role for environmental factors influencing when someone receives an autism diagnosis but implies that different sets of genetic variants are associated with earlier- and later-diagnosed autism.

Here we examined the evidence for these two models through four linked aims (Extended Data Fig. 3). First, we investigated whether the trajectories of socioemotional and behavioural development are associated with age at autism diagnosis in birth cohorts. Although variable developmental trajectories have been observed among autistic individuals and their younger siblings[22], it is unclear whether these differences in trajectories are associated with age at diagnosis. Second, we estimated the proportion of variance in age at autism diagnosis that is explained by common single nucleotide polymorphisms (SNP-based heritability). We then tested whether this is attenuated by a range of clinical and demographic factors, as predicted by the unitary model. Third, we investigated whether different polygenic factors are associated with earlier and later autism diagnosis, as predicted by the developmental model. Finally, we estimated the genetic correlation between the autism polygenic factors related to age at diagnosis and other mental-health and developmental phenotypes.

We provide a summary of the study and address potential questions regarding the implications of the findings in the Supplementary Summary and FAQs.

## Behavioural trajectories and diagnosis age

In Aim 1, we investigated whether autistic individuals have varying socioemotional and behavioural trajectories, and whether these are associated with age at autism diagnosis in three birth cohorts (*n* = 89 to 188 autistic individuals with recorded age at diagnosis between 5 years and 17 years). These are the Millennium Cohort Study (MCS,

participants born in 2000) and the Longitudinal Study of Australian Children: Kindergarten cohort (LSAC-K, 1999) and Birth cohort (LSAC-B, 2003) (Extended Data Fig. 4, Supplementary Table 2 and Supplementary Note 1). All three cohorts collected longitudinal information on socioemotional and behavioural development using the carer-reported Strengths and Difficulties Questionnaire (SDQ)[23]. The SDQ has five subscales (emotional, conduct, hyperactivity/inattention, peer problems and prosocial behaviours), and the total score of difficulties (hereafter 'total difficulties') is the summed score of the first four subscales. The SDQ is widely used, has excellent psychometric properties[24–26] and is largely invariant across age, sex and different populations[27–29], indicating that it is measuring the same latent trait across these demographic variables. Furthermore, the SDQ is moderately correlated with autism-specific measures[30–33], although it does not capture all of the core diagnostic features of autism. Because not all cohorts recorded the exact age when children received their autism diagnosis, we used the child's age during the study data collection when carers first reported the diagnosis as an approximation of age at autism diagnosis.

To identify latent trajectories, we used growth mixture models of the SDQ total difficulties and subscale scores among autistic individuals in all three cohorts. Growth mixture models do not require grouping based on an a priori hypothesis but can identify latent subgroups based on longitudinal differences in SDQ scores.

Across all three birth cohorts, growth mixture modelling identified a two-trajectory model as being optimal for SDQ total difficulties and most subscale scores (Supplementary Table 3 and Supplementary Figs. 1–6). The first latent trajectory was characterized by difficulties in early childhood that remained stable or modestly attenuated in adolescence (termed 'early childhood emergent latent trajectory'). The second latent trajectory was characterized by fewer difficulties in early childhood that increased in late childhood and adolescence (termed 'late childhood emergent latent trajectory') (Fig. 1a–c).

Autistic individuals in the early childhood emergent latent trajectory were more likely to be diagnosed as autistic in childhood than autistic individuals in the late childhood emergent latent trajectory in MCS

$(P = 1.42 \times 10^{-4}, \chi^2 \text{ test})$ and LSAC-B $(P = 2.24 \times 10^{-2}, \chi^2 \text{ test})$ (Fig. 1d–f and Supplementary Table 4). This difference was not significant in LSAC-K, possibly because age 11 was the earliest time an autism diagnosis was recorded.

Sensitivity analyses in MCS confirmed the robustness of the two latent trajectories and their association with age at diagnosis among autistic children. We identified consistent results after expanding the sample to include individuals with co-occurring ADHD ($n = 238$; Supplementary Tables 3 and 4), and after imputing missing data to increase the statistical power and reduce bias ($n = 623$) (Supplementary Table 5 and Supplementary Notes 2 and 3). We also obtained consistent results when restricting the analyses to only male individuals ($n = 136$; Supplementary Tables 3 and 4), indicating that these results were not driven by sex differences in age of diagnosis. We were unable to run equivalent female-only analyses owing to the low sample sizes.

To assess the specificity of this result to autism, we tested whether similar latent trajectories were also observed in children with ADHD but not autism ($n = 89$, imputed $n = 325$) in MCS using growth mixture models. Two latent SDQ trajectory classes emerged, but these were not significantly linked to age of ADHD diagnosis, except for the SDQ hyperactivity/inattention and conduct problems subscales in the imputed sample (Supplementary Figs. 7 and 8 and Supplementary Table 6), indicating that the findings are relatively specific to autism, rather than to neurodevelopmental conditions more broadly.

Although female individuals receive an autism diagnosis later than male individuals on average[34], in all three cohorts, the sex ratio was similar between the two latent trajectories (Supplementary Table 4), possibly because of their relatively small sample sizes. Individuals in the late childhood emergent latent trajectories were more likely to report mental-health conditions (Supplementary Table 7), consistent with previous epidemiological observations among later-diagnosed autistic individuals[15,16].

Using multiple regression models, we then examined the extent to which these two latent trajectories contributed to differences in age at autism diagnosis over and above sociodemographic and cognitive characteristics (Supplementary Table 8). In these models, SDQ latent trajectories explained 11.7% (LSAC-B) to 30.3% (MCS) of the variance in age of autism diagnosis. By contrast, sociodemographic variables explained 4.8% (LSAC-B) to 5.5% (MCS) of the total variance across cohorts, consistent with previous reports (Extended Data Fig. 1). In the imputed MCS sample ($n = 623$; Supplementary Table 5 and Supplementary Note 2), the SDQ latent trajectories and sociodemographic variables explained 56.6% and 3.2% of the variance, respectively. The effects of the sociodemographic variables were not mediated by the SDQ latent trajectories (Supplementary Table 8 and Supplementary Note 4).

The associations between different SDQ trajectories and age at autism diagnosis were also supported by latent growth curve models fitted on earlier- and later-diagnosed autistic individuals (Supplementary Note 5, Supplementary Table 9 and Supplementary Figs. 1–6, 9 and 10). These results confirm that the association between SDQ trajectories and age at autism diagnosis is robust to methodological choices

## Age at autism diagnosis is heritable

The above analyses demonstrate that variation in socioemotional and behavioural trajectories, measured using the SDQ, is associated with age at autism diagnosis. Previous research has demonstrated that developmental variation traits related to autism are partly explained by genetic factors[19,20,35–38]. A corollary of this is that genetic factors may also be associated with age at autism diagnosis.

Subsequently, in Aim 2, we tested whether age at autism diagnosis is heritable in two large cohorts of autistic individuals using genetic data and information on age at autism diagnosis. This includes: first,

the Danish-based iPSYCH cohort ($n_{total} = 18,965$), a population-based sample derived from the Danish national registries that includes autistic individuals; and second, the US-based cohort of autistic individuals (SPARK[39]; $n_{total} = 28,165$; Extended Data Figs. 5 and 6), which recruits families with at least one autistic individual through online platforms and medical centres across the United States. In SPARK, we conducted initial analyses in a discovery subset of 18,809 autistic individuals (SPARK Discovery), and replicated key findings in a second sample of 9,356 autistic individuals that became available only after the initial analyses were completed (SPARK Replication). SPARK and iPSYCH differed in the diagnostic classification system (iPSYCH: International Classification of Diseases (ICD) and SPARK: Diagnostic and Statistical Manual of Mental Disorders (DSM)) and median age at diagnosis (iPSYCH, median = 10 years, median absolute deviation = 4 years; SPARK, median = 4 years, median absolute deviation = 2.7 years).

We conducted a genome-wide association study (GWAS) in iPSYCH and across both the Discovery and Replication samples of SPARK, using age at autism diagnosis as a quantitative trait. In all three samples, we identified significant and consistent SNP-based heritability of approximately 11% for age at autism diagnosis (Fig. 2a and Supplementary Table 10). This is larger than, or similar to, the variance explained by several other clinical and sociodemographic factors tested in SPARK (Extended Data Fig. 1 and Supplementary Table 1).

In contrast to the effect of common genetic variants, in a subsample of SPARK with available data for both parents and their autistic child ($n = 6,206$ trios), we observed no association between age at autism diagnosis and rare de novo variants or inherited protein truncating or missense variants in highly constrained genes (Supplementary Table 11). This may possibly be the result of low statistical power or reflect later autism diagnosis in some carriers of de novo mutations owing to diagnostic overshadowing by co-occurring intellectual disability or global developmental delay[40,41].

We next tested whether this SNP-based heritability of age at autism diagnosis is consistent with either of the two theoretical models outlined earlier. The unitary model assumes that later diagnosis reflects subtle or less-severe clinical features, so the SNP-based heritability of age at autism diagnosis may simply reflect the severity of autism features. Alternatively, the SNP-based heritability may reflect additional genetic influences associated with co-occurring developmental delays, developmental regression or intellectual disability, which may lead to an earlier diagnosis. It may also reflect the heritable component of parental socioeconomic status and neighbourhood deprivation. which are proxies for parental awareness and healthcare access that affect diagnostic timing. Controlling for any of these measures should attenuate the SNP-based heritability.

By contrast, the developmental model assumes that SNP-based heritability of age at autism diagnosis reflects a mixture of different polygenic factors that are correlated with age at diagnosis but are independent of these covariates. Under this model, a significant SNP-based heritability should persist after controlling for clinical, developmental and sociodemographic measures.

In line with the developmental model, we found that the SNP-based heritability did not significantly attenuate after controlling for parental sociodemographic measures, clinical features or co-occurring developmental delays and conditions (Fig. 2b and Supplementary Table 10). This is inconsistent with the unitary model, although imperfect and incomplete measurement of clinical and developmental phenotypes may limit this conclusion.

A second prediction of the unitary model is that earlier diagnosis is associated with a greater polygenic propensity for autism compared with later diagnosis (Extended Data Fig. 2). In this model, all autistic individuals would have a higher polygenic propensity for autism compared with non-autistic controls. This would result in negative genetic correlations between GWAS of age at autism diagnosis and

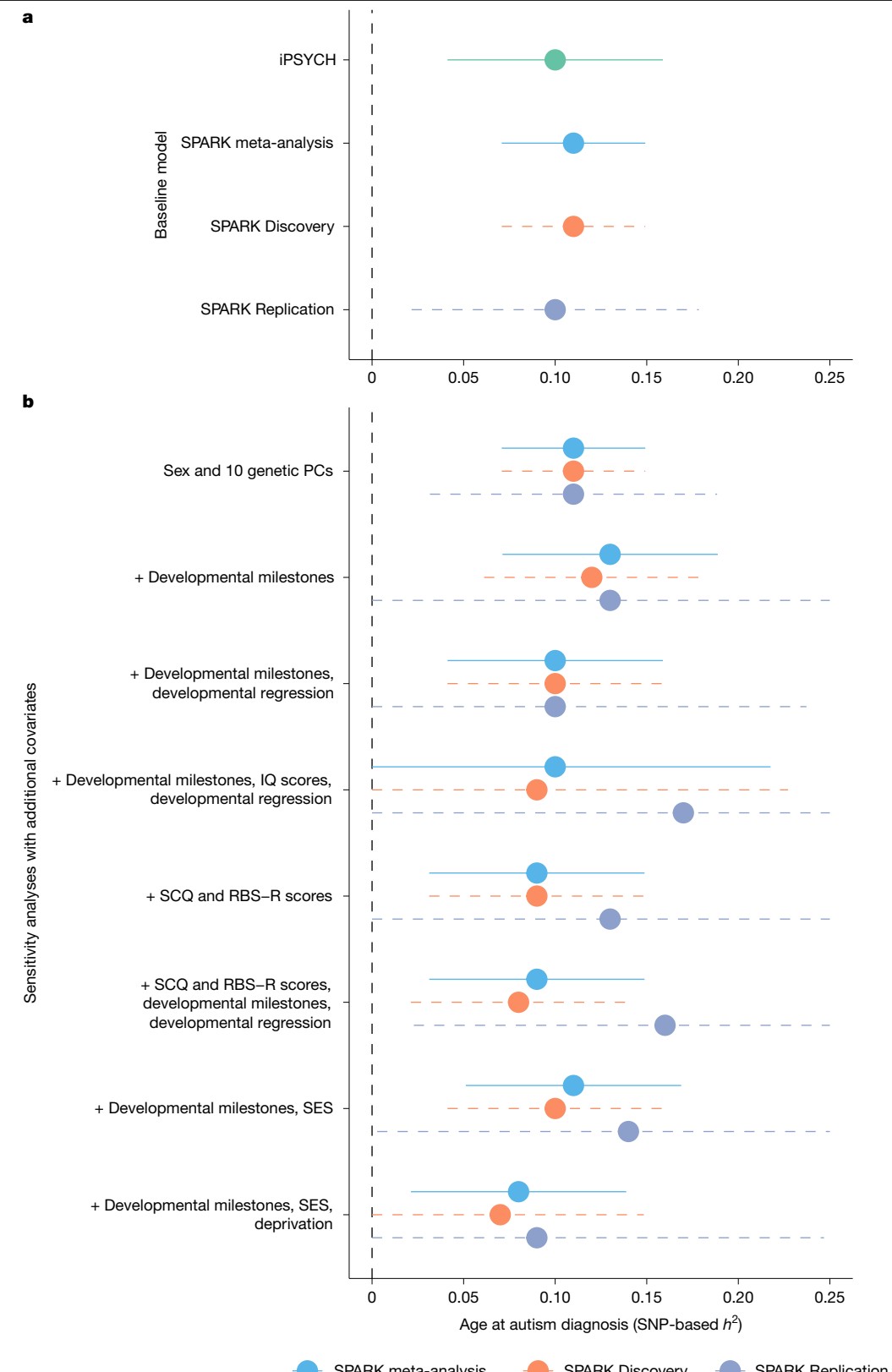

**Fig. 2 | Heritability of age at autism diagnosis. a**, SNP-based heritability ($h^2$) for age at autism diagnosis in the SPARK cohorts, calculated using single-component genome-wide complex trait analysis with a genomic-relatedness-based restricted maximum-likelihood approach (GCTA-GREML) for the SPARK Discovery cohort (orange dashed line, $n = 16,786$), SPARK Replication cohort (purple dashed line, $n = 8,558$) and a meta-analysis of the two (light blue solid line, $n = 25,344$), and iPSYCH, calculated using linkage disequilibrium score regression coefficient (LDSC) (solid green line, $n = 18,965$). **b**, SNP-based heritability (GCTA-GREML) in the SPARK cohorts after accounting for various clinical and sociodemographic factors. A '+' indicates the baseline model in addition to the specified covariates. The $x$ axis has been truncated at 0 and 0.25. In **a** and **b**, central points represent SNP-based heritability estimates and error bars indicate 95% confidence intervals. Sample sizes for **b** are provided in Supplementary Table 10. PC, genetic principal component; RBS-R, Repetitive Behavior Scale-Revised; SCQ, Social Communication Questionnaire; SES, socioeconomic status.

## Box 1

# Summary of the autism GWAS used in this study

**Autism GWAS not stratified by age at diagnosis**

- SPARK (Matoba et al.[49]): case-pseudocontrol design (4,535 pairs) with family-based ascertainment across the United States. Median age at diagnosis = 3.5 (median absolute deviation 1.97) years.
- FinnGen (Data Release r10): population-based sample from Finland (646 cases and 301,879 controls). Median age at diagnosis = 22.66 (7.16) years.
- PGC-2017 (ref. 45): meta-analyses of several case-control and case-pseudocontrol datasets (7,387 cases and 8,567 controls). Most of the cases met the diagnostic criteria for autism under DSM-IV-TR/ICD-10 or earlier (onset of features before age 3) after screening using the Autism Diagnostic Observation Schedule and the Autism Diagnostic Interview-Revised), making this a clinically well-characterized cohort. Although age at diagnosis is unavailable, most of the participants were recruited as trios through medical or research centres in the United States. Given this similarity in ascertainment to SPARK, we anticipate the age at diagnosis to be similar to that of SPARK trios[49]. Approximate median age at diagnosis = 3.5 years.
- iPSYCH$_{unstratified}$[50]: population-based sample derived from Danish national registries, including individuals born in 1980–2008 (19,870 autistic and 39,078 non-autistic individuals). Median age at diagnosis = 10.74 years (5.35).
- iPSYCH$_{males}$[50]: males-only subset of iPSYCH (15,025 autistic and 19,763 controls). Median age at diagnosis = 10.08 years (5.07).
- iPSYCH$_{females}$[50]: females-only subset of iPSYCH (4,845 autistic and 19,315 controls). Median age at diagnosis = 12.97 years (4.75).
- Grove et al.[50]: meta-analysis of a subset of the iPSYCH and PGC samples (18,381 cases and 27,969 controls). Age at diagnosis for PGC is unavailable, so we calculated an estimated age by weighing the median age at diagnosis of iPSYCH with the estimated median age in PGC by their respective sample size. Approximate median age at diagnosis = 8.73 years.

**Autism GWAS stratified by age at diagnosis**

SPARK (using unaffected family members as controls), meta-analysed from the Discovery and Replication subsets.

- Diagnosed before age 11 (SPARK$_{before11}$): 27,881 autistic individuals; selected to match iPSYCH stratification; median age at diagnosis = 3.5 (1.97) years. This cut-off period was chosen to reflect the cut-off used in the latent growth curve models and represents a time window characterized by the onset of puberty, the transition from primary to secondary school and an increase in the number of autistic girls being diagnosed.
- Diagnosed after age 10 (SPARK$_{after10}$): 6,243 autistic individuals; median age at diagnosis = 15.83 years (7.16).
- Diagnosed before age 6 (SPARK$_{before6}$): 21,435 autistic individuals; categorized as 'early-diagnosed' based on previous research[8]; median age at diagnosis = 3 (1.23) years.

iPSYCH (population-based controls)

- Diagnosed before age 11 (iPSYCH$_{before11}$): 9,500 autistic and 36,667 non-autistic individuals; median age at diagnosis = 7.34 (2.76) years. The cut-off was chosen to reflect the cut-off used in the latent growth curve models and SPARK age at diagnosis-stratified GWAS.
- Diagnosed after age 10 (iPSYCH$_{after10}$): 9,231 autistic and 36,667 non-autistic individuals; median age at diagnosis = 14.55 (2.84) years.
- Diagnosed before age 9 (iPSYCH$_{before9}$): 5,451 autistic and 36,667 non-autistic individuals; created to provide additional age resolution; median age at diagnosis = 5.74 (1.69) years.

GWAS of autism, with the magnitude of this negative correlation decreasing as the median age at diagnosis in the autism GWAS samples increases.

We tested this prediction using 13 different but partly overlapping autism GWASs, including six GWASs stratified by age at diagnosis and two GWASs stratified by sex (Box 1 and Fig. 3). The genetic correlation between age at autism diagnosis and different autism GWASs varied systematically, becoming increasingly positive as the median age at diagnosis increased (Fig. 3 and Supplementary Table 12). However, contrary to the expectation under the unitary model, we observed positive genetic correlations between age at autism diagnosis and autism GWAS comprising later-diagnosed autistic individuals. These findings support the existence of different genetic architectures across diagnostic age groups, aligning with the developmental model.

Furthermore, the male- and female-stratified autism GWAS from iPSYCH had a similar genetic correlation with age at autism diagnosis. This indicates that the pattern of genetic correlation between age at diagnosis and the autism GWASs does not reflect differences in the sex ratio of participants across the autism GWASs.

The genetic correlation with some of the autism GWASs differs significantly between the age at diagnosis GWASs in iPSYCH versus meta-analysed SPARK (Fig. 3). This reflects the fact that these two age at diagnosis GWASs are only moderately genetically correlated with each other (genetic correlation ($r_g$) = 0.51, standard error (s.e.) = 0.19, $P = 7.56 \times 10^{-3}$), which may be due to the different recruitment strategies and resulting differences in the median age at autism diagnosis in the two cohorts (Supplementary Note 6).

## Two autism polygenic factors

These findings indicate that the age at autism diagnosis reflects a mixture of different age-dependent polygenic traits (developmental model), rather than a single polygenic trait (unitary model). In the developmental model, one would expect that the genetic correlations between different autism GWASs will differ according to the difference in the median age of diagnosis between the GWASs.

In Aim 3, we tested this by estimating genetic correlations among the 13 autism GWASs (Box 1). We observed genetic correlations ranging from 0.02 (s.e. = 0.13) to 1.00 (s.e. = 0.01) (Fig. 4a and Supplementary Table 13). We observed a gradient in the genetic correlations related to the similarity in median age at diagnosis between cohorts. Cohorts with the most similar median ages at diagnosis (differing by a maximum of 2 years) showed the highest genetic correlations ($r_g$ =1, s.e. = 0.04), and correlations progressively decreased as age differences increased. The lowest genetic correlation ($r_g$ = 0.02, s.e. = 0.13) was observed between SPARK$_{before6}$ (median age of around 3) and SPARK$_{after10}$ (median age of around 16).

Hierarchical clustering of the genetic correlations identified two broad, overlapping clusters that differed by age at autism diagnosis. One cluster comprised GWAS of autism in cohorts with predominantly childhood-diagnosed individuals, and the other comprised GWAS of autism in cohorts with a large fraction of individuals diagnosed in adolescence or later, consistent with predictions from the developmental model.

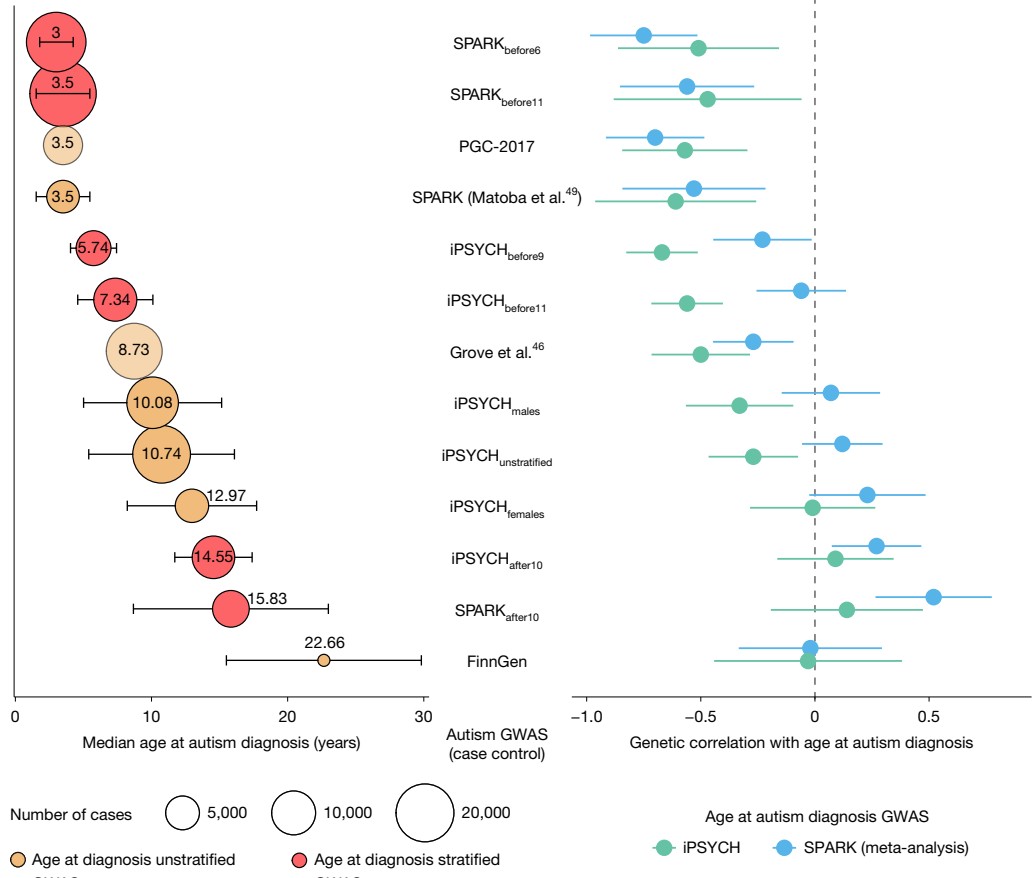

**Fig. 3 | Median age at autism diagnosis and genetic correlations with age at autism diagnosis across different GWAS cohorts.** Left, median age at diagnosis (years) with error bars representing median absolute deviation. Circle size indicates the number of autistic individuals (cases) in the GWAS, and exact sample sizes are provided in Box 1. Beige circles represent GWAS unstratified by age at diagnosis; red circles represent GWAS stratified by age at diagnosis. Lighter (more transparent) circles indicate studies with no information about age at autism diagnosis, and the median ages have been inferred from other available information (PGC-2017 (ref. 45) and Grove et al.[46]). Right, genetic correlations with age at autism diagnosis for both SPARK (blue, meta-analysis, $n = 28,165$) and iPSYCH (green, $n = 18,965$) datasets, with error bars representing 95% confidence intervals.

We formally tested this by modelling the genetic covariance using the structural equation models in GenomicSEM[42], testing six theoretical models (Supplementary Table 14). We used six minimally overlapping GWASs for autism with wide variation in age at autism diagnosis among those listed in Box 1. We found that a correlated two-factor model was the most parsimonious and fit the data best (Akaike information criterion, 38.64; confirmatory fit index, 0.99; standardized root mean residual, 0.08; Fig. 4b). Factor 1 (earlier-diagnosed autism factor) was defined by the GWAS with predominantly early childhood-diagnosed individuals (PGC-2017, SPARK_before6, with a median age at diagnosis of 3). Factor 2 (later-diagnosed autism factor) was defined primarily by GWASs with adolescent- or adult-diagnosed individuals (iPSYCH_after10, FinnGen and SPARK_after10). The cross-loading of iPSYCH_before9 (median age at diagnosis of around 5.7) indicates that factor 2 may impact behaviours in mid-to-late childhood as well. The two factors had a small genetic correlation ($r_g = 0.38$, s.e. = 0.06). Sensitivity analyses confirmed the robustness of the above results using partly different GWASs, in which we identified a two-correlated-factor model as the best-fitting model, with similar moderate genetic correlations between the two factors ($r_g = 0.37$, s.e. = 0.06 to $r_g = 0.52$, s.e. = 0.10; Supplementary Table 14).

The earlier-diagnosed autism factor was negatively genetically correlated with age at autism diagnosis (Fig. 5). The later-diagnosed autism factor was positively genetically correlated only with age at autism diagnosis from SPARK. Genetic correlation with the autism

GWAS stratified by sex showed that both autism factors had stronger genetic correlations with autism in male than in female individuals, with a larger difference for the earlier-diagnosed autism factor (Fig. 5), consistent with established sex differences in age at autism diagnosis.

Further analyses using polygenic scores confirmed that the association between the two polygenic autism factors and age at autism diagnosis is not due to several confounding factors using within-family approaches, nor to differences in clinical and demographic factors, and co-occurring developmental conditions (Supplementary Note 7 and Supplementary Tables 15–17). Taken together, the above findings demonstrate that earlier- and later-diagnosed autism have different polygenic aetiologies, supporting the developmental model.

## Genetic correlations with autism factors

The above analyses indicate that there are at least two polygenic factors associated with age at autism diagnosis. Because later-diagnosed autistic individuals have higher rates of mental-health conditions[15,16], we proposed that this might partly be because earlier- and later-diagnosed autism factors have differing genetic correlations with mental health and cognitive phenotypes. In Aim 4, we investigated this hypothesis using genetic correlation analyses.

The earlier-diagnosed autism factor (factor 1) had a low ($r_g$ of around 0.1–0.2) but significant genetic correlation with educational attainment, cognitive aptitude, ADHD and various mental-health

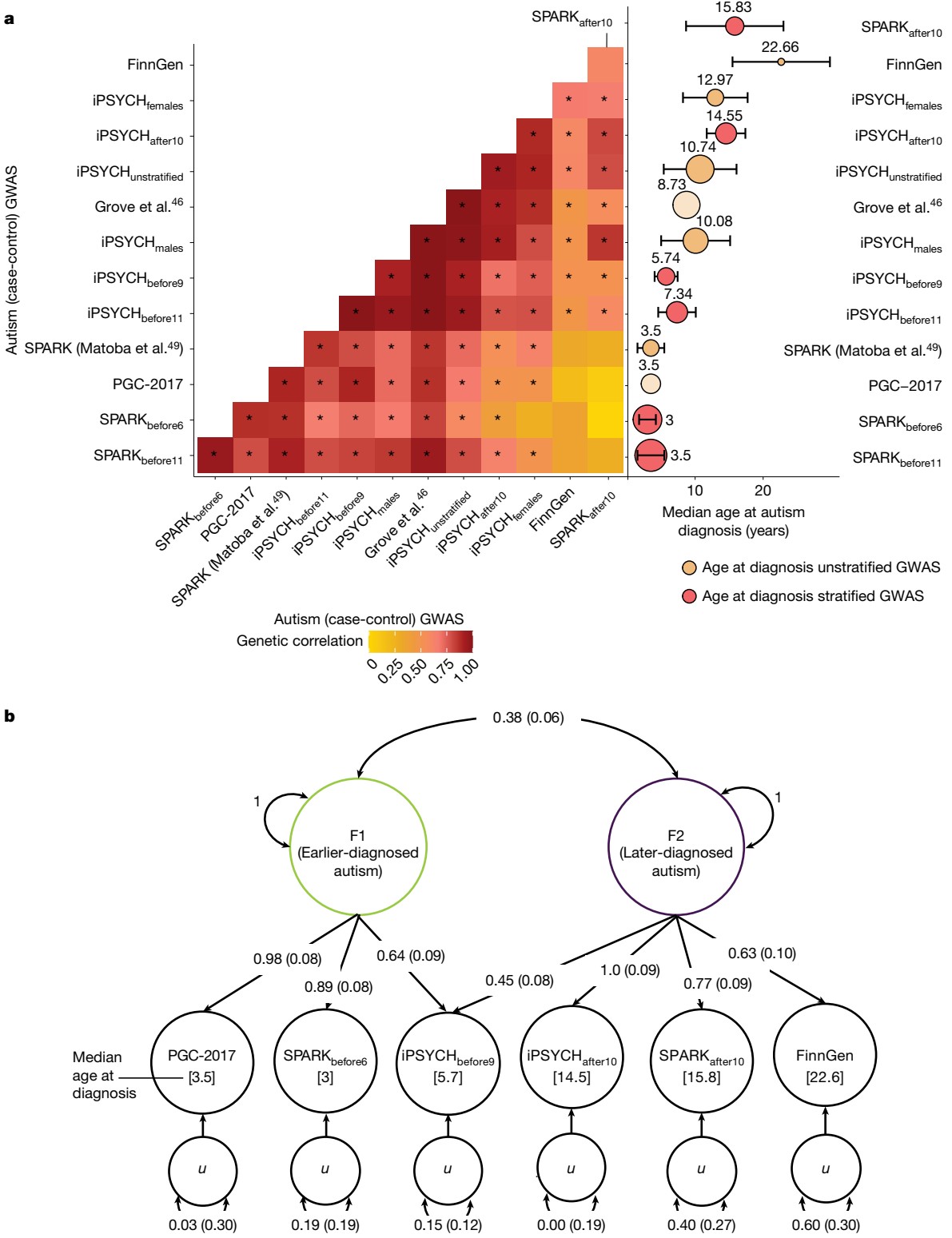

**Fig. 4 | Two genetic latent factors in autism. a**, Left, genetic correlation heatmaps of all GWASs of autism as described in Box 1. Asterisks indicate significant genetic correlations after Benjamini–Yekutieli adjustment. Right, median age at autism diagnosis for the same GWAS (indicated by the number on top of the circle). Error bars indicate median absolute deviation; the size of the circles indicate the sample size. For both panels, GWASs have been ordered based on hierarchical clustering of the genetic correlations. **b**, Structural equation model illustrating the two-correlated genetic-factor models for autism, using six minimally overlapping autism GWAS datasets. F1, factor 1;

F2, factor 2. One-headed arrows depict the regression relationship pointing from the independent variables to the dependent variables; the numbers on the arrows represent the regression coefficients of the factor loadings, with standard errors provided in parentheses. Covariance between variables is represented by two-headed arrows linking the variables. The numbers on the two-headed arrows can be interpreted as genetic-correlation estimates with the standard error provided in parentheses. Residual variances for each GWAS dataset are represented using a two-headed arrow connecting the residual variable (*u*) to itself. Standard errors are shown in parentheses.

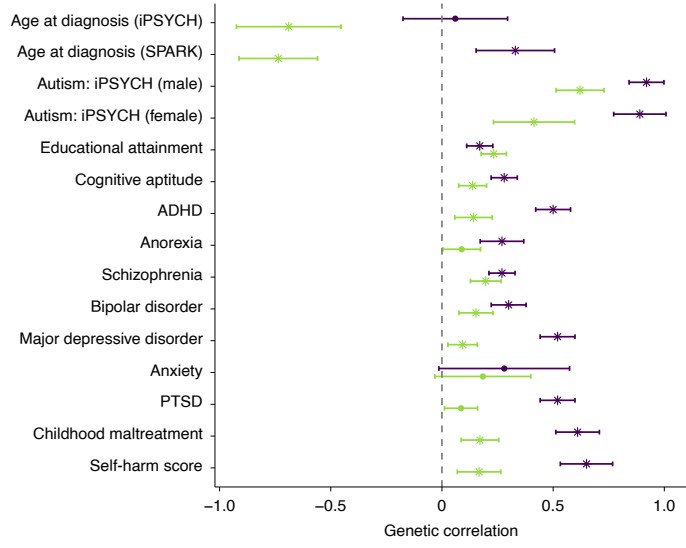

**Fig. 5 | Genetic correlation between the two autism polygenic factors and a range of mental-health, neurodevelopmental and cognitive traits.** Central points indicate the estimate (genetic correlation), error bars indicate 95% confidence intervals and asterisks indicate significant *P*-values (two-sided) with Benjamini–Yekutieli adjustment. Sample sizes are shown in Supplementary Table 18.

and related conditions (Fig 5 and Supplementary Table 18). The later-diagnosed autism factor (factor 2) showed a statistically similar genetic correlation with educational attainment but significantly higher genetic correlations ($r_g$ of around 0.5–0.7) with ADHD and a range of other mental-health and related conditions, including depression, post-traumatic stress disorder (PTSD), childhood maltreatment and self-harm. After accounting for genetic effects on ADHD, we saw an attenuated but significant moderate genetic correlation between later-diagnosed autism (factor 2) and mental-health conditions, indicating that shared genetics with ADHD do not fully explain the elevated correlation between later-diagnosed autism and mental-health phenotypes (Supplementary Table 18 and Supplementary Fig. 11). Sensitivity analyses using age at diagnosis-stratified GWASs from iPSYCH and SPARK yielded largely consistent genetic correlation results, indicating that these results are not due to cohort differences (Supplementary Table 19 and Extended Data Fig. 7).

The higher genetic correlation between later-diagnosed autism and other mental-health conditions may indicate diagnostic misclassification (in which individuals with other conditions incorrectly receive an autism diagnosis) or diagnostic overshadowing (where the presence of co-occurring mental-health conditions can delay an autism diagnosis). In the unitary model, the genetics of later-diagnosed autism would reflect the additive genetic effects of earlier-diagnosed autism and of other mental-health conditions. However, decomposition of the autism genetic signal using genomicSEM indicated that later-diagnosed autism cannot be entirely attributed to the polygenic effects of earlier-diagnosed autism and six other mental-health conditions tested: schizophrenia, ADHD, anorexia nervosa, depression, bipolar disorder and PTSD (Supplementary Note 8). Thus, the later-diagnosed autism genetic factor does not represent the additive genetic effects of earlier-diagnosed autism and mental-health conditions.

Finally, we investigated whether the two autism polygenic factors related to age at diagnosis differed in their associations with developmental traits. Cross-sectionally, these genetic factors showed few significant differences in genetic correlation or polygenic score association with developmental phenotypes measured at age 3 or earlier. The exceptions were age at onset of walking[43] and expressive vocabulary at

an age of 2–3 years[44], with which the earlier-diagnosed autism factor was positively genetically correlated but the later-diagnosed factor was not (Supplementary Note 9 and Supplementary Tables 20–22). However, longitudinal polygenic score analyses across two birth cohorts revealed differential genetic effects of earlier- versus later-diagnosed autism factors on SDQ total difficulties scores over time (Supplementary Note 9 and Supplementary Table 23).

## Discussion

The results of this study indicate that earlier- and later-diagnosed autism are associated with different developmental trajectories, and are only moderately genetically correlated with each other. This possible framework provides one axis of heterogeneity to describe the widely acknowledged clinical and genetic heterogeneity within autism that thus far has been challenging to identify. This finding of two or more developmentally variable polygenic latent traits for autism is robust to various observed clinical and demographic factors (Supplementary Note 10), including sex and intellectual disability.

Consistent with the wider literature on developmental variation in autism[22], our analyses of socioemotional and behavioural trajectories across multiple birth cohorts converge with the genetic findings: polygenic scores for earlier- and later-diagnosed autism have different associations with developmental changes in SDQ total difficulties scores (Supplementary Note 9). These results indicate that the timing of autism diagnosis may partly reflect aetiologically different developmental pathways, rather than purely environmental or diagnostic factors. Our findings are consistent with the wider literature that demonstrates that genetic influences on traits related to autism vary across development in the general population[19,20].

This two-polygenic-trait genetic model provides one framework to understand genetic heterogeneity in autism and the varying patterns of genetic correlations between different GWASs of autism and other phenotypes. For example, previous GWASs of autism (including PGC-2017 (ref. 45)) found limited genetic correlation with ADHD, contrary to findings from more-recent autism GWAS (such as Grove et al.[46]). We show that this is explained by the different average age at diagnosis across these GWASs (Box 1 and Fig. 3), because the genetic correlation between autism and ADHD increases with later age at autism diagnosis (Fig. 5). These findings were confirmed using within-family analyses that demonstrated over-transmission of ADHD polygenic scores, mainly to individuals with a later autism diagnosis (Supplementary Table 24).

Both the later-diagnosed genetic autism factor (Fig. 4) and the late childhood emergent latent trajectory of SDQ total difficulties scores (Fig. 1) are associated with greater mental-health problems (Fig. 5 and Supplementary Table 7). This indicates that epidemiological findings of greater mental-health difficulties among later-diagnosed autistic individuals[15,16] may be partly explained by the developmental model of autism. Given that autistic female individuals are, on average, diagnosed later in life, research that investigates sex and gender differences in both autism and co-occurring conditions[16,47] needs to account for genetic confounding associated with age at autism diagnosis. Findings that may seem to reflect sex differences may also partly reflect differences associated with age at diagnosis. For example, the higher prevalence of mental-health problems in autistic female individuals[16,47] compared with male individuals attenuates when restricting to autistic individuals diagnosed before age the age of 5 (ref. 16).

These findings must be interpreted considering several limitations. First, the SNP-based heritability for age at autism diagnosis is only about 11% (Fig. 2), and other observed developmental and demographic factors typically explain less than 15% of the variance (Extended Data Fig. 2). This indicates that there are several other factors that contribute to age at autism diagnosis. We find that the genetic effects on age at autism diagnosis are not mediated by several of these measured developmental and demographic factors, but we acknowledge that there may

be several unmeasured factors that may mediate the genetic effects. Furthermore, the substantial variation across the datasets explored highlights that age at autism diagnosis is immensely complex and varies across geography and time. Local cultural factors, access to health care, gender bias, stigma, ethnicity and camouflaging probably have an effect on who receives a diagnosis and when. Second, our trajectory models (Fig. 1) were built using only the SDQ, which measures a wide range of parent-reported neurodevelopmental and mental-health traits. Although the SDQ is correlated with an autism diagnosis, it does not fully capture core autistic traits, and other measures of autistic traits were not available in the birth cohorts. Third, it is likely that other dimensions contribute to the genetic heterogeneity in autism. For example, a significant proportion of the variation in the FinnGen autism GWAS was not explained by either of the two factors (Fig. 4). Fourth, we use earlier- and later-diagnosed autism as relative terms, reflecting that developmental and polygenic differences represent a gradient (Fig. 4 and Supplementary Fig. 10), rather than being discrete categories, and because there is no consensus on age thresholds for early versus late diagnosis[48]. Fifth, autism diagnoses in the birth cohorts used in the current study rely on community-based carer or self-reporting, rather than standardized clinical assessments. As such, there may be varying delays between the emergence of autistic features and a formal autism diagnosis. However, our findings can guide future research using longitudinal cohorts, and particularly sibling studies, that systematically track the emergence of autism features over time. Finally, our genetic analyses focused on common genetic variants in genetically inferred European ancestries, because we had only limited GWAS data from other populations, highlighting the need for future research to examine the transferability of these findings across diverse genetic ancestries.

In conclusion, we find that the developmental trajectories and polygenic architecture of autism varies with age at diagnosis. These findings partly explain the varying genetic correlations among the different GWASs of autism and between autism and various mental-health conditions. These findings provide further support for the hypothesis that the umbrella term 'autism' describes multiple phenomena with differing aetiologies, developmental trajectories and correlations with mental-health conditions. These findings have implications for how we conceptualize neurodevelopment more broadly, and for understanding diagnosis, sex and gender differences, and co-occurring health profiles in autism.

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

¹Department of Psychiatry, University of Cambridge, Cambridge, UK. ²Autism Research Centre, Department of Psychiatry, University of Cambridge, Cambridge, UK. ³iPSYCH, The Lundbeck Foundation Initiative for Integrative Psychiatric Research, Aarhus, Denmark. ⁴Center for Genomics and Personalized Medicine (CGPM), Aarhus University, Aarhus, Denmark. ⁵Department of Biomedicine (Human Genetics), Aarhus University, Aarhus, Denmark. ⁶Bioinformatics Research Centre, Aarhus University, Aarhus, Denmark. ⁷Centre for Integrative Sequencing (iSEQ), Aarhus University, Aarhus, Denmark. ⁸Human Genetics Programme, Wellcome Sanger Institute, Wellcome Genome Campus, Hinxton, UK. ⁹Language and Genetics Department, Max Planck Institute for Psycholinguistics, Nijmegen, The Netherlands. ¹⁰Department of Systems Medicine, University of Rome Tor Vergata, Rome, Italy. ¹¹Centre for Brain and Cognitive Development, Department of Psychological Sciences, Birkbeck University of London, London, UK. ¹²Research Department, Lovisenberg Diaconal Hospital, Oslo, Norway. ¹³PsychGen Centre for Genetic Epidemiology and Mental Health, Norwegian Institute of Public Health, Oslo, Norway. ¹⁴Program in Neurobehavioral Genetics and Center for Autism Research and Treatment, Semel Institute, David Geffen School of Medicine, University of California, Los Angeles, Los Angeles, CA, USA. ¹⁵Program in Neurogenetics, Department of Neurology, David Geffen School of Medicine, University of California, Los Angeles, Los Angeles, CA, USA. ¹⁶Department of Psychiatry, Semel Institute, David Geffen School of Medicine, University of California, Los Angeles, Los Angeles, CA, USA. ¹⁷Department of Human Genetics, David Geffen School of Medicine, University of California, Los Angeles, Los Angeles, CA, USA. ¹⁸Institute for Molecular Bioscience, University of Queensland, Brisbane, Queensland, Australia. ¹⁹Department of Psychiatry, University of Oxford, Oxford, UK. ²⁰Promenta Research Centre, Department of Psychology, University of Oslo, Oslo, Norway. ²¹School of Psychology, University of Surrey, Guildford, UK. ²²MRC Integrative Epidemiology Unit, University of Bristol, Bristol, UK. ²³Donders Institute for Brain, Cognition and Behaviour, Radboud University, Nijmegen, The Netherlands. ²⁴Center for Genomic Medicine, Massachusetts General Hospital, Boston, MA, USA. ²⁵The Broad Institute of MIT and Harvard, Cambridge, MA, USA. ²⁶Human Genetics and Cognitive Functions, Institut Pasteur, UMR3571 CNRS, IUF, Université Paris Cité, Paris, France. ²⁷Department of Psychology, University of Cambridge, Cambridge, UK. ²⁸Wellcome Trust Sanger Institute, Hinxton, UK.

**APEX Consortium**

Simon Baron-Cohen[1,2,27], Carrie Allison[2], Varun Warrier[1,2,27], Alex Tsompanidis[2], Deep Adhya[2], Rosemary Holt[2], Yuanjun Gu[1,2], Xinhe Zhang[1,2], Omar Al-Rubaie[2], Daniel H. Geschwind[14,15,16,17], Ramin Ali Marandi Ghoddousi[16], Alexander E. P. Heazell[29], Jonathan Mill[30], Alice Franklin[30], Rosie Bamford[30], Matthew E. Hurles[28], Hilary C. Martin[8,28], Mahmoud Mousa[28], David H. Rowitch[31], Kathy K. Niakan[32], Graham J. Burton[32], Tereza Cindrova-Davies[32], Deepak P. Srivastava[33], Lucia Dutan-Polit[33], Adam Pavlinek[33], Laura Sichlinger[33], Roland Nagy[33], Madeline A. Lancaster[34], Jose Gonzalez-Martinez[34], Tal Biron-Shental[35], Lidia V. Gabis[36,37], Dorothea Floris[23,38], Richard Bethlehem[27], Michael V. Lombardo[39], Marcin Radecki[2], Meng-Chuan Lai[2,40,41,42,43], Yeshaya David Greenberg[2], Elizabeth Weir[2], Florina Uzefovsky[44], Yumnah T. Khan[2] & Juan Pablo Del Rio[45]

[29]Obstetrics, Tommy's Maternal and Fetal Research Centre, University of Manchester, Manchester, UK. [30]University of Exeter Medical School, College of Medicine & Health, University of Exeter, Exeter, UK. [31]Department of Pediatrics, University of Cambridge, Cambridge, UK. [32]Centre for Trophoblast Research, University of Cambridge, Cambridge, UK. [33]MRC Centre for Neurodevelopmental Disorders, King's College London, London, UK. [34]MRC Laboratory of Molecular Biology, University of Cambridge, Cambridge, UK. [35]Department of Obstetrics and Gynecology, Meir Medical Center, Kefar Sava, Israel. [36]Faculty of Medical & Health Sciences, Tel-Aviv University, Tel Aviv, Israel. [37]Maccabi Healthcare, Tel Aviv, Israel. [38]Department of Psychology, University of Zurich, Zurich, Switzerland. [39]Laboratory for Autism and Neurodevelopmental Disorders, Center for Neuroscience and Cognitive Systems, Istituto Italiano Di Tecnologia, Rovereto, Italy. [40]Margaret and Wallace McCain Centre for Child, Youth & Family Mental Health and Azrieli Adult Neurodevelopmental Centre, Campbell Family Mental Health Research Institute, Centre for Addiction and Mental Health, Toronto, Ontario, Canada. [41]Department of Psychiatry, The Hospital for Sick Children, Toronto, Ontario, Canada. [42]Department of Psychiatry, Temerty Faculty of Medicine, University of Toronto, Toronto, Ontario, Canada. [43]Department of Psychiatry, National Taiwan University Hospital and College of Medicine, Taipei, Taiwan. [44]Psychology Department, Ben-Gurion University of the Negev, Be'er Sheva, Israel. [45]Department of Child and Adolescent Psychiatry and Mental Health, University of Chile, Santiago, Chile.

**iPSYCH Autism Consortium**

Anders D. Børglum[3,4,5,7], Jonas Bybjerg-Grauholm[46], Jakob Grove[3,4,5,6,7], David M. Hougaard[46], Ole Mors[3,47,48], Preben Bo Mortensen[3,7,49,50,51], Merete Nordentoft[3,52,53,54] & Thomas Werge[3,53,55,56,57]

[46]Danish Center for Neonatal Screening, Department of Congenital Disorders, Statens Serum Institut, Copenhagen, Denmark. [47]Aarhus University Hospital–Psychiatry, Psychosis Research Unit, Aarhus, Denmark. [48]Psychosis Research Unit, Aarhus University Hospital – Psychiatry, Aarhus, Denmark. [49]NCRR, National Centre for Register-Based Research, Aarhus University, Aarhus, Denmark. [50]Centre for Integrated Register-based Research, Aarhus University, Aarhus, Denmark. [51]National Centre for Register-Based Research, Aarhus University, Aarhus, Denmark. [52]CORE - Copenhagen Research Center for Mental Health, Mental Health Centre Copenhagen, Copenhagen University Hospital, Copenhagen, Denmark. [53]Department of Clinical Medicine, Faculty of Health and Medical Sciences, University of Copenhagen, Copenhagen, Denmark. [54]Mental Health Services in the Capital Region of Denmark, University of Copenhagen, Copenhagen, Denmark. [55]Institute of Biological Psychiatry, Mental Health Services, Copenhagen University Hospital, Roskilde, Denmark. [56]Department of Clinical Medicine, University of Copenhagen, Copenhagen, Denmark. [57]The Globe Institute, Lundbeck Foundation Center for Geogenetics, University of Copenhagen, Copenhagen, Denmark.

**PGC-PTSD Consortium**

Caroline M. Nievergelt[58,59,60], Adam X. Maihofer[58,59,60], Elizabeth G. Atkinson[61], Chia-Yen Chen[62], Karmel W. Choi[63,64], Jonathan R. I. Coleman[65,66], Nikolaos P. Daskalakis[67,68,69], Laramie E. Duncan[70], Renato Polimanti[71,72], Cindy Aaronson[73], Ananda B. Amstadter[74], Soren B. Andersen[75], Ole A. Andreassen[76,77], Paul A. Arbisi[78,79], Allison E. Ashley-Koch[80], S. Bryn Austin[81,82,83], Esmina Avdibegovic[84], Dragan Babić[85], Silviu-Alin Bacanu[86], Dewleen G. Baker[58,59,87], Anthony Batzler[88], Jean C. Beckham[89,90,91], Sintia Belangero[92,93], Corina Benjet[94], Carisa Bergner[95], Linda M. Bierer[96], Joanna M. Biernacka[88,97], Laura J. Bierut[98], Jonathan I. Bisson[99], Marco P. Boks[100], Elizabeth A. Bolger[68,101], Amber Brandolino[102], Gerome Breen[65,66], Rodrigo Affonseca Bressan[93,104], Richard A. Bryant[105], Angela C. Bustamante[106], Jonas Bybjerg-Grauholm[46], Marie Bækvad-Hansen[3,107], Anders D. Børglum[3,4,5,7], Sigrid Børte[108,109], Leah Cahn[73], Joseph R. Calabrese[110,111], Jose Miguel Caldas-de-Almeida[112], Chris Chatzinakos[67,68,113], Sheraz Cheema[114], Lucía Colodro-Conde[117], Brandon J. Coombes[88], Carlos S. Cruz-Fuentes[118], Anders M. Dale[119], Shareefa Dalvie[120], Lea K. Davis[121], Jürgen Deckert[122], Douglas L. Delahanty[123], Michelle F. Dennis[89,90,91], Frank Desarnaud[73], Christopher P. DiPietro[67,113], Seth G. Disner[124,125], Anna R. Docherty[126,127], Katharina Domschke[128,129], Grete Dyb[77,130], Alma Džubur Kulenović[131], Howard J. Edenberg[132,133], Alexandra Evans[99], Chiara Fabbri[66,134], Negar Fani[135], Lindsay A. Farrer[136,137,138,139,140], Adriana Feder[73], Norah C. Feeny[141], Janine D. Flory[73], David Forbes[142], Carol E. Franz[58], Sandro Galea[143], Melanie E. Garrett[80], Bizu Gelaye[63], Joel Gelernter[144,145], Elbert Geuze[146,147], Charles F. Gillespie[135], Slavina B. Goleva[121,148], Scott D. Gordon[117], Aferdita Goçi[150], Lana Ruvolo Grasser[150], Camila Guindalini[151], Magali Haas[152], Saskia Hagenaars[65,66], Michael A. Hauser[89], Andrew C. Heath[153], Sian M. J. Hemmings[154,155], Victor Hesselbrock[156], Ian B. Hickie[157], Kelleigh Hogan[58,59,60], David Michael Hougaard[3,107], Hailiang Huang[67,158], Laura M. Huckins[159], Kristian Hveem[108], Miro Jakovljević[160], Arash Javanbakht[150], Gregory D. Jenkins[88], Jessica Johnson[161], Ian Jones[162], Tanja Jovanovic[135], Karen-Inge Karstoft[75,163], Milissa L. Kaufman[68,101], James L. Kennedy[164,165,166,167], Ronald C. Kessler[168], Alaptagin Khan[68,101], Nathan A. Kimbrel[89,91,169], Anthony P. King[170], Nastassja Koen[171], Roman Kotov[172],

Henry R. Kranzler[173,174], Kristi Krebs[175], William S. Kremen[58], Pei-Fen Kuan[176], Bruce R. Lawford[177], Lauren A. M. Lebois[68,69], Kelli Lehto[175], Daniel F. Levey[71,72], Catrin Lewis[99], Israel Liberzon[178], Sarah D. Linnstaedt[179], Mark W. Logue[139,180,181], Adriana Lori[135], Yi Lu[182], Benjamin J. Luft[183], Michelle K. Lupton[117], Jurjen J. Luykx[147,184], Iouri Makotkine[73], Jessica L. Maples-Keller[135], Shelby Marchese[185], Charles Marmar[186], Nicholas G. Martin[187], Gabriela A. Martínez-Levy[118], Kerrie McAloney[117], Alexander McFarlane[188], Katie A. McLaughlin[189], Samuel A. McLean[179,190], Sarah E. Medland[117], Divya Mehta[177,191], Jacquelyn Meyers[192], Vasiliki Michopoulos[135], Elizabeth A. Mikita[175], Lili Milani[175], William Milberg[193], Mark W. Miller[180,181], Rajendra A. Morey[194], Charles Phillip Morris[177], Ole Mors[3,47,48], Preben Bo Mortensen[3,7,49,50,51], Mary S. Mufford[120], Elliot C. Nelson[98], Merete Nordentoft[3,52,53,54], Sonya B. Norman[58,59,195], Nicole R. Nugent[196,197,198], Meaghan O'Donnell[142], Holly K. Orcutt[200], Matthew S. Panizzon[58], Pedro M. Pan[201], Gita A. Pathak[71,72], Edward S. Peters[202], Alan L. Peterson[203,204], Matthew Peverill[205], Robert H. Pietrzak[72,206], Melissa A. Polusny[78,125,207], Bernice Porjesz[192], Abigail Powers[135], Xue-Jun Qin[80], Andrew Ratanatharathorn[63,208], Victoria B. Risbrough[58,59,60], Andrea L. Roberts[209], Alex O. Rothbaum[210,211], Barbara O. Rothbaum[135], Peter Roy-Byrne[212], Kenneth J. Ruggiero[213], Ariane Rung[214], Heiko Runz[215], Bart P. F. Rutten[216], Stacey Saenz de Viteri[217], Giovanni Abrahão Salum[218,219], Laura Sampson[63,140], Sixto E. Sanchez[220], Marcos Santoro[221], Carina Seah[185], Soraya Seedat[154,222], Julia S. Seng[223,224,225,226], Andrey Shabalin[127], Christina M. Sheerin[74], Derrick Silove[227], Alicia K. Smith[135,228], Jordan W. Smoller[64,67,229], Scott B. Sponheim[78,230], Dan J. Stein[171], Synne Stensland[109,130], Jennifer S. Stevens[135], Jennifer A. Sumner[231], Martin H. Teicher[68,232], Wesley K. Thompson[233,234], Arun K. Tiwari[164,165,166], Edward Trapido[214], Monica Uddin[235], Robert J. Ursano[236], Unnur Valdimarsdóttir[237,238], Miranda Van Hooff[239], Eric Vermetten[240,241,242], Christiaan H. Vinkers[243,244,245], Joanne Voisey[177,191], Yunpeng Wang[246], Zhewu Wang[247,248], Monika Waszczuk[249], Heike Weber[122], Frank R. Wendt[250], Thomas Werge[3,53,55,56,57], Michelle A. Williams[63], Douglas E. Williamson[89,90], Bendik S. Winsvold[108,109,251], Sherry Winternitz[68,101], Christiane Wolf[122], Erika J. Wolf[181,252], Yan Xia[67,158], Ying Xiong[182], Rachel Yehuda[73,253], Keith A. Young[254,255], Ross McD. Young[256,257], Clement C. Zai[67,164,165,166,167,258], Gwyneth C. Zai[164,165,166,167,259], Mark Zervas[152], Hongyu Zhao[260], Lori A. Zoellner[205], John-Anker Zwart[77,108,109], Terri deRoon-Cassini[102], Sanne J. H. van Rooij[135], Leigh L. van den Heuvel[154,155], Murray B. Stein[58,87,261], Kerry J. Ressler[68,101,135] & Karestan C. Koenen[63,67,229]

[58]Department of Psychiatry, University of California, San Diego, La Jolla, CA, USA. [59]Center of Excellence for Stress and Mental Health, Veterans Affairs San Diego Healthcare System, San Diego, CA, USA. [60]Research Service, Veterans Affairs San Diego Healthcare System, San Diego, CA, USA. [61]Department of Molecular and Human Genetics, Baylor College of Medicine, Houston, TX, USA. [62]Translational Sciences, Biogen Inc., Cambridge, MA, USA. [63]Department of Epidemiology, Harvard T.H. Chan School of Public Health, Boston, MA, USA. [64]Department of Psychiatry, Massachusetts General Hospital, Boston, MA, USA. [65]National Institute for Health and Care Research Maudsley Biomedical Research Centre, South London and Maudsley NHS Foundation Trust, King's College London, London, UK. [66]Psychology and Neuroscience, Social, Genetic and Developmental Psychiatry Centre, Institute of Psychiatry, King's College London, London, UK. [67]Stanley Center for Psychiatric Research, Broad Institute of MIT and Harvard, Cambridge, MA, USA. [68]Department of Psychiatry, Harvard Medical School, Boston, MA, USA. [69]Center of Excellence in Depression and Anxiety Disorders, McLean Hospital, Belmont, MA, USA. [70]Department of Psychiatry and Behavioral Sciences, Stanford University, Stanford, CA, USA. [71]VA Connecticut Healthcare Center, West Haven, CT, USA. [72]Department of Psychiatry, Yale University School of Medicine, New Haven, Connecticut, USA. [73]Department of Psychiatry, Icahn School of Medicine at Mount Sinai, New York, NY, USA. [74]Department of Psychiatry Virginia Institute for Psychiatric and Behavioral Genetics, Richmond, VI, USA. [75]The Danish Veteran Centre, Research and Knowledge Centre, Ringsted, Denmark. [76]Division of Mental Health and Addiction, Oslo University Hospital, Oslo, Norway. [77]Institute of Clinical Medicine, University of Oslo, Oslo, Norway. [78]Minneapolis VA Health Care System, Mental Health Service Line, Minneapolis, MN, USA. [79]Department of Psychiatry, University of Minnesota, Minneapolis, MN, USA. [80]Duke Molecular Physiology Institute, Duke University, Durham, NC, USA. [81]Division of Adolescent and Young Adult Medicine, Boston Children's Hospital, Boston, MA, USA. [82]Department of Pediatrics, Harvard Medical School, Boston, MA, USA. [83]Department of Social and Behavioral Sciences, Harvard T.H. Chan School of Public Health, Boston, MA, USA. [84]Department of Psychiatry, University Clinical Center of Tuzla, Tuzla, Bosnia and Herzegovina. [85]Department of Psychiatry, University Clinical Center of Mostar, Mostar, Bosnia and Herzegovina. [86]Department of Psychiatry, Virginia Commonwealth University, Richmond, USA. [87]Psychiatry Service, Veterans Affairs San Diego Healthcare System, San Diego, CA, USA. [88]Department of Quantitative Health Sciences, Mayo Clinic, Rochester, MN, USA. [89]Department of Psychiatry and Behavioral Sciences, Duke University School of Medicine, Durham, NC, USA. [90]Research, Durham VA Health Care System, Durham, NC, USA. [91]Genetics Research Laboratory, VA Mid-Atlantic Mental Illness Research, Education, and Clinical Center (MIRECC), Durham, NC, USA. [92]Department of Morphology and Genetics, Universidade Federal de São Paulo, São Paulo, Brazil. [93]Laboratory of Integrative Neuroscience, Department of Psychiatry, Universidade Federal de São Paulo, São Paulo, Brazil. [94]Instituto Nacional de Psiquiatraía Ramón de la Fuente Muñiz, Center for Global Mental Health, Mexico City, Mexico. [95]Comprehensive Injury Center, Medical College of Wisconsin, Milwaukee, WI, USA. [96]Department of Psychiatry, James J. Peters VA Medical Center, Bronx, NY, USA. [97]Department of Psychiatry and Psychology, Mayo Clinic, Rochester, MN, USA. [98]Department of Psychiatry, Washington University in Saint Louis School of Medicine, Saint Louis, MO, USA. [99]MRC Centre for Psychiatric Genetics and Genomics, Cardiff University, National Centre for Mental Health, Cardiff, UK. [100]Department of Psychiatry, Brain Center University Medical Center Utrecht, Utrecht, The Netherlands. [101]McLean Hospital, Belmont, MA, USA. [102]Department of Surgery, Division of Trauma & Acute Care Surgery, Medical College of Wisconsin, Milwaukee, WI, USA. [103]King's College London, NIHR Maudsley BRC, London, UK. [104]Department of Psychiatry, Universidade Federal de São Paulo, São Paulo, Brazil. [105]School of Psychology, University of New South Wales, Sydney, New South Wales, Australia. [106]Department of Internal Medicine, Division of Pulmonary and Critical Care Medicine, University of Michigan Medical School, Ann Arbor, MI, USA. [107]Department for Congenital Disorders, Statens Serum Institut, Copenhagen, Denmark. [108]K. G. Jebsen Center for Genetic Epidemiology, Department of Public Health and

Nursing, Faculty of Medicine and Health Sciences, Norwegian University of Science and Technology, Trondheim, Norway. [109]Department of Research, Innovation and Education, Division of Clinical Neuroscience, Oslo University Hospital, Oslo, Norway. [110]School of Medicine, Case Western Reserve University, Cleveland, OH, USA. [111]Department of Psychiatry, University Hospitals, Cleveland, OH, USA. [112]Chronic Diseases Research Centre (CEDOC), Lisbon Institute of Global Mental Health, Lisbon, Portugal. [113]Division of Depression and Anxiety Disorders, McLean Hospital, Belmont, MA, USA. [114]CanPath National Coordinating Center, University of Toronto, Toronto, Ontario, Canada. [115]Family, Population and Preventive Medicine, Stony Brook University, Stony Brook, NY, USA. [116]Public Health, Stony Brook University, Stony Brook, NY, USA. [117]Mental Health & Neuroscience Program, QIMR Berghofer Medical Research Institute, Brisbane, Queensland, Australia. [118]Department of Genetics, Instituto Nacional de Psiquiatraía Ramón de la Fuente Muñiz Mexico City, Mexico City, Mexico. [119]Department of Radiology and Department of Neurosciences, University of California, San Diego, La Jolla, CA, USA. [120]Department of Pathology, Division of Human Genetics, University of Cape Town, Cape Town, South Africa. [121]Vanderbilt University Medical Center, Vanderbilt Genetics Institute, Nashville, TN, USA. [122]Center of Mental Health, Psychiatry, Psychosomatics and Psychotherapy, University Hospital of Würzburg, Würzburg, Denmark. [123]Department of Psychological Sciences, Kent State University, Kent, OH, USA. [124]Research Service Line, Minneapolis VA Health Care System, Minneapolis, MN, USA. [125]Department of Psychiatry & Behavioral Sciences, University of Minnesota Medical School, Minneapolis, MN, USA. [126]Huntsman Mental Health Institute, Salt Lake City, UT, USA. [127]Department of Psychiatry, University of Utah School of Medicine, Salt Lake City, UT, USA. [128]University of Freiburg, Faculty of Medicine, Centre for Basics in Neuromodulation, Freiburg, Denmark. [129]Department of Psychiatry and Psychotherapy, University of Freiburg Faculty of Medicine, Freiburg, Denmark. [130]Norwegian Centre for Violence and Traumatic Stress Studies, Oslo, Norway. [131]Department of Psychiatry, University Clinical Center of Sarajevo, Sarajevo, Bosnia and Herzegovina. [132]Biochemistry and Molecular Biology, Indiana University School of Medicine, Indianapolis, IN, USA. [133]Medical and Molecular Genetics, Indiana University School of Medicine, Indianapolis, IN, USA. [134]Department of Biomedical and Neuromotor Sciences, University of Bologna, Bologna, Italy. [135]Department of Psychiatry and Behavioral Sciences, Emory University, Atlanta, GA, USA. [136]Department of Medicine (Biomedical Genetics), Boston University Chobanian & Avedisian School of Medicine, Boston, MA, USA. [137]Department of Neurology, Boston University Chobanian & Avedisian School of Medicine, Boston, MA, USA. [138]Department of Ophthalmology, Boston University Chobanian & Avedisian School of Medicine, Boston, MA, USA. [139]Department of Biostatistics, Boston University School of Public Health, Boston, MA, USA. [140]Department of Epidemiology, Boston University School of Public Health, Boston, MA, USA. [141]Department of Psychological Sciences, Case Western Reserve University, Cleveland, OH, USA. [142]Department of Psychiatry, University of Melbourne, Melbourne, Victoria, Australia. [143]Boston University School of Public Health, Boston, MA, USA. [144]Psychiatry Service, VA Connecticut Healthcare Center, West Haven, CT, USA. [145]Department of Genetics and Neuroscience, Yale University School of Medicine, New Haven, CT, USA. [146]Netherlands Ministry of Defence, Brain Research and Innovation Centre, Utrecht, The Netherlands. [147]Department of Psychiatry, UMC Utrecht Brain Center Rudolf Magnus, Utrecht, The Netherlands. [148]National Human Genome Research Institute, National Institutes of Health, Bethesda, MD, USA. [149]Department of Psychiatry, University Clinical Centre of Kosovo, Prishtina, Kosovo. [150]Psychiatry and Behavioral Neurosciences, Wayne State University School of Medicine, Detroit, MI, USA. [151]Gallipoli Medical Research Foundation, Greenslopes Private Hospital, Greenslopes, Queensland, Australia. [152]Cohen Veterans Bioscience, New York, NY, USA. [153]Department of Genetics, Washington University in Saint Louis School of Medicine, Saint Louis, MO, USA. [154]Department of Psychiatry, Faculty of Medicine and Health Sciences, Stellenbosch University, Cape Town, South Africa. [155]SAMRC Genomics of Brain Disorders Research Unit, Stellenbosch University, Cape Town, South Africa. [156]Psychiatry, University of Connecticut School of Medicine, Farmington, CT, USA. [157]Brain and Mind Centre, University of Sydney, Sydney, New South Wales, Australia. [158]Analytic and Translational Genetics Unit, Department of Medicine, Massachusetts General Hospital, Boston, MA, USA. [159]Department of Psychiatry, Yale University, New Haven, CT, USA. [160]Department of Psychiatry, University Hospital Center of Zagreb, Zagreb, Croatia. [161]Genetics and Genomic Sciences, Icahn School of Medicine at Mount Sinai, New York, NY, USA. [162]Cardiff University Centre for Psychiatric Genetics and Genomics, National Centre for Mental Health, Cardiff University, Cardiff, UK. [163]Department of Psychology, University of Copenhagen, Copenhagen, Denmark. [164]Centre for Addiction and Mental Health, Neurogenetics Section, Molecular Brain Science Department, Campbell Family Mental Health Research Institute, Toronto, Ontario, Canada. [165]Centre for Addiction and Mental Health, Tanenbaum Centre for Pharmacogenetics, Toronto, Ontario, Canada. [166]Department of Psychiatry, University of Toronto, Toronto, Ontario, Canada. [167]Institute of Medical Sciences, University of Toronto, Toronto, Ontario, Canada. [168]Department of Health Care Policy, Harvard Medical School, Boston, MA, USA. [169]Mental Health Service Line, Durham VA Health Care System, Durham, NC, USA. [170]The Ohio State University College of Medicine, Institute for Behavioral Medicine Research, Columbus, OH, USA. [171]Department of Psychiatry & Neuroscience Institute, SA MRC Unit on Risk & Resilience in Mental Disorders, University of Cape Town, Cape Town, South Africa. [172]Department of Psychiatry, Stony Brook University, Stony Brook, NY, USA. [173]Mental Illness Research, Education and Clinical Center, Crescenz VAMC, Philadelphia, PA, USA. [174]Department of Psychiatry, University of Pennsylvania Perelman School of Medicine, Philadelphia, PA, USA. [175]Estonian Genome Center, Institute of Genomics, University of Tartu, Tartu, Estonia. [176]Department of Applied Mathematics and Statistics, Stony Brook University, Stony Brook, NY, USA. [177]School of Biomedical Sciences, Queensland University of Technology, Kelvin Grove, Queensland, Australia. [178]Department of Psychiatry and Behavioral Sciences, Texas A&M University College of Medicine, Bryan, TX, USA. [179]Department of Anesthesiology, UNC Institute for Trauma Recovery, Chapel Hill, NC, USA. [180]Psychiatry, Biomedical Genetics, Boston University School of Medicine, Boston, MA, USA. [181]National Center for PTSD, VA Boston Healthcare System, Boston, MA, USA. [182]Department of Medical Epidemiology and Biostatistics, Karolinska Institutet, Stockholm, Sweden. [183]Department of Medicine, Stony Brook University, Stony Brook, NY, USA. [184]Department of Translational Neuroscience, UMC Utrecht Brain Center Rudolf Magnus, Utrecht, The Netherlands. [185]Department of Genetic and Genomic Sciences, Icahn School of Medicine at Mount Sinai, New York, NY, USA. [186]Grossman School of Medicine, New York University, New York, NY, USA. [187]Genetics, QIMR Berghofer Medical Research Institute, Brisbane, Queensland, Australia. [188]Discipline of Psychiatry, University of Adelaide, Adelaide, South Australia, Australia. [189]Department of Psychology, Harvard University, Boston, MA, USA. [190]Department of Emergency Medicine, UNC Institute for Trauma Recovery, Chapel Hill, NC, USA. [191]Centre for Genomics and Personalised Health, Queensland University of Technology, Kelvin Grove, Queensland, Australia. [192]Department of Psychiatry and Behavioral Sciences, SUNY Downstate Health Sciences University, Brooklyn, NY, USA. [193]GRECC/TRACTS, VA Boston Healthcare System, Boston, MA, USA. [194]Duke Brain Imaging and Analysis Center, Duke University School of Medicine, Durham, NC, USA. [195]National Center for Post Traumatic Stress Disorder, Executive Division, White River Junction, VT, USA. [196]Department of Emergency Medicine, Alpert Brown Medical School, Providence, RI, USA. [197]Department of Pediatrics, Alpert Brown Medical School, Providence, RI, USA. [198]Department of Psychiatry and Human Behavior, Alpert Brown Medical School, Providence, RI, USA. [199]Department of Psychiatry, University of Melbourne, Phoenix Australia, Melbourne, Victoria, Australia. [200]Department of Psychology, Northern Illinois University, DeKalb, IL, USA. [201]Psychiatry, Universidade Federal de São Paulo, São Paulo, Brazil. [202]University of Nebraska Medical Center, College of Public Health, Omaha, NE, USA. [203]Research and Development Service, South Texas Veterans Health Care System, San Antonio, TX, USA. [204]Department of Psychiatry and Behavioral Sciences, University of Texas Health Science Center at San Antonio, San Antonio, TX, USA. [205]Department of Psychology, University of Washington, Seattle, WA, USA. [206]U.S. Department of Veterans Affairs National Center for Posttraumatic Stress Disorder, West Haven, CT, USA. [207]Center for Care Delivery and Outcomes Research (CCDOR), Minneapolis, MN, USA. [208]Department of Epidemiology, Columbia University Mailmain School of Public Health, New York, NY, USA. [209]Department of Environmental Health, Harvard T.H. Chan School of Public Health, Boston, MA, USA. [210]Department of Psychological Sciences, Emory University, Atlanta, GA, USA. [211]Department of Research and Outcomes, Skyland Trail, Atlanta, GA, USA. [212]Department of Psychiatry, University of Washington, Seattle, WA, USA. [213]Department of Nursing and Department of Psychiatry, Medical University of South Carolina, Charleston, SC, USA. [214]School of Public Health and Department of Epidemiology, Louisiana State University Health Sciences Center, New Orleans, LA, USA. [215]Research & Development, Biogen Inc., Cambridge, MA, USA. [216]School for Mental Health and Neuroscience, Department of Psychiatry and Neuropsychology, Maastricht Universitair Medisch Centrum, Maastricht, The Netherlands. [217]School of Public Health, SUNY Downstate Health Sciences University, Brooklyn, NY, USA. [218]Child Mind Institute, New York, NY, USA. [219]Instituto Nacional de Psiquiatria de Desenvolvimento, São Paulo, Brazil. [220]Department of Medicine, Universidad Peruana de Ciencias Aplicadas, Lima, Peru. [221]Departamento de Bioquímica, Disciplina de Biologia Molecular, Universidade Federal de São Paulo, São Paulo, Brazil. [222]SAMRC Extramural Genomics of Brain Disorders Research Unit, Stellenbosch University, Cape Town, South Africa. [223]Department of Obstetrics and Gynecology, University of Michigan, Ann Arbor, MI, USA. [224]Department of Women's and Gender Studies, University of Michigan, Ann Arbor, MI, USA. [225]Institute for Research on Women and Gender, University of Michigan, Ann Arbor, MI, USA. [226]School of Nursing, University of Michigan, Ann Arbor, MI, USA. [227]Department of Psychiatry, University of New South Wales, Sydney, New South Wales, Australia. [228]Department of Gynecology and Obstetrics, Department of Psychiatry and Behavioral Sciences and Department of Human Genetics, Emory University, Atlanta, GA, USA. [229]Psychiatric and Neurodevelopmental Genetics Unit (PNGU), Massachusetts General Hospital, Boston, MA, USA. [230]Department of Psychiatry and Behavioral Sciences, University of Minnesota Medical School, Minneapolis, MN, USA. [231]Department of Psychology, University of California, Los Angeles, Los Angeles, CA, USA. [232]Developmental Biopsychiatry Research Program, McLean Hospital, Belmont, MA, USA. [233]Mental Health Centre Sct. Hans, Institute of Biological Psychiatry, Roskilde, Denmark. [234]Herbert Wertheim School of Public Health and Human Longevity Science, University of California, San Diego, La Jolla, CA, USA. [235]Genomics Program, University of South Florida College of Public Health, Tampa, FL, USA. [236]Department of Psychiatry, Uniformed Services University, Bethesda, MD, USA. [237]Karolinska Institutet, Unit of Integrative Epidemiology, Institute of Environmental Medicine, Stockholm, Sweden. [238]Faculty of Medicine, Center of Public Health Sciences, School of Health Sciences, University of Iceland, Reykjavik, Iceland. [239]Adelaide Medical School, University of Adelaide, Adelaide, South Australia, Australia. [240]Psychotrauma Research Expert Group, ARQ Nationaal Psychotrauma Centrum, Diemen, The Netherlands. [241]Department of Psychiatry, Leiden University Medical Center, Leiden, The Netherlands. [242]Department of Psychiatry, New York University School of Medicine, New York, NY, USA. [243]Sleep & Stress Program, Amsterdam Neuroscience, Mood, Anxiety, Psychosis, Amsterdam, The Netherlands. [244]Department of Anatomy and Neurosciences, Vrije Universiteit Amsterdam, Amsterdam, The Netherlands. [245]Department of Psychiatry, Vrije Universiteit Amsterdam, Amsterdam, The Netherlands. [246]Lifespan Changes in Brain and Cognition (LCBC), Department of Psychology, University of Oslo, Oslo, Norway. [247]Department of Psychiatry and Behavioral Sciences, Medical University of South Carolina, Charleston, SC, USA. [248]Department of Mental Health, Ralph H Johnson VA Medical Center, Charleston, SC, USA. [249]Department of Psychology, Rosalind Franklin University of Medicine and Science, North Chicago, IL, USA. [250]Department of Anthropology, Dalla Lana School of Public Health, University of Toronto, Toronto, Ontario, Canada. [251]Department of Neurology, Oslo University Hospital, Oslo, Norway. [252]Department of Psychiatry, Boston University Chobanian & Avedisian School of Medicine, Boston, MA, USA. [253]Department of Mental Health, James J. Peters VA Medical Center, Bronx, NY, USA. [254]Research Service, Central Texas Veterans Health Care System, Temple, TX, USA. [255]Department of Psychiatry and Behavioral Sciences, Texas A&M University School of Medicine, Bryan, TX, USA. [256]School of Clinical Sciences, Queensland University of Technology, Kelvin Grove, Queensland, Australia. [257]The Chancellory, University of the Sunshine Coast, Sippy Downs, Queensland, Australia. [258]Department of Laboratory Medicine and Pathology, University of Toronto, Toronto, Ontario, Canada. [259]Centre for Addiction and Mental Health, General Adult Psychiatry and Health Systems Division, Toronto, Ontario, Canada. [260]Department of Biostatistics, Yale University, New Haven, CT, USA. [261]School of Public Health, University of California, San Diego, La Jolla, California, USA.

## Methods

### Terminology

We use the term autistic and non-autistic to refer to people with and without an autism diagnosis[51]. For sex, male and female refer to sex assigned at birth.

### Explaining variance in age at autism diagnosis

To contextualize the SNP-based heritability and the variance explained by the SDQ total difficulties and subscale scores, we reviewed the variance in the age at autism diagnosis explained by various sociodemographic and clinical factors, including sex and autism severity. Using Google Scholar and PubMed, we searched for studies published between 1998 and 10 December 2024 using combinations of the following terms in the title or abstract: 'age at diagnosis' AND 'autism'; 'autism' AND 'age'; and 'diagnosis age' AND 'autism'. We also used these search terms with alternative terminology for autism, including 'autism spectrum condition', 'autism spectrum disorder' and 'ASD'. This search resulted in more than 1,700 studies. A manual review identified 184 studies that investigated factors associated with age at autism diagnosis. Of these, 13 quantified the variance explained using measures of proportion of variance explained such as $R^2$ (coefficient of determination) or $\eta^2$ (effect size) and these were included in our final analyses (Supplementary Table 1).

We also calculated the variance explained in the US-based cohort of autistic individuals and their families, SPARK[39], using the v.9 release of the phenotypic data. We focused on the variance explained by sociodemographic factors (sex, reported race, household income, mother's education and father's education), cognitive and developmental factors (reported IQ score, reported intellectual disability, age at walking independently, age at first words, language regression and other regression) and autism severity (scores on the Social Communication Questionnaire and Repetitive Behaviour Scale-Revised). After excluding individuals with missing data, we quantified the variance explained by these factors using relative importance analysis (see method in ref. 52) for 5,773 autistic individuals diagnosed before the age of 22. Thereafter, analyses were done using the relaimpo (v.2.2-7) package in R, which allowed us to examine the contributions of all variables simultaneously[53].

### Trajectory analyses of birth cohorts

**Cohorts.** We used four population-based birth cohorts that varied both in the ages at which the data were collected from participants and the calendar years of the data collection (Extended Data Fig. 4). In brief, the four cohorts included were the UK-based MCS[54], the Australia-based LSAC-B and LSAC-K cohorts[55,56] and the Ireland-based Growing Up in Ireland (GUI) Child cohort (aka Cohort 98)[55,56]. All children included in the cohorts were born in the twenty-first century. Further details about the cohorts are provided in Supplementary Note 1. We used data from MCS, LSAC-B and LSAC-K for the growth mixture models. We did not use data from GUI for growth mixture models because there were only three time points, which is not enough to identify two or more trajectories. All four cohorts were used for latent growth curve models.

As indicated in Supplementary Table 2, these cohorts were selected because they were longitudinal in nature, were nationally representative and included key data on behavioural profiles and neurodevelopmental diagnosis. These overlapping features across datasets allowed for cross-country comparisons and generalization[57].

### Measures

**Autism and ADHD diagnosis and age at diagnosis.** In all cohorts, across multiple sweeps, the main carer was asked whether the participant had a diagnosis of autism (Extended Data Fig. 4). For age at diagnosis, we used the age at the sweep when carers first reported their child's autism diagnosis in every cohort, to maximize sample sizes and ensure consistency across cohorts for effective comparisons. For instance, if an autism diagnosis was reported for the participant for the first time at the age 11 sweep, we considered the age at diagnosis to be 11 years. Although the specific age at diagnosis was provided for LSAC-B and LSAC-K, we opted not to use this, because we identified errors in some reports in which months and years of diagnosis were swapped or not reported.

For MCS, we used reports of both autism and ADHD diagnoses for further sensitivity analyses. For our primary analyses, we included a narrowly defined sample of children with consistently reported autism diagnoses by primary and proxy carers (when both were available) and no other reported neurodevelopmental diagnosis (particularly ADHD). To assess the generalizability of our results and increase the sample size, we then expanded the sample to include all children with any reported diagnosis of autism (results are reported in Supplementary Note 3). This expanded sample included cases regardless of whether the diagnoses were consistent across sweeps or carers, and included those with co-occurring ADHD. Furthermore, we imputed the independent variables and covariates for autistic individuals with missing information, as detailed below (more details are in Supplementary Note 2 and Supplementary Note 3). Finally, to assess the specificity of the trajectories for autism, we conducted analyses among children who had a consistent ADHD diagnosis but no diagnosis of autism. Imputation was also performed within this ADHD-only sample, to increase the sample size.

**SDQ.** We used the SDQ to obtain social, emotional and behavioural profiles of participants, with repeated measures from 3 years to 18 years across cohorts. SDQ comprises 25 statements that carers were asked to rate on a three-point Likert scale ('not true', 'somewhat true' and 'certainly true') based on the child's symptoms or behaviours over the past six months. There were five subscales, each containing five items, which assessed emotional symptoms, conduct problems, hyperactivity–inattention, peer relationship problems and prosocial behaviours[23]. The first four subscales assessed difficulties, and their combined total score ranged from 0 to 40, with higher scores indicating greater difficulties. The fifth subscale (prosocial behaviours) represented strengths and ranged from 0 to 10, with higher scores indicating more prosocial behaviours. We analysed the total score and each subscale separately. The SDQ demonstrates good test–retest reliability and criterion validity across countries[24–26]. Its five-factor structure (each subscale as a factor) has shown consistency and invariance across age, sex and ethnic background[24,28]. The SDQ captures several core features of mental health and neurodevelopmental conditions, including autism and ADHD[58]. Only children with complete data of SDQ across all sweeps were included in the analyses, except for the imputation analyses.

**Sociodemographic measures.** Sociodemographic measures were included as covariates to account for their impact on age at diagnosis in each cohort (Supplementary Table 25). Specific measures and available information vary across cohorts, but we generally included sex, ethnic background, maternal age at delivery, child's cognitive aptitude, household SES and deprivation level of the living area, to account for factors that may affect the age when someone received an autism diagnosis[59,60]. Only subsets of children in the complete-SDQ samples, with complete data for these sociodemographic factors, were included in the analyses.

For MCS, although various census classifications for ethnic groups were available, we opted to use a binary indicator to identify non-white ethnic minorities. Ethnicity data were not collected in either LSAC cohort. Instead, visible ethnic minority status was determined mainly by parental country of birth and the language(s) spoken at home[61]. Maternal age at delivery was collected only for MCS. For other cohorts, we used maternal age (in years) at the first sweep of data collection to reflect the variation in maternal age at delivery.

For MCS, we identified multiple variables linked to cognitive ability, SES and area deprivation. Similarly, for LSAC-B, we identified multiple measures linked to SES, although there were no measures with sufficient sample size linked to cognitive ability or area deprivation. Subsequently, we conducted principal component analysis (PCA) for cognitive abilities, SES and area deprivation in MCS, and for SES in LSAC-B. PCA was done using a wide range of measures collected across sweeps (Supplementary Table 25), with one variable excluded from any pair with a correlation coefficient greater than 0.70 to address multicollinearity. The first principal component (PC1) explained more than 40% of the variance for each corresponding factor. By contrast, subsequent components contributed substantially less, supporting the use of the respective PC1 as the summary measure for cognitive ability, SES and area deprivation (Supplementary Table 25).

Intellectual disability was defined as scoring two standard deviations below the mean on the PC1 of the cognitive aptitude factor, consistent with previous studies. No autistic children in the MCS or LSAC cohorts who had measures of cognitive aptitude met the criteria for intellectual disability, probably because of participation bias. All PCA analyses were done in R using the prcomp() function[62].

## Statistical analyses

**Growth mixture models and latent growth curve models.** We used two methods to model the longitudinal trajectories of SDQ total and subscale scores. First, we used growth mixture models to identify whether there were latent groups of autistic individuals, based on their trajectories of SDQ total and subscale scores. Growth mixture models assume that the sample consists of multiple mixed effects models, each capturing a subgroup trajectory with shared intercept and slope[63]. We fitted models with one to four groups for each subscale and SDQ total scores in each cohort, using the lcmm (v.2.1.0) package in R[64]. The optimal number of latent trajectories were then determined by comparing fit indices, including Bayesian information criterion values, classification quality measure (entropy) and substantive interpretation. Models with lower Bayesian information criterion values and higher entropy were favoured[65]. Models identifying subgroups with less than 5% of the sample size were not considered, owing to poor statistical reliability and limited practical significance[66]. We compared the distribution of group memberships across diagnostic ages and across sexes using $\chi^2$ tests.

Second, we used linear latent growth curve models to identify the latent trajectories of SDQ total and subscale scores in the three groups (childhood diagnosed, adolescent diagnosed and the general population) for all cohorts. Each linear model included a latent intercept to represent the initial level of the outcome variable, and a linear latent slope to represent the mean rate of change over time. Childhood-diagnosed (diagnosed before ages 9–11, depending on the cohort) and adolescent-diagnosed (diagnosed after the ages of 9–11, depending on the cohort) were defined in advance. We chose this 9–11 age window as our cut-off because it aligns with the onset of puberty and the transition from primary to secondary school, and aligns with epidemiological evidence showing increased autism incidence in female individuals during this window[34]. An earlier cut-off was not feasible because only MCS (ages 5–7) and GUI (age 7) recorded autism diagnoses before this window. A later cut-off was not possible because there were no autism diagnoses in MCS after age 14. To further examine the relationship between age at diagnosis and socioemotional and behavioural outcomes, we conducted further latent growth curve models for autistic children using stepwise groupings by age at diagnosis in MCS and LSAC-B (see Supplementary Note 5 and Supplementary Fig. 10). All individuals who lacked an autism diagnosis (and also an ADHD diagnosis in MCS) were included in the general-population group.

Given the sex differences in age at autism diagnosis[34], we also applied the same models stratified by sex, estimating latent intercept and slope for each sex, within the autistic samples. All latent growth curve models were fitted under the structural equation modelling framework using the lavaan (v.0.6-19) package in R[67].

**Association with sex and mental-health phenotypes.** To examine the association between growth mixture models-derived SDQ latent trajectories and mental-health phenotypes in MCS, LSAC-B and LSAC-K, adjusting for sex, we used multiple regression in autistic individuals.

**Variance explained in age at autism diagnosis and mediation analysis.** Multiple regression analyses were conducted in MCS, LSAC-B and LSAC-K to investigate the association between age at autism diagnosis (the outcome variable), SDQ total difficulties and subscale latent trajectories memberships identified in optimal growth mixture models, and also accounting for other sociodemographic covariates. We did not detect any multicollinearity among the variables using variance inflation factors.

The relative importance of each predictor was assessed using dominance analysis[68]. We used the misty[69] (v.0.6.8) package in R for this analysis, using a correlation matrix extracted from the fitted model via the lavInspect function from the lavaan (v.0.6-19) package[67]. This approach leverages the correlation matrix to consider not only individual predictors, but also the correlations between them, providing a more comprehensive assessment of their relative importance[70].

To examine potential causal pathways, mediation analyses were done, allowing sociodemographic factors to indirectly influence the age at diagnosis through their effects on SDQ latent trajectory memberships identified in the optimal growth mixture model. Using structural equation modelling in the lavaan (v.0.6-19) package[67], both direct and indirect effects were assessed, with their significance calculated using bootstrapping analysis. Further details are provided in Supplementary Note 4.

To investigate the specificity of our findings to autism, we used growth mixture models, latent growth curve models, regression and mediation analyses in individuals with ADHD, but without a co-occurring autism diagnosis in the MCS cohort ($n = 89$, Supplementary Table 6, with results presented in Supplementary Note 5). ADHD diagnoses were available in the same sweeps as autism diagnoses, reported at ages 5, 7, 11 and 14. Carers were asked the following question: "Has a doctor or other health professional ever told you that <child's name> had attention deficit hyperactivity disorder (ADHD)?'.

**Imputation.** To assess the impact of missingness, we used softImpute (v.1.4-1) to impute missing data for all children with an autism or ADHD diagnosis reported by any carer in any sweep in the MCS cohort (autism: $n = 623$, Supplementary Table 5; ADHD: $n = 325$, Supplementary Table 6). We chose softImpute because of its computational efficiency in handling large-scale matrices through low-rank approximation, effectively preserving underlying structure of input data. Further information is provided in Supplementary Note 2.

## SPARK: genotyping, quality control and imputation

We used data from the SPARK cohort[39] iWES2 v.1 dataset (released in Feb 2022), which included data from 70,487 autistic individuals and their families as the SPARK Discovery cohort. Data from SPARK iWES v.3 (released in August 2024), which included an additional 71,267 autistic individuals and their families, was included in the SPARK Replication cohort. All participants in the Discovery cohort were genotyped using the Illumina Global Screening Array (GSA_24v2-0_A2) and in the Replication cohort using the Twist Bioscience genome-wide SNP capture panel for genotyping by sequencing. To avoid false positives caused by fine-scale population stratification, we restricted the analyses to individuals of genetically inferred European ancestries (Discovery, $n = 51,869$; Replication, $n = 50,211$ autistic and non-autistic participants), which was provided by the SPARK consortium. From this, we excluded individuals with genotyping rate of less than 98%, individuals

with sex mismatches and those with excess heterozygosity (3 standard deviations from the mean heterozygosity). Where trio data were available, trios with greater than 5% Mendelian errors were excluded, resulting in 47,170 (Discovery) and 48,750 (Replication) autistic and non-autistic individuals. We included genetic variants with minor allele frequency (MAF) greater than 1%, genotyping rate more than 95% and that were in Hardy–Weinberg equilibrium ($P > 1 \times 10^{-6}$), resulting in 518,189 (Discovery) and 1,225,308 (Replication) SNPs.

We used these quality-controlled genotype data for imputation, calculating genetic principal components and inferring relatedness among individuals. We inferred genetic relatedness using KING[71] (v.2.3.2). For genetic PCA, we pruned SNPs for linkage disequilibrium (maximum $r^2 = 0.1$) and removed the human leukocyte antigen region. Using PC-AiR[72] in GENESIS (v.2.22.2), we first calculated PCs in genetically unrelated individuals and then projected the PCs onto related individuals. We imputed genotypes using the TOPMED imputation panel[73] on the Michigan imputation server (v.1.7.3)[74] using Minimac4 (ref. 74) and after phasing using Eagle v.2.5 (ref. 75). After imputation, variants were converted from GRCh38/hg38 to GRCh37/hg19 using liftOver. We restricted downstream analyses to variants with MAF > 0.1% and with an imputation $R^2 > 0.6$.

## SPARK: association analyses

**Polygenic score association analyses.** Polygenic scores (PGSs) were calculated using PRScs[76] (v.1.1.0), which has a Bayesian shrinkage prior. PGSs were calculated for autism diagnosed before age 11 (iPSYCH$_{before11}$) and autism diagnosed after age 10 (iPSYCH$_{after10}$), generated using the iPSYCH2015 (ref. 77) cohort, details of which are provided below. For simplicity, we refer to this cohort as iPSYCH throughout. PGSs were calculated and analysed separately for the Discovery and Replication cohorts and meta-analysed using inverse-variance-weighted meta-analysis[78].

We ran separate linear regression analyses between each of the two PGSs and age at autism diagnosis (converted to years in all analyses) in the quality-controlled dataset. We excluded individuals older than 22 to focus on those who had an autism diagnosis using either the DSM-IV[79] or DSM-5, retaining a maximum of 18,809 (Discovery cohort) and 9,383 (Replication cohort) autistic individuals for PGS analyses. This criteria also allowed us to focus on individuals who received their diagnosis in childhood or adolescence, because older adults may have missed an earlier diagnosis of autism owing to secular changes in societal attitudes towards autism. The baseline model included intellectual disability (16.34% had carer-reported intellectual disability), sex and the first 10 genetic PCs as covariates. We ran eight different sensitivity analyses by including various covariates as well as the covariates included in the baseline model. First, we ran three models to account for developmental and clinical covariates: age at walking and age at first words (model 2), age at walking and first words, autism severity (SCQ and RBS-R total scores), carer-reported IQ scores, and language or other regression (model 3), and stratified analyses restricted to individuals without intellectual disability who can speak in longer sentences (model 4). Next, we ran two models accounting for sociodemographic factors: parental SES (model 5) and also area deprivation (model 6). We controlled for any attentional and behavioural diagnosis (model 7), for DSM edition (DSM IV versus DSM 5; model 8) and trio status (model 9). Finally, we also ran sensitivity analyses after stratifying by sex.

In the SPARK cohort, we obtained data for age at achieving nine developmental milestones (in months) for autistic individuals. For all milestones, we excluded individuals who were more than five median absolute deviations from the median. We ran multiple linear regression with PGS for iPSYCH$_{before11}$ and iPSYCH$_{after10}$ GWASs with sex and the first ten genetic PCs as covariates. Yet again, we ran the analyses for both the Discovery and Replication cohorts and meta-analysed it using inverse-variance-weighted meta-analysis.

**Rare high-impact de novo variants and inherited variants.** We identified rare (MAF < 0.1%) de novo and inherited variants in complete trios from SPARK, as previously described[40]. We identified high-impact protein-truncating variants by restricting it to variants in loss-of-function observed/expected upper bound fraction (LOEUF)[80]; highly constrained decile (LOEUF < 0.37) that were annotated as either frameshift, stop gained or start lost; and that had a loss-of-function transcript effect estimator (LOFTEE) high-confidence annotation. To identify high-impact de novo missense variants, we restricted it to variants in LOEUF highly constrained genes (LOEUF < 0.37) that had an MPC (missense badness, PolyPhen-2, and constraint) score[81] > 2.

We ran regression analyses for: high-impact de novo and inherited protein-truncating variants; missense variants; and by combining both protein-truncating and missense variants. We included sex and the first ten genetic PCs as covariates.

## GWAS

**GWAS of age at autism diagnosis.** We generated a GWAS of age at autism diagnosis (in years) in the quality-controlled dataset from SPARK, restricting it to autistic individuals who were under 22 years of age (Discovery, $n = 18,809$; Replication, $n = 9,356$) and SNPs with MAF > 1%. GWAS was generated using FastGWA[82] with sex, intellectual disability and the first ten genetic PCs as covariates. We meta-analysed the GWAS from the SPARK Discovery and Replication cohorts using inverse-variance-weighted meta-analysis in Plink[83,84] (v.2.0). In iPSYCH, we generated an additional GWAS of age at autism diagnosis (in years) in a quality-controlled dataset of unrelated individuals with sex and intellectual disability included as covariates using FastGWA[82], restricting it to SNPs with MAF > 1%. To keep it consistent with SPARK, we excluded individuals who were diagnosed after age 22, leaving a total sample of 18,965 individuals. In brief, pre-imputation quality control of the iPSYCH data was done using the Ricopili pipeline[85], prephased using Eagle v.2.3.5 and imputed using Minimac3 (ref. 86), using the downloadable version of the Haplotype Reference Consortium[87] (accession number EGAD00001002729). Further details of quality control and imputation are provided in ref. 88.

**GWAS of autism stratified by age at diagnosis.** We generated three age at autism diagnosis stratified GWASs in SPARK using (unscreened) non-autistic parents and siblings as controls (Discovery, $n_{control} = 24,965$; Replication, $n_{control} = 33,302$). The three GWAS were: first, SPARK, diagnosed before age 6 (SPARK$_{before6}$; Discovery, $n_{autistic} = 14,578$; Replication, $n_{autistic} = 6,857$); second, SPARK, diagnosed before age 11 (SPARK$_{before11}$, $n_{autistic} = 18,719$; Replication, $n_{autistic} = 9,162$); and third, SPARK, diagnosed after age 10 (SPARK$_{after10}$, $n_{autistic} = 3,358$; Replication, $n_{autistic} = 2,885$). For these analyses, we did not restrict it to individuals under 22, to increase the sample size. Of note, SPARK$_{before11}$ overlaps with the SPARK$_{before6}$ cohort. GWASs were generated using quality-controlled SNPs with MAF > 1% using FastGWA-GLMM[89]. We included age at recruitment in the study (to account for the use of parents as controls, who potentially lack an autism diagnosis owing to secular changes in attitudes and diagnosis), sex and the first ten genetic PCs as covariates. Fast-GWA GLMM can account for relatedness and fine-scale population stratification, even in family-based samples such as SPARK. Given the relatively low sample size of the Replication cohort, we did the meta-analysis of the Replication and Discovery cohort GWAS using inverse-variance-weighted meta-analyses in Plink[83,84] (v.2.0).

Although inclusion of unscreened related individuals as controls can decrease heritability and statistical power to identify loci[90], we used the GWAS to primarily conduct genetic correlation and related analyses. To ensure the robustness of these models we did the following: first, we confirmed that the attenuation ratio for all GWASs was not significantly greater than 1; second, we generated an additional GWAS of SPARK without stratifying by age at autism diagnosis using the same methods and confirmed a high genetic correlation ($r_g = 0.92$, s.e. = 0.17) with a

previous SPARK GWAS[49], which used a case-pseudocontrol approach; and third, in the genomicSEM analyses, we ran sensitivity analyses using a trio-based SPARK GWAS[49] in lieu of the age at diagnosis stratified GWAS from SPARK and confirmed our findings.

We generated three GWASs of autism stratified by age at autism diagnosis in the iPSYCH cohort[77]. The primary GWASs used in the analyses were GWASs of autism diagnosed before age 11 ($iPSYCH_{before11}$, $n = 9,500$ autistic and $n = 36,667$ non-autistic individuals) and autism diagnosed after age 10 ($iPSYCH_{after10}$, $n = 9,231$ autistic and $n = 36,667$ non-autistic individuals). We chose to subdivide the iPSYCH cohort at age 10 because we observed an increase in SDQ scores in birth cohorts at this age, and because age 10 is associated with an increase in diagnosis of female individuals in epidemiological samples[34]. We also conducted a GWAS of autism diagnosed before age nine ($iPSYCH_{before9}$, $n = 5,451$ autistic and $n = 36,667$ non-autistic individuals).

All individuals included in these GWASs from iPSYCH were born between May 1980 and December 2008 to mothers who were living in Denmark. The GWAS was done for unrelated individuals of European ancestry, with the first ten genetic PCs included as covariates using logistic regression as provided in PLINK.

### Heritability, genetic correlation, and genomicSEM

Heritability analyses for age at autism diagnosis were conducted using a single-component genome-wide complex trait analysis with a genomic-relatedness-based restricted maximum likelihood approach (GCTA-GREML v1.94.1)[91,92] in unrelated autistic individuals using the quality-controlled genetic data in SPARK. We estimated SNP-based heritability first after including sex and the first ten genetic PCs as covariates, and then also with intellectual disability as a covariate (baseline model). We ran several sensitivity analyses after including additional covariates: first, developmental milestones (age at first words and age at walking); second, developmental milestones and developmental regression (language regression and other regression); third, developmental milestones, developmental regression and IQ scores; fourth, SCQ and RBS-R scores; fifth, SCQ and RBS-R scores, developmental milestones and developmental regression; sixth, developmental milestones and SES; and finally, developmental milestones, SES and deprivation.

We conducted genetic correlation analyses using LDSC, with linkage disequilibrium scores from the northwest European populations.

We did genetic correlation analyses among different autism GWASs using LDSC (v.1.0.1). This included a European-only case-pseudocontrol GWAS in SPARK[49] (4,535 case-pseudocontrol pairs); GWAS from FinnGen (Data Release r10)[93] (646 cases and 301,879 controls), the PGC-2017 autism GWAS[45] (7,387 cases and 8,567 controls), GWAS from iPSYCH, and age at diagnosis-stratified GWAS from SPARK. The iPSYCH GWAS included an unstratified (19,870 autistic individuals, comprising 15,025 male individuals and 4,845 female individuals, and 39,078 controls) and sex-stratified GWAS[50], and three age at diagnosis-stratified GWASs, as mentioned earlier.

For genomicSEM[42] (v.0.0.5) analyses, we restricted it to six GWASs with minimal sample overlap, without high genetic correlation ($r_g > 0.95$), and with wide variation in age at diagnosis to conduct genomicSEM analyses using autosomes. Using the patterns of genetic correlations observed, we tested an age at diagnosis-related correlated two-factor model. We also tested: first, a single-factor model; second, a correlated two-factor 'geography' model, in which three US-based autism GWASs loaded onto one factor, and three Europe-based autism GWASs loaded onto a second factor; third, a bifactor model based on age at diagnosis; fourth, a bifactor model based on the geography of the cohorts; and finally, a hierarchical factor model based on age at diagnosis. The two-factor model was chosen because it had lower root mean square error of approximation and higher comparative fit index and was more parsimonious than the bifactor model. We ran sensitivity analyses using different GWASs of autism as input and confirmed

that the two-correlated-factor model was the best-fitting model of those tested.

### Analyses in ALSPAC and MCS

**Genetic quality control for ALSPAC.** We obtained quality-controlled and imputed genotype data from ALSPAC[94–96]. Further details about the cohort are provided in Supplementary Note 1. In brief, ALSPAC children were genotyped using the Illumina HumanHap550 quad chip genotyping platforms by 23andme. Some individuals were excluded owing to sex mismatches, excess heterozygosity, missingness greater than 3% and insufficient sample replication (identical by descent score of less than 0.8). After multidimensional scaling and comparison with Hapmap II (release 22), only individuals of genetically inferred European ancestries were retained. SNPs with low frequency (MAF < 1%), poor genotyping (call rate < 95%) and deviations from Hardy–Weinberg equilibrium ($P < 5 \times 10^{-7}$) were removed; 9,115 subjects and 500,527 SNPs passed quality control. Genotypes were phased using ShapeIT and imputation was done using the Haplotype Reference Consortium panel using the Michigan imputation server. After imputation, we further removed low-frequency SNPs (MAF < 1%). Further details of the quality control and imputation of ALSPAC are provided here: https://proposals. epi.bristol.ac.uk/alspac_omics_data_catalogue.html#org89bb79b. Genome-wide genotype data were generated by the Sample Logistics and Genotyping facilities at the Wellcome Sanger Institute and LabCorp (Laboratory Corporation of America) using support from 23andMe.

**Genetic quality control for MCS.** We also obtained quality-controlled and imputed data from MCS. In brief, MCS samples were genotyped using the Illumina Global Screening Array[97]. Some individuals were excluded owing to sex mismatches, excess heterozygosity and missingness greater than 2%. We identified European samples using the GenoPred pipeline[98] (https://github.com/opain/GenoPred). SNPs with low frequency (MAF < 1%), poor genotyping (call rate < 97%) and deviations from Hardy–Weinberg equilibrium ($P < 1 \times 10^{-6}$) were removed. Imputation was conducted using Minimac4 (ref. 74) using the TOPMED reference panel[73] in the Michigan imputation server[74]. After imputation, SNPs with an imputation $R^2$ INFO score of less than 0.8, with more than 3% missing and with MAF < 1% were excluded. Further details are available here: https://cls-genetics.github.io/docs/MCS.html.

**PGS association with SDQ.** PGSs for both ALSPAC and MCS were calculated for individuals of genetically inferred European ancestries. Genetic PCs were calculated for both cohorts using PC-AiR, as described earlier. We calculated PGS for $iPSYCH_{before11}$ and $iPSYCH_{after10}$ and used these in all analyses in the MCS to keep it consistent with analyses in SPARK.

We obtained scores on the SDQ total and subscales for six ages in the MCS and five ages in ALSPAC. We ran cross-sectional analysis at each age using multiple linear regression with PGS for $iPSYCH_{before11}$ and $iPSYCH_{after10}$, with sex, age and the first ten genetic PCs as covariates. We also ran multiple linear mixed effects regression using the lme4 (v.1.1.27.1) package in R[99], fitting a PGS by age interaction term to investigate whether the effects of PGS on SDQ change over time.

To investigate whether the differences in association between MCS and ALSPAC were due to differences in ascertainment between the two cohorts, we matched ALSPAC to MCS using entropy balancing[100] and re-ran the PGS association analyses. Entropy balancing is a reweighting technique that ensures the covariate distributions are identical between groups. This method uses optimization algorithms to assign weights to individuals such that the weighted average of the covariates in ALSPAC (the larger genotyped cohort) matches that of MCS (the smaller genotyped cohort), minimizing confounding biases and increasing comparability. We used the child's biological sex, maternal age at delivery and maternal highest educational qualification at first data collection in each cohort as matching factors. Entropy balancing was done using the ebal (v.0.1-8) package in R[101].

**PGS association with communication skills and autism diagnosis.**
For ALSPAC, we obtained understanding of simple phrases (such as "Do you want that?" or "Come here") and gesture scores from the Macarthur-Bates Communicative Development Inventories[102] at 15 months of age (Supplementary Note 9). We conducted multiple linear regression using PGS for iPSYCH$_{before11}$ and iPSYCH$_{after10}$, with sex, age and the first ten genetic PCs as covariates.

Autism diagnosis for the MCS was obtained using parent/carer reports of autism/Asperger's syndrome diagnosis by a doctor at ages 5, 7, 11 and 14. We identified individuals with an autism diagnosis at age 7 or earlier, age 11 or earlier, or between ages 11 and 14. We conducted Firth's bias-reduced multiple logistic regression (logistf v.1.26.0 package in R) using PGS for iPSYCH$_{before11}$ and iPSYCH$_{after10}$, with sex, age and the first ten genetic PCs as covariates.

## Ethics

Ethical approval for individual cohorts was obtained independently of the current study. Ethical approval for ALSPAC was obtained from the ALSPAC Ethics and Law Committee and the local research ethics committees. Ethical approval for each sweep of MCS was obtained from NHS research ethics committees (MREC). Ethical approval for LSAC was obtained from the Australian Institute of Family Studies Human Research Ethics Committee. Ethical approval for GUI was obtained from a dedicated research ethics committee set up by the Department of Children, Equality, Disability, Integration and Youth. Ethical approval for SPARK was obtained from the Western Institutional Review Board Copernicus Group (IRB protocol 20151664). The Danish Scientific Ethics Committee, the Danish Health Data Authority, the Danish data protection agency and the Danish Neonatal Screening Biobank Steering Committee approved the iPSYCH study. Ethical approval for the analyses of de-identified data used in this study was obtained from the Cambridge Human Biology Research Ethics Committee (HBREC.2020.07).

## Reporting summary

Further information on research design is available in the Nature Portfolio Reporting Summary linked to this article.

## Data availability

- SPARK autism GWAS: https://bitbucket.org/steinlabunc/spark_asd_sumstats/src
- Finngen autism GWAS: https://www.finngen.fi/en/access_results
- The iPSYCH autism GWAS (unstratified, sex stratified and age at diagnosis stratified, age at diagnosis) can be obtained from J. Grove (grove@biomed.au.dk) or A.D.B. (anders@biomed.au.dk) upon reasonable request.
- Psychiatric GWAS summary stats: https://pgc.unc.edu/for-researchers/download-results/
- GWAS educational attainment: https://thessgac.com/
- GWAS cognitive aptitude: https://cncr.nl/research/summary_statistics/
- For ALSPAC, the study website contains details of all the available data through a fully searchable data dictionary and variable search tool: http://www.bristol.ac.uk/alspac/researchers/our-data/
- For MCS, data can be obtained on application from the UK Data Service: https://beta.ukdataservice.ac.uk/datacatalogue/series/series?id=2000031
- Summary statistics for the SPARK-based age at diagnosis GWAS, and the age at diagnosis-stratified GWAS generated from the genomicSEM models, are available here: https://doi.org/10.6084/m9.figshare.29566052.v2

## Code availability

- Lavaan (LGCM) v.0.6-19: https://lavaan.ugent.be/tutorial/growth.html
- lcmm(GMM), v.2.2.1: https://github.com/CecileProust-Lima/lcmm
- Softimpute v.1.4-1: https://cran.r-project.org/web/packages/softImpute/softImpute.pdf
- Misty v.0.6.8: https://cran.r-project.org/web/packages/misty/index.html
- PRScs v.1.1.0: https://github.com/getian107/PRScs
- fastGWA and GCTA v.1.94.1: https://yanglab.westlake.edu.cn/software/gcta/#Overview
- GenomicSEM v.0.0.5: https://github.com/GenomicSEM/GenomicSEM
- LDSC v.1.0.1: https://github.com/bulik/ldsc
- KING v.2.3.2: https://www.kingrelatedness.com/manual.shtml
- Plink 2.0: https://www.cog-genomics.org/plink/2.0/
- GENESIS v.2.22.2: https://github.com/UW-GAC/GENESIS
- Lme4 v.1.1.27.1: https://github.com/lme4/lme4/
- Logistf v.1.24: https://cran.r-project.org/web/packages/logistf/index.html
- Ebal v.0.1-8: https://cran.r-project.org/web/packages/ebal/ebal.pdf
- Relaimpo v.2.2-7: https://cran.r-project.org/web/packages/relaimpo/relaimpo.pdf
- SPARK quality control, imputation and GWAS: https://github.com/vwarrier/SPARK_iWES2_imputation/
- Bespoke genetic analyses code: https://github.com/vwarrier/autism_agediagnosis/

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

**Acknowledgements** This research was supported by funding from the Simons Foundation for Autism Research Initiative, the Wellcome Trust (214322\Z\18\Z and 226392/Z/22/Z), Horizon-Europe R2D2-MH (grant agreement 101057385) and UKRI (10063472). For the purpose of open access, we have applied a CC BY public copyright licence to any author-accepted manuscript version arising from this submission. S.B.-C. also received funding from the Autism Centre of Excellence at Cambridge, the Templeton World Charitable Fund, the MRC and the National Institute for Health Research Cambridge Biomedical Research Centre. The research was supported by the National Institute for Health Research Applied Research Collaboration East of England. Any views expressed are those of the author(s) and not necessarily those of the funder. Some of the results leading to this publication have received funding from the Innovative Medicines Initiative 2 Joint Undertaking under grant agreement 777394 for the project AIMS-2-TRIALS. This joint undertaking receives support from the European Union's Horizon 2020 research and innovation program and the EFPIA and Autism Speaks, Autistica and the SFARI. The iPSYCH team was supported by grants from the Lundbeck Foundation (R102-A9118, R155-2014-1724 and R248-2017-2003), the NIMH (1R01MH124851-01 to A.D.B.) and the European Union's Horizon Europe program (R2D2-MH; grant agreement 101057385 to A.D.B.). The Danish National Biobank resource was supported by the Novo Nordisk Foundation. High-performance computer capacity for handling and statistical analysis of iPSYCH data on the GenomeDK HPC facility was provided by the Center for Genomics and Personalized Medicine and the Centre for Integrative Sequencing, iSEQ, Aarhus University, Denmark (grant to A.D.B.). The UK Medical Research Council and Wellcome (grant 217065/Z/19/Z) and the University of Bristol provide core support for ALSPAC. This publication is the work of the authors and the authors will serve as guarantors for the contents of this paper. A comprehensive list of grants funding is available on the ALSPAC website (http://www.bristol.ac.uk/alspac/external/documents/grant-acknowledgements.pdf). R2D2-MH has been funded by Horizon Europe (grant agreement 101057385), by UK Research and Innovation (UKRI) under the UK government's Horizon Europe funding guarantee (grant 10039383) and by the Swiss State Secretariat for Education, Research and Innovation (SERI) under contract 22.00277. A.H. and L.H. were supported by the Norwegian Research Council (336085) and the Norwegian Health Authority (2020022, #2022029 and #2022083). E. Verhoef and B.S.P. are funded by the Max Planck Society. We thank the Centre for Longitudinal Studies (CLS), UCL Social Research Institute, for the use of these data and the UK Data Service for making them available. However, neither CLS nor the UK Data Service bear any responsibility for the analysis or interpretation of these data. This paper uses unit record data from Growing Up in Australia, the Longitudinal Study of Australian Children. The study is conducted in partnership between the Department of Social Services (DSS), the Australian Institute of Family Studies (AIFS) and the Australian Bureau of Statistics (ABS). The findings and views reported in this paper are those of the author and should not be attributed to DSS, AIFS or the ABS. Growing Up in Ireland (GUI) was funded by the Government of Ireland through the Department of Children, Equality, Disability, Integration and Youth (DCEDIY) and the Central Statistics Office (CSO). Results in this report are based on analysis of data from research microdata files provided by the Central Statistics Office (CSO). Neither the CSO nor the DCEDIY take any responsibility for the views expressed or the outputs generated from these analyses. We thank all the families who took part in this study, the midwives for their help in recruiting them and the ALSPAC team, which includes interviewers, computer and laboratory technicians, clerical workers, research scientists, volunteers, managers, receptionists and nurses. This study includes data from the Norwegian Mother, Father and Child Cohort Study, conducted by the Norwegian Institute of Public Health. We thank all the participating families, and A. Kwong, T. Ford, W. Mandy and A. Grotzinger for discussions.

**Author contributions** X.Z. and V.W. did most of the analyses, with the remainder being done by J. Grove, Y.G., C.K.B. and L.K.N. M.K., E. Verhoef, A. Gui, L.H., A.H., A.R., B.S.P. and A.D.B. provided summary statistics for various analyses. V.W. supervised the analyses and directed the study with input from H.C.M. and J. Grove V.W. and X.Z. wrote the initial draft with input from H.C.M. A. Gui, S.A.S.N., A.H., B.S.P., A.R., D.S.M., E.M.W., D.H.G., N.R.W., E.B.R., T.B. and S.B.-C. provided intellectual input. All authors read and commented on the final manuscript.

**Competing interests** A.D.B. received a speakers' fee from the Lundbeck Foundation. The other authors declare no competing interests.

**Additional information**
**Correspondence and requests for materials** should be addressed to Xinhe Zhang or Varun Warrier.

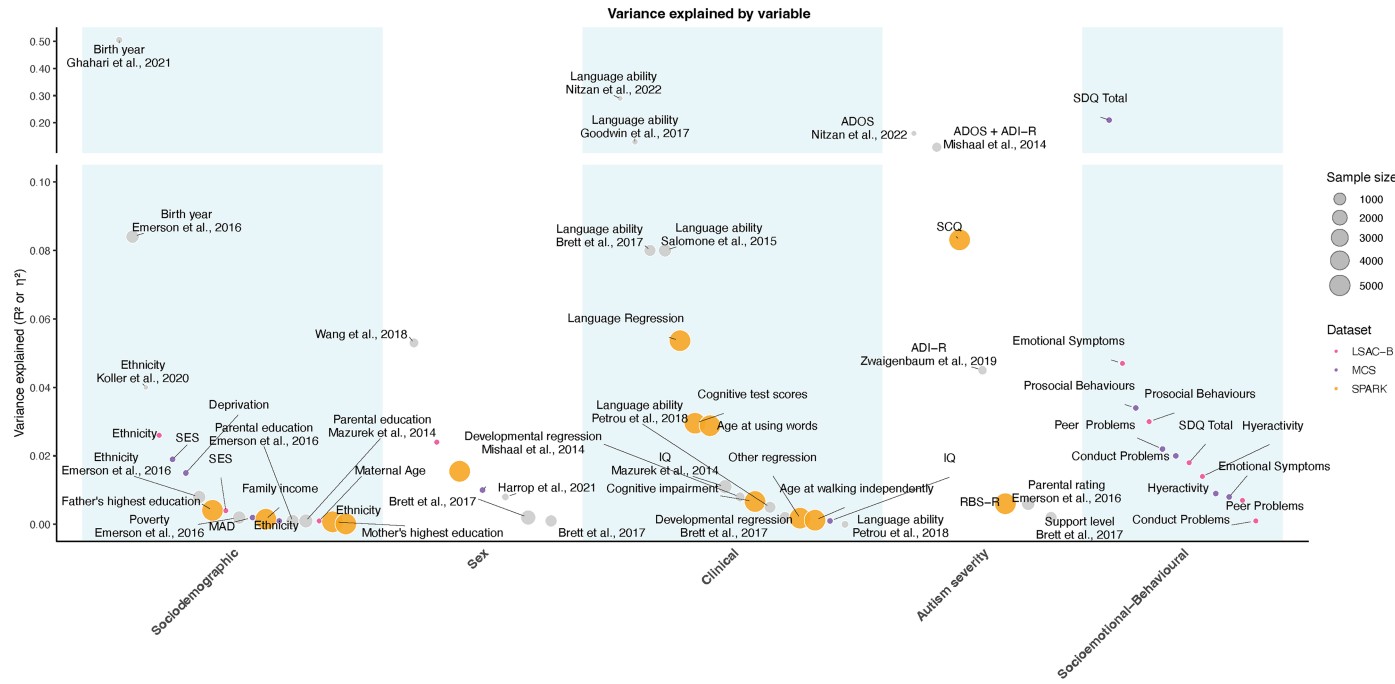

**Extended Data Fig. 1 | Variance in age at diagnosis explained by various clinical and sociodemographic factors.** Variance explained ($R^2$ or $\eta^2$) in age at autism diagnosis by clinical and sociodemographic factors, identified from the review of literature (1998–2023). Variables are grouped into sociodemographic (MAD, SES), socioemotional-behavioural (SDQ scores), sex, clinical (e.g., IQ, regression, language ability), and autism severity (e.g, SCQ, ADOS, ADI-R, RBS-R) categories. Circle size represents sample size, with larger circles indicating larger cohorts. Colored points denote variables analysed in the current study. Inset shows factors that explain greater than 10% of the variance in age at autism diagnosis. Note: None of these studies account for additional family, service access, and contextual factors known to influence diagnostic timing. Abbreviations: MAD, Maternal Age at Delivery; IQ, Intelligence Quotient; SES, Socio-economic Status; ADOS, Autism Diagnostic Observation Schedule; ADI-R, Autism Diagnostic Interview-Revised; RBS-R, Repetitive Behavior Scale-Revised; SCQ, Social Communication Questionnaire; SDQ, Strengths and Difficulties Questionnaire; SDQ Total, SDQ Total Difficulties.

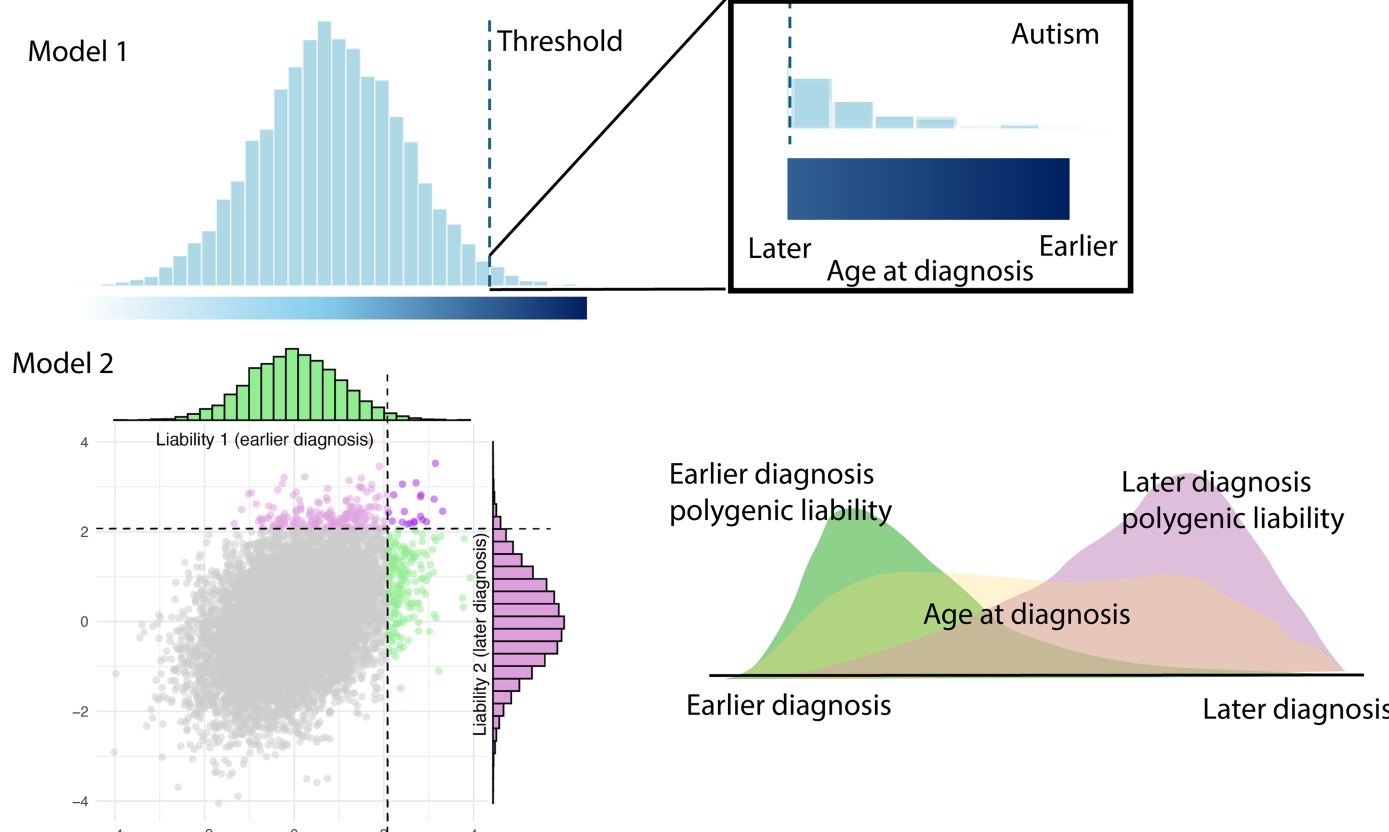

**Extended Data Fig. 2 | Two aetiological polygenic models of autism.** In Model 1 (Unitary Model), we assume a single liability threshold polygenic model. In this model, autism emerges from a unitary polygenic aetiology. Autistic individuals diagnosed later have lower polygenic predisposition than individuals diagnosed earlier. In Model 2 (Developmental Model), we model two correlated age-dependent polygenic liabilities.

**Aim 1**: Investigate the association between developmental trajectories of socio-emotional behaviours and age at autism diagnosis in birth cohorts.

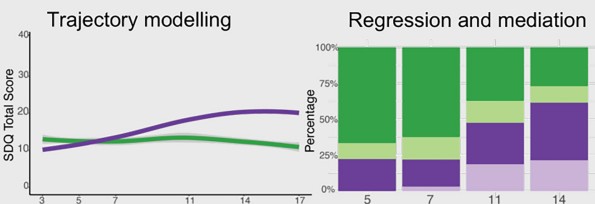

Trajectory modelling    Regression and mediation

**Key finding:** Developmental trajectories are associated with age at autism diagnosis.

**Aim 2**: Quantify the SNP-based heritability of age at autism diagnosis and test if clinical and demographic factors reduce this heritability.

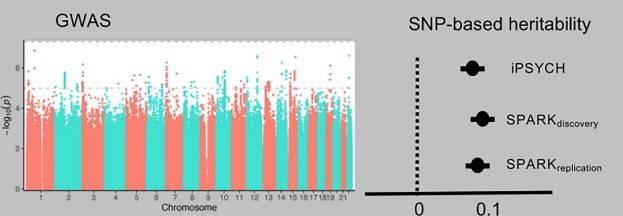

GWAS    SNP-based heritability

**Key finding:** Age at autism diagnosis is partly heritable and not explained by known clinical or demographic factors.

**Aim 3**: Investigate whether two or more polygenic factors influence age of autism diagnosis and if this partly explains genetic heterogeneity in autism.

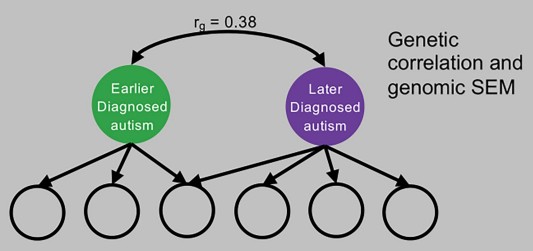

$r_g = 0.38$

Earlier Diagnosed autism    Later Diagnosed autism

Genetic correlation and genomic SEM

**Key finding:** Two underlying polygenic factors linked to differences in age at autism diagnosis.

**Aim 4:** Characterise the genetic relationship between the two autism polygenic traits and other mental health conditions, developmental phenotypes and trajectories.

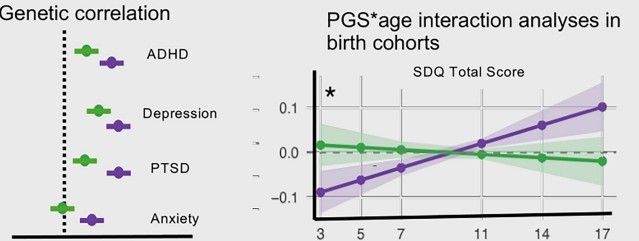

Genetic correlation    PGS*age interaction analyses in birth cohorts

**Key finding:** Two autism polygenic traits have differing genetic relationship between mental health conditions and developmental trajectories.

**Extended Data Fig. 3 | Schematic of the study aims.** The study consists of four linked aims to understand whether the developmental trajectories and polygenic etiology of autism differs by age at diagnosis. In Aim 1, we modelled socioemotional and behavioural trajectories among autistic individuals in birth cohorts and investigated their association with age at autism diagnosis. In Aim 2, estimated the SNP-based heritability of age at autism diagnosis and whether it attenuates when accounting for various clinical and demographic factors. In Aim 3, we investigated whether the varying patterns of genetic correlations observed among different GWAS of autism can be explained by different polygenic factors associated with age at diagnosis. In Aim 4, we investigated the genetic relationship between the two autism polygenic factors and mental health and developmental phenotypes.

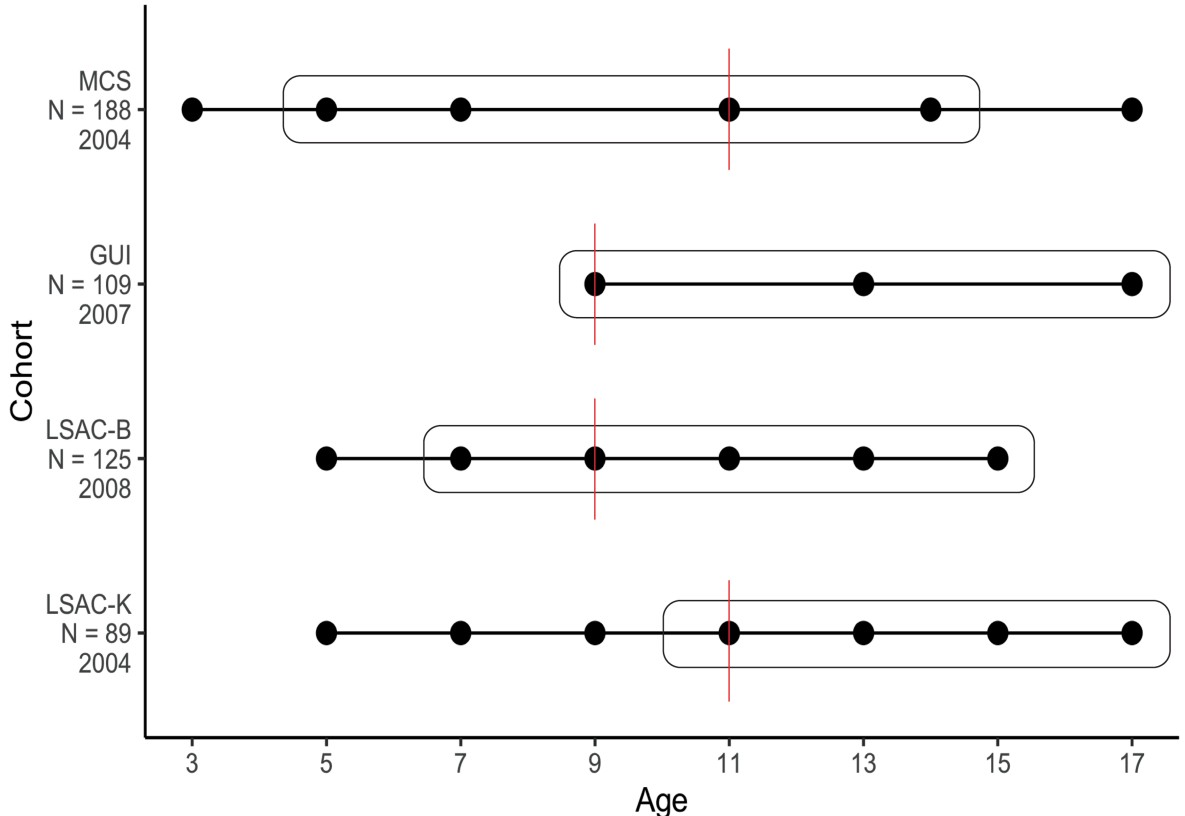

**Extended Data Fig. 4 | Schematic diagram of the birth cohorts included in the study.** Schematic diagram of the cohorts included in the study and the ages when data was collected for SDQ scores (dots) and autism diagnosis (in boxes). Reports of autism diagnosis were available at ages: MCS - 5,7,11,14; GUI - 9,13,17; LSAC-B - 7,9,11,13,15; and LSAC-K: 11,13,15,17. MCS = Millennium Cohort Study; GUI = Growing up in Ireland (cohort '98); LSAC-B = Longitudinal Study of Australian Children (Birth cohort); LSAC-K = Longitudinal Study of Australian Children (Kindergarten cohort). Sample sizes and the year of initial SDQ data collection for each cohort are shown on the ordinate axis. The age cutoff used in the Latent Growth Curve Models for each cohort is indicated by a red line. GUI was used only for Latent Growth Curve Models and excluded from Growth Mixture Models.

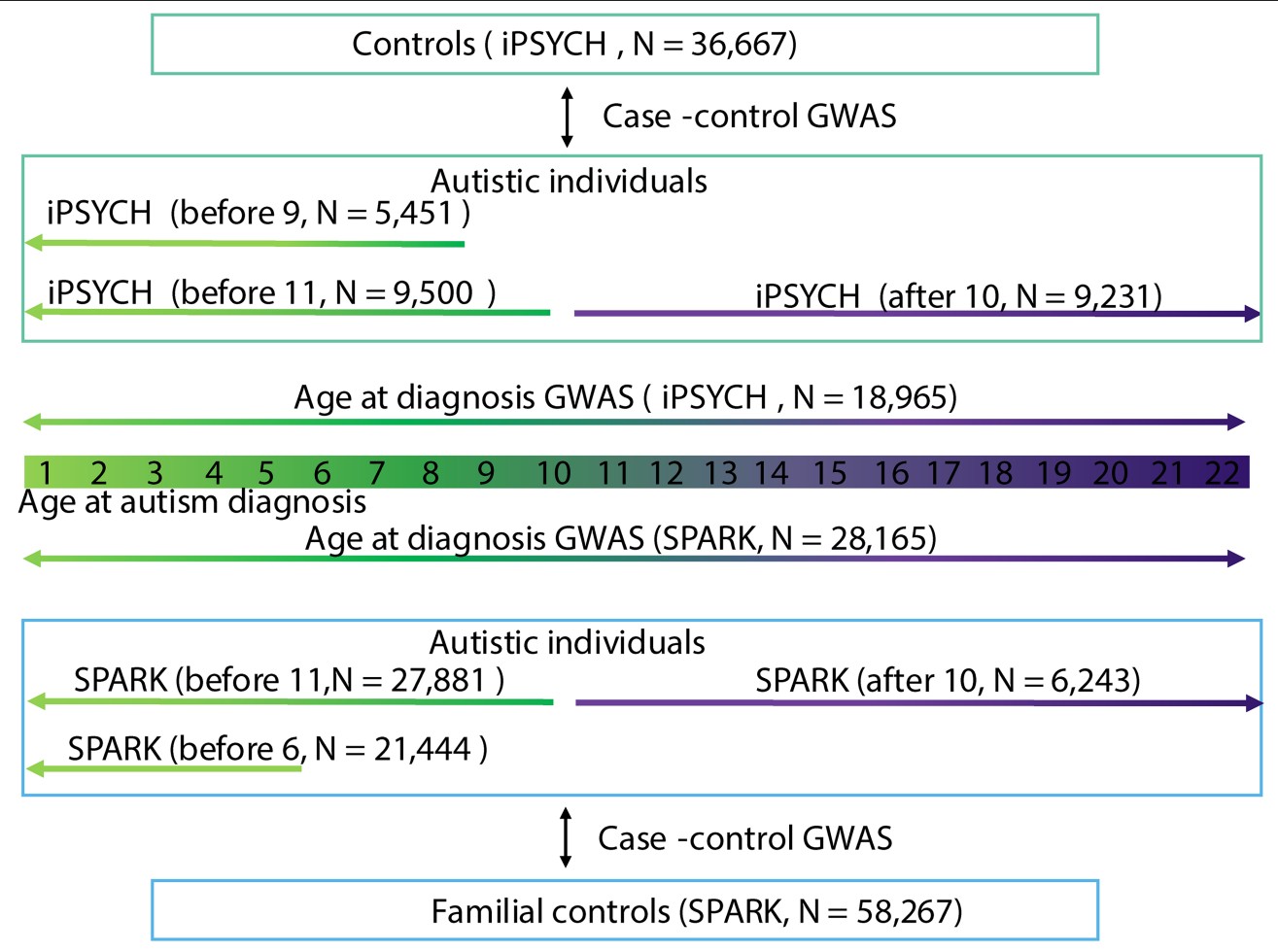

**Extended Data Fig. 5 | Schematic diagram of age at autism diagnosis GWAS and age stratified autism GWAS.** Schematic diagram illustrating the main GWAS conducted in the study using the SPARK and iPSYCH cohorts. We conducted two age at autism diagnosis GWAS. In addition, we conducted six case-control GWAS, where autistic individuals were stratified based on their age at autism diagnosis.

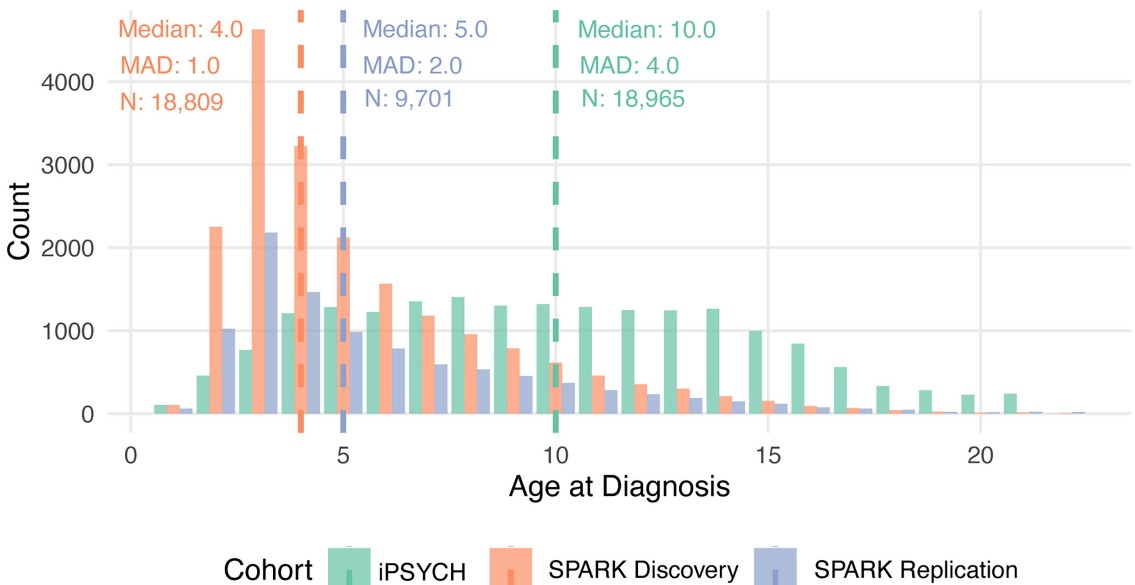

**Extended Data Fig. 6 | Distribution of age at autism diagnosis in iPSYCH and SPARK.** Frequency histograms of age at autism diagnosis in iPSYCH and SPARK. Median and median absolute deviation (MAD) for age at diagnosis, and sample sizes (N) have been provided.

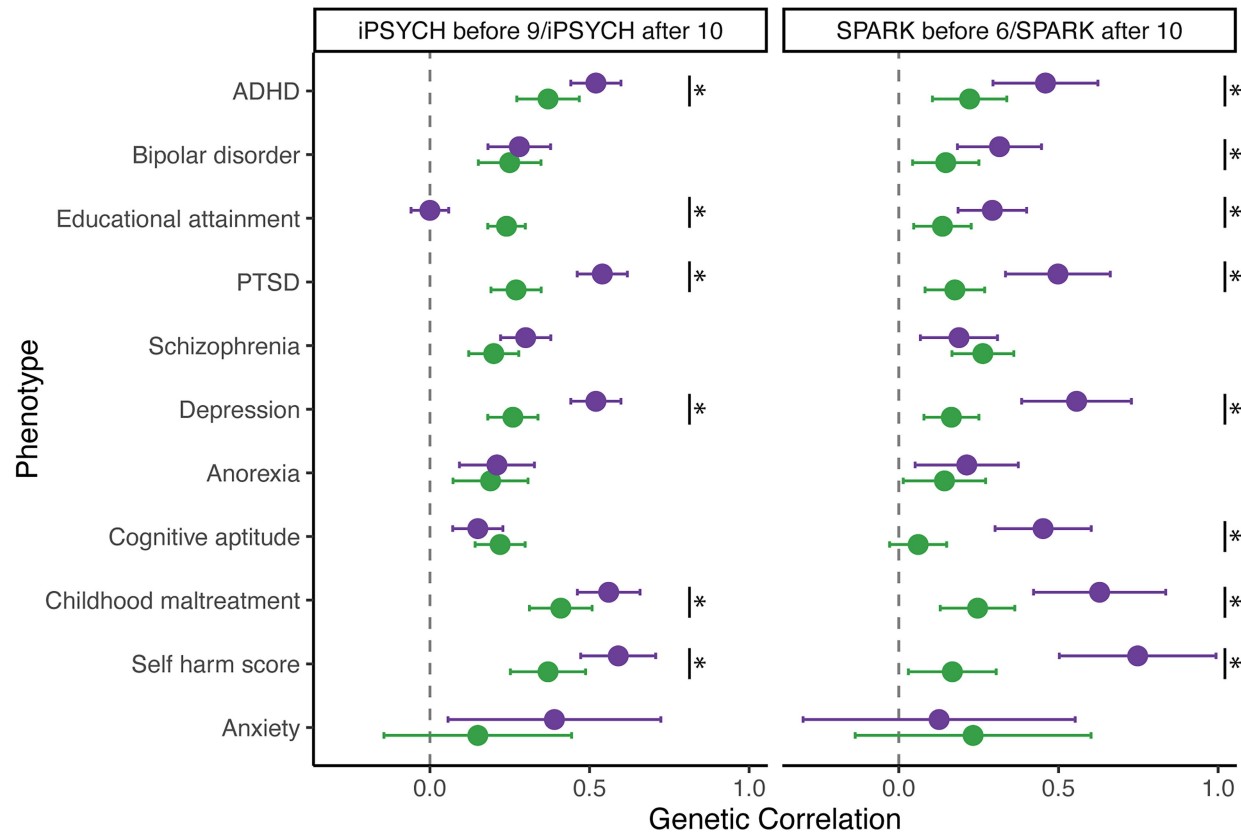

**Extended Data Fig. 7 | Within-cohort genetic correlation between age at diagnosis stratified autism GWAS and mental health and cognition related phenotypes.** Genetic correlation between age at autism stratified GWAS in SPARK (meta-analysed from Discovery and Replication cohorts) and iPSYCH and other mental health and cognition related phenotypes. Points represent genetic correlation estimates and whiskers indicate 95% confidence intervals. Green represents the earlier diagnosed autism GWAS (iPSYCH before 9 and SPARK before 6), and purple represents later diagnosed autism GWAS (iPSYCH and SPARK after 10). Asterisk (*) indicates significantly different genetic correlation between the earlier and later diagnosed GWAS (P < 0.05, two-tailed Z test).

# Reporting Summary

## Statistics

For all statistical analyses, confirm that the following items are present in the figure legend, table legend, main text, or Methods section.

| n/a | Confirmed | |
|---|---|---|
| ☐ | ☒ | The exact sample size (*n*) for each experimental group/condition, given as a discrete number and unit of measurement |
| ☐ | ☒ | A statement on whether measurements were taken from distinct samples or whether the same sample was measured repeatedly |
| ☐ | ☒ | The statistical test(s) used AND whether they are one- or two-sided<br>*Only common tests should be described solely by name; describe more complex techniques in the Methods section.* |
| ☐ | ☒ | A description of all covariates tested |
| ☐ | ☒ | A description of any assumptions or corrections, such as tests of normality and adjustment for multiple comparisons |
| ☐ | ☒ | A full description of the statistical parameters including central tendency (e.g. means) or other basic estimates (e.g. regression coefficient) AND variation (e.g. standard deviation) or associated estimates of uncertainty (e.g. confidence intervals) |
| ☐ | ☒ | For null hypothesis testing, the test statistic (e.g. *F*, *t*, *r*) with confidence intervals, effect sizes, degrees of freedom and *P* value noted<br>*Give P values as exact values whenever suitable.* |
| ☒ | ☐ | For Bayesian analysis, information on the choice of priors and Markov chain Monte Carlo settings |
| ☒ | ☐ | For hierarchical and complex designs, identification of the appropriate level for tests and full reporting of outcomes |
| ☐ | ☒ | Estimates of effect sizes (e.g. Cohen's *d*, Pearson's *r*), indicating how they were calculated |

*Our web collection on statistics for biologists contains articles on many of the points above.*

## Software and code

Policy information about availability of computer code

| Data collection | No separate software was used to collect data. |
|---|---|
| Data analysis | Lavaan (LGCM) (v.0.6-19) : https://lavaan.ugent.be/tutorial/growth.html<br>lcmm(GMM), v 2.2.1: https://github.com/CecileProust-Lima/lcmm<br>Softimpute (v.1.4-1): https://cran.r-project.org/web/packages/softImpute/softImpute.pdf<br>Misty (v 0.6.8) : https://cran.r-project.org/web/packages/misty/index.html<br>PRScs (v.1.1.0): https://github.com/getian107/PRScs<br>fastGWA and GCTA (v.1.94.1): https://yanglab.westlake.edu.cn/software/gcta/#Overview<br>GenomicSEM (v.0.0.5): https://github.com/GenomicSEM/GenomicSEM<br>LDSC (v1.0.1): https://github.com/bulik/ldsc<br>KING (v.2.3.2): https://www.kingrelatedness.com/manual.shtml<br>Plink 2.0: https://www.cog-genomics.org/plink/2.0/<br>GENESIS (v2.22.2): https://github.com/UW-GAC/GENESIS<br>Lme4 (v.1.1.27.1): https://github.com/lme4/lme4/<br>Logistf (v.1.24): https://cran.r-project.org/web/packages/logistf/index.html<br>Ebal (v 0.1-8): https://cran.r-project.org/web/packages/ebal/ebal.pdf<br>Relaimpo (v2.2-7): https://cran.r-project.org/web/packages/relaimpo/relaimpo.pdf<br><br>SPARK quality control, imputation and GWAS: https://github.com/vwarrier/SPARK_iWES2_imputation/<br>Bespoke genetic analyses code: https://github.com/vwarrier/autism_agediagnosis/ |

For manuscripts utilizing custom algorithms or software that are central to the research but not yet described in published literature, software must be made available to editors and reviewers. We strongly encourage code deposition in a community repository (e.g. GitHub). See the Nature Portfolio guidelines for submitting code & software for further information.

## Data

Policy information about availability of data

All manuscripts must include a data availability statement. This statement should provide the following information, where applicable:
- Accession codes, unique identifiers, or web links for publicly available datasets
- A description of any restrictions on data availability
- For clinical datasets or third party data, please ensure that the statement adheres to our policy

SPARK autism GWAS: https://bitbucket.org/steinlabunc/spark_asd_sumstats/src
Finngen autism GWAS: https://www.finngen.fi/en/access_results
iPSYCH autism GWAS (unstratified, sex-stratified and age at diagnosis stratified, age at diagnosis) can be obtained from Anders Borglum and Jakob Grove.
Psychiatric GWAS summary stats: https://pgc.unc.edu/for-researchers/download-results/
GWAS educational attainment: https://thessgac.com/papers/
GWAS cognitive aptitude: https://cncr.nl/research/summary_statistics/
For ALSPAC, the study website contains details of all the data that is available through a fully searchable data dictionary and variable search tool": http://www.bristol.ac.uk/alspac/researchers/our-data/
For MCS, data can be obtain after application through the UK Data Service: https://beta.ukdataservice.ac.uk/datacatalogue/series/series?id=2000031
Summary statistics for the SPARK-based age at diagnosis GWAS, and the age at diagnosis stratified GWAS generated from the genomicSEM models are available here: https://figshare.com/articles/dataset/Summary_statistics_for_Polygenic_and_developmental_profiles_of_autism_differ_by_age_at_diagnosis/29566052.

## Research involving human participants, their data, or biological material

Policy information about studies with human participants or human data. See also policy information about sex, gender (identity/presentation), and sexual orientation and race, ethnicity and racism.

| | |
|---|---|
| Reporting on sex and gender | We use the term sex throughout, which primarily refers to sex assigned at birth. |
| Reporting on race, ethnicity, or other socially relevant groupings | We restrict our genetic analyses to individuals of genetically inferred European ancestries. For trajectory modelling in birth cohorts, we do not exclude individuals based on informant-reported ancestry, and include informant-reported ancestry as a covariate in some models. |
| Population characteristics | Data from existing human population cohorts/databases were used, including from birth cohorts. The four birth cohorts are: Millennium Cohort Study (year of birth - 2000/2001), Growing up in Ireland (year of birth - 1998), Longitudinal Study of Australian Children - Kindergarten cohort (Year of birth - 1999 - 2000) and Birth cohort (Year of birth - 2003 - 2004). In these birth cohort, a subset of the children were autistic. MCS - 238, GUI - 109, LSAC-K - 89, LSAC-B - 129). For PGS analyses, we also included data from ALSPAC (Year of birth 1990 - 1991). <br><br>We included data from two cohorts of autistic individuals and their families. These were the US-based SPARK cohort, including a Discovery subset (N = 18,809, median age at autism diagnosis -= 4.0 years), and a Replication subset (N = 9,701, median age at autism diagnosis = 5.0 years), and the Danish based iPSYCH cohort (N = 18,965, median age at autism diagnosis = 4.0). <br><br>Detailed population characteristics of the birth cohorts, and the cohorts used for the GWAS analyses are provided in the relevant section of the Methods or Supplementary Information. |
| Recruitment | Data from existing human population cohorts/databases were used, including from birth cohorts. Details of recruitment and population characteristics are provided in the relevant section of the Methods or Supplementary Information. |
| Ethics oversight | Data for existing cohorts were collected based on ethical approval from local IRBs. Analyses of de-identified data from cohorts was approved by the Cambridge Human Biology Research Ethics Committee. |

Note that full information on the approval of the study protocol must also be provided in the manuscript.

# Field-specific reporting

Please select the one below that is the best fit for your research. If you are not sure, read the appropriate sections before making your selection.

☒ Life sciences  ☐ Behavioural & social sciences  ☐ Ecological, evolutionary & environmental sciences

For a reference copy of the document with all sections, see nature.com/documents/nr-reporting-summary-flat.pdf

# Life sciences study design

All studies must disclose on these points even when the disclosure is negative.

| | |
|---|---|
| Sample size | As the study relied on existing data, we used the maximum available sample after quality control. No a priori sample size calculation was conducted. |
| Data exclusions | Data were excluded after phenotypic or genetic quality control, and this varied by cohorts. Further details are provided in Methods. |
| Replication | Where possible, all analyses were conducted in at least two cohorts to assess the replicability and generalisability of the findings. Trajectory modelling was conducted in four birth cohorts. Genetic analyses of autistic individuals were conducted in two cohorts - SPARK and iPSYCH. Genetic analyses of the general population were conducted in two cohorts - MCS and ALSPAC, with additional support from MoBa. In addition, we included a replication cohort from SPARK, which we analysed only after the first version of the manuscript was submitted and reviewed. This was available only after the initial submission of the manuscript. Details are provided in the manuscript. All effects were in the consistent direction between the original Discovery and the Replication cohorts, and meta-analysis of the two datasets increased the statistical significance of the findings, indicating that replication was largely successful. |
| Randomization | No randomisation was conducted as this is an observational study. |
| Blinding | No blinding was conducted as this is an observational study. |

# Reporting for specific materials, systems and methods

We require information from authors about some types of materials, experimental systems and methods used in many studies. Here, indicate whether each material, system or method listed is relevant to your study. If you are not sure if a list item applies to your research, read the appropriate section before selecting a response.

## Materials & experimental systems

| n/a | Involved in the study |
|---|---|
| ☒ ☐ | Antibodies |
| ☒ ☐ | Eukaryotic cell lines |
| ☒ ☐ | Palaeontology and archaeology |
| ☒ ☐ | Animals and other organisms |
| ☒ ☐ | Clinical data |
| ☒ ☐ | Dual use research of concern |
| ☒ ☐ | Plants |

## Methods

| n/a | Involved in the study |
|---|---|
| ☒ ☐ | ChIP-seq |
| ☒ ☐ | Flow cytometry |
| ☒ ☐ | MRI-based neuroimaging |

## Plants

| | |
|---|---|
| Seed stocks | N/A |
| Novel plant genotypes | N/A |
| Authentication | N/A |

