## [Peer Review File · Nature]

Polygenic and developmental profiles of autism differ by age at diagnosis

Corresponding Author: Dr Varun Warriar

Version 0:

Reviewer comments:

Referee #1

(Remarks to the Author)

Summary of key results:

The authors utilized four birth cohorts from different areas of the world (UK, Ireland and Australia) to examine 'socio-behavioural trajectories'; that is, change over time in 'Strengths and Difficulties Questionnaire' scores, as assessed using Latent Growth Curve Models. They assessed whether trajectories differed based on whether a child had ever been diagnosed with autism, and whether there were differences in trajectories among autistic children, related to age of autism diagnosis, dichotomized at age 9 or 11, depending on cohort. Sensitivity analyses indicated that these trajectories were relatively specific to autism (vs ADHD) and did not reflect sex differences in age of diagnosis. SCQ trajectory membership was also associated with mental health outcomes; specifically, the 'late childhood emerging' trajectory was associated with higher rates, of anxiety, depression, self-harm and suicidal ideation. In a separate US-based cohort ('SPARK') with available genetic data, the authors reported that age of diagnosis showed significant heritability (based on SNP data), not accounted for by early developmental milestones (e.g. age of walking and first words), intellectual disability nor by parental socio-economic status. The authors also reported SNP-based heritability of age of diagnosis in the Danish iPsych cohort, replicating this main finding, although the mean age of diagnosis markedly differed with the SPARK cohort (mean age 10.98 yr and 4.79 yr for iPsych and SPARK, respectively). The authors concluded that 'age at autism diagnosis is partly heritable and is associated with two moderately correlated autism polygenic factors. One of these factors is associated with earlier diagnosis of autism, lower social and communication abilities in early childhood. The second factor is associated with later autism diagnosis, increased socio-emotional difficulties in adolescence, and has moderate to high positive genetic correlations with Attention-Deficit/Hyperactivity Disorder, mental health conditions, and trauma,' as reported in the Abstract.

Study strengths include large longitudinal sample, comprised of 4 national birth cohorts with a relevant common measure and robust ascertainment for autism, sensitivity analyses to examine for potential effects of sex and co-occurring ADHD, and reporting of mental health profiles. Genetic modeling also involves large (albeit not population) cohorts, and in-depth analysis of potential covariates and confounders that might account for the age of diagnosis with SNP-based and polygenic indices of genetic variation.

Overall, the findings are more robust for the SCQ-based socio-behavioural trajectories derived from the longitudinal population-based cohorts, whereas there are some questions regarding the interpretation of polygenic factors associated with age-of-diagnosis, linked to the cross-sectional cohorts (see below). The justification for integrating the two sets of findings is also somewhat underdeveloped (i.e., since ultimately come from independent samples) and could benefit from further elaboration.

Validity

The main weakness of the manuscript is insufficient attention to the fundamental question of whether 'age of diagnosis' is a biological construct that can be heritable (vs familial) and directly influenced by specific genetic mechanisms. 'Age of diagnosis', as the authors acknowledge in their selection of covariates to assess mediating and moderating relationships, likely reflects the complex interplay among multiple factors. These include the child's clinical profile and other individual factors such as sex, parents/caregivers' actions related to sharing concerns, seeking specialised assessment and access to supports and services, and health systems factors including intensity and accuracy of surveillance for features of autism, accessibility of diagnostic assessment and potential structural barriers to system access and navigation that interact with family-level factors such as race/ethnicity and socio-economic status in ways that can influence age of diagnosis

independent of child-level factors. The conclusion that 'age at autism diagnosis is (partly) heritable' implies that the timing of diagnosis is independent of that broader environmental/health systems context, and reflects only the child's genetics, or perhaps, familial genetic factors (although the authors do not explore that latter).

There are at least two potential influences on age of diagnosis that would imply a more complex relationship than a direct effect of the child's polygenetic indices, both of which the authors explored, but arguably, the available data are insufficient to fully resolve.

The first is that specific clinical features (e.g., language and cognitive level, severity of autistic features) mediate the relationship between polygenetic indices and age of diagnosis. There is extensive evidence that such factors influence timing of diagnosis in community studies (reviewed by Daniels and Mandell, 2014 <https://www.ncbi.nlm.nih.gov/pmc/articles/PMC4775077/>; also see Zwaigenbaum et al, 2019 <https://www.ncbi.nlm.nih.gov/pmc/articles/PMC6376294/>) and in studies of increased likelihood cohorts Ozonoff et al., 2015 <https://www.ncbi.nlm.nih.gov/pmc/articles/PMC4532646/>). It is more biologically plausible that polygenic indices influence such individual-level clinical features, which then influence timing of identification/diagnosis, rather than genetic indices directly influencing timing of diagnosis. In the body of the paper, it is noted that 'child's cognitive aptitude' was incorporated by summary scores derived from principal components analysis (PCA) to capture information measured by diverse scales (pg 23, line 42-43), and the reader is referred to the supplementary materials. The Supplementary Notes provide a general description of the four birth cohorts. Given that these are general population cohorts, it is not surprising that they do not include observational measures of language and intellectual functioning. However, it would be helpful to have further details about how 'development was tracked'; no specific measures were specified. Section 3 of the Supplement on 'Clinical factors: Intellectual Disability and co-occurring developmental delays' largely discounts the possibility that delays contributed to age in diagnosis. However, the authors also note that 'none of the autistic participants had intellectual disability', which raises questions about both the sensitivity of the ascertainment process (which presumably relied on parent report and whether participants had actually received a cognitive assessment), and of how variation in intellectual function was determined. It is surprising that none of the autistic participants had an intellectual disability (ID) given the known association, which suggests under-ascertainment. Data on developmental milestones is insufficient to address these important issues. The authors provide additional sensitivity analyses from the SPARK and iPSYCH dataset in the Supplement ("we find similar over-transmission of the polygenic scores (PGS) for iPSYCHbefore11 and iPSYCHafter10 among autistic individuals with and without ID" and "In SPARK, PGS for iPSYCHafter10 has a similar positive association with age at autism diagnosis even after excluding individuals with ID and limited verbal ability"). However, again, there is little detail about how these details were ascertained, and dichotomizing intellectual and verbal abilities (ie ID or not, limited vs some verbal ability) may have obscured associations. Further clarity is needed regarding what data were available, and consideration of whether these data were insufficient to exclude the possibility that such clinical indices mediated the relationship between polygenic scores and age of diagnosis.

Second, prior research suggests that there are racial disparities in identification/diagnosis of autism. For example, Black (African-American) children in the US are diagnosed less frequently, at later ages, and with apparently increased rates of intellectual disability, suggesting increased diagnostic threshold by some clinicians (Constantino et al (2020). Timing of the diagnosis of autism in African American children. *Pediatrics*, 146(3), e20193629. <https://doi.org/10.1542/peds.2019-3629>; Durkin et al. (2017) Autism spectrum disorder among US children (2002–2010): socioeconomic, racial, and ethnic disparities. *American Journal of Public Health*, 107(11), 1818–1826. <https://doi.org/10.2105/AJPH.2017.304032>; Habayeb et al. (2022). Still Left Behind: Fewer Black School-Aged Youth Receive ASD Diagnoses Compared to White Youth. *J Autism Dev Disord* 52, 2274–2283. <https://doi.org/10.1007/s10803-021-05118-1>. There is insufficient granularity in race/ethnicity data reported in the manuscript, particularly for SPARK and iPSYCH cohorts which provide the polygenic scores data. Ethnicity is measured as 'ethnic minority status' (presumably white vs non-white), which lacks attention to the potential that specific ethnic groups face systemic barriers to access to diagnosis, or that clinical biases may favour identification of autistic diagnoses earlier vs later in some groups. Compounding these potential challenges is that it was not possible to discern the percentage of participants with 'ethnic minority status' in the genetic cohorts (ie the US SPARK and Denmark iPsych cohorts) despite review of the primary manuscript and supplemental materials (notably, Supplementary Figures but not Supplementary Tables referred to in the Supplement were included). 'Ethnic Minority' is listed in the Excel Datafile only in reference to regression analyses (tab SD3); data for the 'birth cohorts used' (tab SD1) is limited to sex and age of diagnosis for the longitudinal cohorts; no descriptive data is provided for the cross-sectional cohorts. The SPARK website <https://www.sfari.org/spark-demographic-and-clinical-information/> (accessed Sep 12, 2024) indicates that 73.1% of participants are white (this may or may not be reflected in the cohort reported in the current manuscript). Notably, the non-white group are diverse and reflected in multiple race/ethnic categories, including 7.1% African American. It is concerning to attribute age-of-diagnosis to genetic differences without sufficient attention to how race/ethnicity may contribute to disparities in access to appropriate assessments and/or clinical biases. This needs further consideration in the manuscript.

Further clarification is also needed regarding how age of diagnosis was stratified to 'childhood vs adolescence'. Cut-off was at age 9 vs 11 years, depending on cohort. The authors noted that 'specific cut off age was cohort-dependent, as different birth cohorts collected information at different ages'. However, Extended Figure 1 suggests some inconsistency in the application of a decision rule to determine the cut-off – LSAC-B and LSAC-K include assessments at both 9 and 11 years, yet a 9-year cut-off was applied to the former and a 11-year cut-off to the latter. Relatedly, not clear why the authors did not apply the cut-off somewhat later (e.g., consistently from 11 years), if their intent was to identify a subgroup diagnosed in adolescence. That said, the authors report SNP heritability for age of diagnosis in the iPsych dataset which indicates a similar pattern whether the cut point is 9 or 11 years (pg 10-11).

Originality and significance:

Perhaps because the primary focus was on the genetic analyses, the finding of distinctive socio-behavioural trajectories received less attention in the Discussion. Distinct trajectories of co-occurring internal/externalizing behavioural features have been reported extensively in longitudinal autistic cohorts (see recent review by Gentles et al (2024). Trajectory research in children with an autism diagnosis: A scoping review. *Autism*. 2024 Mar;28(3):540-564. doi: 10.1177/136236132311702800. The originality and significance of the current findings could be discussed further in that context. Overall, the findings of the manuscript will be of broad interest to clinicians and researchers, with the caveat of concerns regarding the presentation and interpretation of polygenic scores in relation to age of diagnosis, as discussed earlier.

Data & methodology:

The manuscript has many strengths with respect to available data and the extensive analyses reported. Concerns regarding the analyses are covered in the earlier comments. The Figures are excellent overall, but error bars are not consistently defined.

Conclusions:

Concerns regarding conclusions (especially related to the genetic findings) were discussed earlier.

Suggested improvements:

With regards to concerns regarding limitations in the data, I suspect that more detailed data on child-level clinical factors that might influence age of diagnosis are not available. More granular data on race/ethnicity are available for at least the SPARK cohort that could be incorporated into the age-of-diagnoses analyses. While I realize that an evaluation of the nuanced multi-layered factors that may influence age-of-diagnosis is beyond the scope of the current manuscript, more explicit acknowledgement of the limitations in the data and the interpretation of the polygenic analyses could strengthen the papers.

References:

See earlier comments

Clarity and context:

The Abstract is clear and accessible, but revisions should be aligned with revisions related to earlier comments; likewise, the introduction and conclusions, both in terms of the relevance of the socio-behavioural trajectory findings independent of genetic modelling, and the complexity of factors that may influence age of diagnosis, including those not sufficiently covered in the available datasets/study measurement model.

Inflammatory material:

Although the language within the manuscript is respectful and appropriate, the notion of age-of-diagnosis being genetically determined may be offensive to some (and this is also relevant to Springer Nature's commitment to EDI).

Referee #2

(Remarks to the Author)

This is an impressive and potentially important paper in the autism field. It utilizes data from multiple longitudinal population cohorts on age of autism diagnosis and mental health traits/symptoms (assessed using the SDQ) and then genetic data from two large cohorts iPSYCH and SPARK (and PGC and FinGEN). It is cleverly and competently done and very ambitious and also broad in scope with potential impact on both the science and practice of autism. My expertise relates to clinical phenotypic data, and I will restrict most of my comments to this topic.

Main points:

1. The concept and findings are potentially important to help explain the variability in developmental trajectories of individuals who have autism. In some ways the findings from the genetic association shown in Figure 5 corroborate what we from clinical studies – That individuals with later diagnosed autism are more likely to have additional cooccurring mental health difficulties including ADHD, anorexia, depression and PTSD with associated phenomena such as self-harm and childhood maltreatment. One could alternatively characterize the two autism PGS factors not as early versus later diagnosed autism but as autism with or without the presence of this mental health conditions (at least in adolescences when some at least are more routinely identified by CAMHS services). It is the case in the population cohorts that we know that in many communities, including those from which these longitudinal population cohorts are drawn, autism is under-recognized and often only comes to the attention of carers and clinical services because of early to later adolescent onset of emotional and behavioral difficulties. This may be an equally important way of parsing or characterizing heterogeneity in the developmental presentation of autism (in terms of etiological understanding, developmental pathways and mechanisms, and potential indicators for appropriate targeted intervention support at different ages - which is not mentioned in the manuscript), but it would be perhaps less revolutionary than the idea that the authors put forward. In the 4 population cohorts (MCS, GUI, LSAC-B, LSAC-K) as shown in Extended Figure 1 age of diagnosis data is not captured (and from Supplementary Table 1 would be rare in these cohorts) in the preschool years. This contrasts to the very opening sentence of the paper referring to autism not necessarily being manifest by age 3 years and the age of autism diagnosis in the volunteer US SPARK cohort shown in Extended Figure 2 (where the mean is ~ 5 years).

2. This highlights a perhaps critical limitation of bringing the population cohorts and genetic cohorts together in that 'early diagnosis' means different things in different cohorts (e.g. different in iPSYCH than SPARK in Extended Figure 2 and thus

more like the population cohorts in Extended Figure 1 and Supplementary Table 1). The authors are correct that any age threshold of 'early' vs. 'later' diagnosis is arbitrary but their analysis does not tell us about the preschool vs. later (mid-childhood onwards) age of diagnosis that has been more clearly realized in the clinical field (as they recognize in References #7 to #14). This point compliments my alternative characterization of the genetic factors they identify as being about (in one view) autism with/without mental health difficulties rather than early versus later diagnosis. Furthermore, what is missing in these analyses (of both the population and the genetic cohorts) is any measure of autism severity, aside from categorical presence or absence of diagnosis. There is some evidence at least from the ALSPAC study that it is not consistently across the case across developmental time that those who receive an earlier diagnosis have the most severe autistic symptoms. However, in other clinical cohorts they do. It is something that is missing from consideration in the current manuscript. I partly know what measures on autism severity the different cohorts have (I think all have some) and this should be considered by the authors. Relatedly, I was surprised by the lack of effect (as far as I understand it) of IQ and sex in the early vs. later diagnosis differences. One relevant point (a limitation not a flaw) is the modest size of the population cohorts with a diagnosis and from Supplementary Table 1 the sex ratios at different ages are limited by low sample sizes – and I do think the reader needs to be alerted to this point overall as they might not look-see it in this Table – OK it is on p.4 but should be noted in the Limitations). On this point I was not sure why cell sizes are given as <10 when there must be a number?

More minor points as I read through the manuscript:

3. p.7 – Can we see CIs for sex ratios in the cohorts since male:female ratio varies from 3.92 to 2.11 which is substantial (if not significant). Also, the $p > .05$ is less informative than the actual non-significant p-value – please replace.

4. p.7 and throughout – I was not clear even with some checking again what the 'cognitive aptitude' scores was in the different cohorts – primarily a direct IQ / cognitive assessment or something else?

5. p.9, line 17 – Females more likely to be diagnosed with autism later – in other cohorts yes but was this the case in the 4 population cohorts and main 2 genetic cohorts. Later, the authors also say that mental health problems also higher in autistic females vs. males (p.20, line 24) but the Ref #64 is not a strong one and there is good SR/MA data on this. This relates to my comment above that I do not fully follow (this may be my own limitation) quite how the key clinical factors that might affect age of recognition of both autism and mental health problems – IQ and sex – do not seem to be involved in the early vs. late distinction here – is this true both for the population and genetic cohorts and how does this sit with the broader clinical literature (that does not directly consider genetic information)?

6. On pages 10 and 11 the authors present and test different thresholds for early vs. later diagnosis is in iPSYCH but relating to my point about 10/11 years being a late threshold in some ways for late diagnosis and looking again at Extended Figure 2 for age of diagnosis in iPSYCH and SPARK I wondered why they did not also explore at least in SPARK those diagnosed before versus after 5 years (they cannot do this in iPSYCH probably due to the low numbers). It might inform the point I make above.

7. Can the authors comment (for me) more clearly on how formal ID (intellectual disability) is or is not represented in the population and genetic cohorts. It is one of the key diagnostic overshadowing considerations (in addition to ADHD and perhaps oppositionality and anxiety disorders) and there is a mention in the SM p.13 that 'in the birth cohorts, none of the autistic participants has ID' which cannot in fact be true but might represent the data available (ID still is present in between 25% and 33% of children with autism in population samples).

8. I recognize that the authors include where available SES and some family (including parent) characteristics and do a 'country analysis' at one point but contextual and service factors are very influential in the population cohorts where diagnosis relies on parent report of received diagnosis (and this needs to be made clear – it is not a direct study acquired clinical ascertainment) which is dependent of course on services. This applies differently to the genetic cohorts, perhaps especially SPARK as a volunteer family/parent recruited sample with many young diagnosed children.

9. p.19, line 20 – I would characterize the genetic correlation between the factors as 'moderate' (0.38) not 'low' but that might not make sense to geneticists.

10. p.20, line 36 – The paragraph on the camouflaging concept is interesting but for me almost entirely speculative with respect the data presented. Personally, I remain to be convinced that females camouflage more than males (though the authors state this) not necessarily have more mental health problems (the sampling in this area is often biased) but they do not find sex effects on the two groups they identify and have no direct or indirect measures of the concept. Alternative clinical issues to mention would be about (1) what services need to know and (2) does this inform ages and targets for intervention and support of both impairing autism characteristics and mental health difficulties.

Referee #3

(Remarks to the Author)

This paper reports a series of analyses that explore the differentiation of socio-behavioral trajectories and genetic profiles associated with age at Autism diagnosis. Overall, I found the paper to be interesting but rather exploratory. At times, the results seemed somewhat circular in the sense that the efforts to validate the differentiation of early vs. late diagnostic groups were oftentimes simply tests of whether these groups differed in the timing of their symptom presentations.

One salient feature of the paper, although perhaps stylistic, had a large impact on my reading. The authors frequently use

the phrase, “we wondered” when describing the impetus for their analyses (examples below). This gave a strong impression that the manuscript reports a series of exploratory analyses that lead to one another in an unplanned and somewhat arbitrary fashion. I do think that simple declarative sentences could give a very different impression. For instance, rather than stating “given result X, we wondered whether results would persist after including covariate Z,” one could write “We tested the whether results would persist after including covariate Z.” That said, I do believe that the phrasing chosen may be revealing about the scientific process, and I am therefore generally concerned about how much of this manuscript is a report of a series of serendipitous results that stem from a much larger set of exploratory analyses. A preregistration would have been valuable for combatting this possibility, especially given the wide range of analyses reported, the relatively small sample sizes used for the growth mixture models, and the potential for flexibility when defining age groups using cut points.

“Given the significant association between age at autism diagnosis and the GMM latent class membership in LSAC-B and MCS, we wondered if these socio-behavioural trajectories explain any variance in age at autism diagnosis after accounting for socio demographic factors”

“Consequently, we wondered if the associations between ADHD and educational attainment (EA) PGS on age at autism diagnosis were due to parental indirect genetic effects”

“We wondered if the genetic signal for later diagnosed autism can be explained by diagnostic misclassification.”

“Given increasing number of autistic individuals being diagnosed in adolescence, we wondered if there are broad differences in the trajectories”

I found that the focus on SDQ trajectories in relation to early vs. late Autism diagnosis to be somewhat circular. The analysis finds that pre-adolescent diagnoses are associated with difficulties early on that tended to persist, and the adolescent and later diagnoses were associated with fewer difficulties early on, but an increase in difficulties over development. This doesn't seem to be surprising or especially validating of the onset groupings- I would expect that any disorder that is stratified by age at diagnosis would show early elevations in relevant symptoms for the early onset group and increases in relevant symptoms for the later onset group.

The authors go on to fit growth mixture models to the data from individuals with autism diagnoses only (it seems that, despite the data analyzed being from birth cohorts, these analyses are confined to individuals with an autism diagnosis). When they extract two latent trajectory classes, one corresponding to early difficulties and the other corresponding to increased difficulties over development, they find that these groups correspond well with the timing of autism diagnosis. Again, I'm not sure that there is anything particularly surprising about this finding: I would expect that the timing of symptom elevations to correspond well to the timing of diagnoses... I should also say that- given the nonspecific nature of the SDQ questionnaire- if the growth mixture model was fit in the full sample, there would likely be non-specificity with respect to the mapping of trajectory classes to Autism (as opposed to other childhood psychiatric disorders). The sensitivity test using ADHD diagnoses in place of Autism diagnoses seems rather weak.

The GWAS of age of onset is interesting and valuable. This GWAS requires restricting the sample to cases which does some of the interpretation ambiguous, especially with respect to genetic correlations with traits measured in the general population and case-control traits. That the r_g across samples for this GWAS was only .5 is somewhat concerning. The authors are forthcoming about the fact that the different distributions of age at diagnosis are very different across the two samples but don't seem to explain why (different ascertainment strategies?) The genetic correlation between age of diagnosis and the case control autism trait is variably negative (whereas the authors note that for schizophrenia and depression, the age at diagnosis/onset is largely negatively genetically correlated with the phenotype itself, indicating that earlier diagnosis/onset is associated with greater polygenic propensity for the condition). This, like many of the results reported in this manuscript, is an intriguing observation, but does not give much in the way of decisive answers. The authors do conduct an analysis that splits the case control phenotype into early vs. late onset cases, and this appears to provide more consistent results, but such an analysis (like the growth mixture analysis) simply confirms that different ways of stratifying analyses by age at diagnosis agree at least moderately.

The genetic correlation analysis and Genomic SEM analyses (see Figures 4 and 5) are interesting and persuasive that the genetic architecture of early vs. late autisms diagnosis are partly separable, with the early diagnosis polygenic profiles being more characteristic of male cases and later diagnosis polygenic profiles being more characteristic of female cases. There are some concerns, such as the fact that the GWAS phenotypes themselves overlap in their content (the sample overlap is not an issue for the estimators- it's more that the actual definitions of the cases e.g. ipsych before 11 and ipsych before 9 overlap substantially), which ensures clusters of very high r_g s.

I did not find the analysis that takes into account genetic sharing with ADHD to be especially helpful in ruling out the potential for diagnostic misclassification. The genetic correlations presented in panel 5B indicate more substantial genetic sharing between later diagnosed autism (compared to early diagnosed autism) and a range of disorders, indicating that later diagnoses are far less genetically specific. This could result from genuine differences in genetic architecture by disorder onset, but it could also potentially stem from the differential diagnosis being more challenging and thus later diagnoses being less accurate (and hence more likely to contain individuals who present with autism like features caused by non-autism psychiatric conditions). This is heightened by the fact that females are more likely to be diagnosed later and are also more likely to have less clear presentations of autism (such that differential diagnosis is especially challenging for them).

The figures presenting results of PGS analyses often seem to invite the reader to compare the magnitude (not simply direction) of effects for different PGSs. This is highly problematic, because the magnitude of a PGS effect relates to the power of the discovery GWAS on which it is based (which depends on sample size, heritability and polygenicity). Thus, PGSs based on different GWASs should only be compared with respect to direction of effect, rather than magnitude. In contrast, genetic correlations can be directly compared.

A small note: I'm not sure what s.e.m. (used when reporting nearly all effects) stands for. I believe "standard error of measurement," but I believe it should just be standard error, as standard error of measurement is a term that refers to uncertainty in an individual score whereas standard error refers to uncertainty in a parameter estimate.

Referee #4

(Remarks to the Author)

Zhang et al present a compelling case of genetic heterogeneity in autism that is associated with the age of onset/diagnosis and subsequent differences in the trajectory of mental health profiles.

The study is original and highly significant. The paper presents each step systematically, but following the narrative becomes a challenging task since the study involves too many steps. It is hard to keep track of the meandering storyline, a single overview (figure) would greatly facilitate the messaging.

First, the authors observe in four birth cohorts that SDQ scores follow different trajectories that are linked to age at autism diagnosis. The number of autism cases in each birth cohorts are small but the observation (with SDQ scores) is consistent in all four. The birth cohorts are from the same time-period, which I consider a positive. One of the four cohorts quietly disappears in the story (not shown in Figure 1) because the follow-up GMM analysis does not identify a two-trajectory model in this one. A sensitivity analysis included a male-only analysis but why is there no mentioning of a female-only analysis? In the text (page 7 line 12ff) results are presented of a higher rate of depressive symptoms or anxiety but beyond significant p-values, it remains unclear how much of a difference is observed between the two latent classes.

The focus of the story subsequently changes to the genetic evaluation of age at diagnosis of autism. SPARK is used to perform a SNP heritability measurement showing an estimate of $h^2=0.11$. SPARK is an interesting cohort collected by the Simons Foundation and different from the Simon Simplex Collection(SSC). The latter involved deep and consistent phenotyping while SPARK is a community-based effort with more ambivalent data. I wonder why SSC is completely absent from this study. The other concern is that SPARK's range of age of onset is very much skewed toward <10 (extended Fig 2) which is one of the boundaries used in the paper at a later stage. In other words, the variation in age of onset driving the h^2 signal in SPARK is lacking measurements of the >10 range. How does this affect the downstream observations? A little later, another population cohort (iPSYCH) is added to the analyses. Even though phenotype data may not be as deep as for other cohorts, the population-based design of this Danish birth cohort is outstanding. The age of onset distribution is very different from SPARK and the genetic correlation between the onset traits of SPARK and iPSYCH suggests that we are dealing with a different trait, much like major depression is different from schizophrenia (okay a little better) - it explains the different genetic correlation findings presented in figure 2.

A further characterization of the genetic relationship between age at diagnosis and autism is next (page 10). First, some references are listed for patterns observed in schizophrenia and depression, but no mention of a bipolar study that was published 1-2 years ago. To my knowledge, genetic correlations between age at onset and any of the serious mental illnesses have not yielded consistent results. Around this time, the study shifts fully toward the iPSYCH data for a GWAS of autism separated by age at diagnosis. Yet, the genetic findings presented on page 11, lines 22-25, are very convincing. I am less impressed by the statement (on page 13) that "stratifying by age at diagnosis identifies a similar SNP-based heritability between iPSYCH, SPARK and PGC2017". Why is this so important? A similar heritability estimate says nothing about the underlying biology being similar.

The paper now swerves back to phenotype measures and SDQ scores in the MCS cohort but this time not limited to autism cases. The iPSYCH-after11 PGS is significantly associated with increasing emotional behaviors, SDQ scores and other measurements. The effect was less pronounced in another cohort with SDQ scores, ALSPAC. The subtle switch from SDQ scores in autism cases to SDQ scores in the overall cohorts is easily missed. The study moves back to SPARK again and an age-specific GWAS is performed, this time (i) before age 6, (ii) before age 11, and (iii) after age 10. How much overlap is there between (i) and (ii)? Are these groups exclusive or overlapping? Another issue is that another age category is introduced (namely the before age 6) that has no bearing on the previous analyses and why is there no effort made to perform a GWAS in iPSYCH with age <6 ? I consider the findings presented in figure 4b as central and very insightful, but the way to this point in the paper is arduous.

From Figure 4b we can conclude:

1. GWAS_SPARK_before6 is only weakly correlated with GWAS_SPARK_after10, suggesting a different genetic architecture - or is this largely a power effect with the latter group being too small?
2. there remain modest genetic correlations between iPSYCH and SPARK, but the distribution of correlation suggests again that age at diagnosis categories (e.g. early vs later) show the lowest correlation levels;
3. however, the genetic correlation of iPSYCH_before9 and iPSYCH_after10 or 11 remains high - as if iPSYCH data tells a different story than SPARK.
4. SPARK_after10 shows weak to modest correlations with all other GWAS options.

The narrative continues with a genomicSEM analysis that suggests that two genetic latent classes best explain the

heterogeneity. I am also left with the idea that the different cohort studies (SPARK, iPSYCH, FinnGen) represent different traits with only modest genetic overlap. I am therefore less impressed (and not convinced) with the findings that earlier and later diagnosed autism genetic factors are associated with different mental health profiles. Overall, I agree with the authors that "there is substantial variation across the datasets explored" (page 20, line 30/31).

The authors conclude that age at diagnosis is an important axis of heterogeneity of autism. However, they present a convoluted narrative with diverse cohorts, different types of analyses, and subtle changes that may indeed support their claim. The study certainly highlights the heterogeneity of autism spectrum disorder.

Version 1:

Reviewer comments:

Referee #1

(Remarks to the Author)

I appreciated the thorough and comprehensive response from Zhang and co-authors to the initial set of reviews. The manuscript is considerably strengthened the greater clarity of the focus, from the revised title to the schematic summary of the 4 aims and focus on contrasting specific theoretical models. I will focus my review of the revision on their response to feedback as Reviewer 1, with additional comments on the revised manuscript as a whole.

Re: Comment 1.1 that (a) weakness of the manuscript is insufficient attention to the fundamental question of whether 'age of diagnosis' is a biological construct that can be heritable (vs familial) and directly influenced by specific genetic mechanisms. 'Age of diagnosis', as the authors acknowledge in their selection of covariates to assess mediating and moderating relationships, likely reflects the complex interplay among multiple factors.

The authors now acknowledge in the Discussion, 'There is substantial variation across the datasets explored, highlighting that age at diagnosis of autism is immensely complex, varying across geography and time. Local cultural factors, access to healthcare, gender bias, stigma, ethnicity, and camouflaging, likely have an impact on who receives a diagnosis and when.' And in the FAQs, 'It is important to note that in this study, genetics explain only about 11% of the total variation in when someone receives an autism diagnosis'. They added a detailed analysis of Variance in age at autism diagnosis explained by various sociodemographic and clinical factors in the SPARK cohort (Table 1.1) They also found that 'SNP-based heritability did not significantly attenuate after accounting for several of the child's developmental and clinical phenotypes and parental socio-demographic phenotypes... None of the observed clinical factors mediated the SNP-based heritability of age at autism diagnosis.' They present two theoretical models that could account for SNP heritability for age at diagnosis; ie how age at diagnosis is indexing genetic heterogeneity in autism. This could result from a less penetrant version of the same polygenic propensity for earlier diagnosed autism, resulting in a less prominent manifestation of behavioral traits, or from at least two correlated polygenic aetiologies that are developmentally different with earlier and later differentially associated with the two different polygenic aetiologies. Evidence from further analyses (SNP-based heritability is not attenuated when controlling for a range of child's clinical factors) supports the 2nd model. The authors elaborate and anchor the study analyses and findings as contrasting these two models. This clarification is helpful and for the most part addresses my previous concern. I also appreciate the change in Title to "The polygenic architecture of autism varies by age at diagnosis", which a more straightforward description of the study findings and focus than the original.

Re: Comments 1.2a and 1.2b.

The supplemental analyses are compelling. That said, whether polygenic heritability genuinely influenced age of diagnosis independent of clinical factors is still uncertain. The authors report that there is no meaningful attenuation of the SNP-based heritability from the baseline (sex, ID, and genetic principal components) after accounting for parent-reported IQ scores, age at walking independently, age at first words, language regression, any other regression, SCQ total scores, RBS-R total scores. However, they do not have continuous data on language skills, nor directly assessed cognitive level or autistic symptoms that have previously been reported to correlate with age of diagnosis. The authors refer to developmental phenotypes (e.g., in Aims 1 and 4, but the data in the available cohorts is mainly limited to milestones, presence/absence of intellectual disability and 'cognitive aptitude' derived by factor analysis of available, mainly parent-report measure. This limitation reflects the available data but could be acknowledged more explicitly and represents an important caveat to their conclusion that '(observed) clinical factors did not mediate the SNP-based heritability of age at autism diagnosis'. Indeed, in clarifying 'What might this SNP heritability for age at autism diagnosis reflect?' (pg 7 of their Response) they note that the contrasting theoretical models would lead to different patterns and distributions of clinical manifestations at different ages. Would the authors disagree that age of diagnosis ultimately is a proxy for these differences in onset and expression? How else could age of diagnosis be influenced by genetic factors? So it is important to consider the possibility that the lack of evidence of mediation is related to limitations in the available phenotypic data.

Re: the authors comment in the rebuttal regarding my original comment 1b ('Regarding "Tracking of Development", we are unsure exactly what the Reviewer is requesting', pg 15).

This is a specific reference to the original Supplemental material which did refer to developmental tracking/follow-up of the longitudinal cohorts. Specifically, "The GUI study aims to describe and understand the lives of children in Ireland, with Cohort 98' tracking their development from childhood to adulthood" (pg 3, 487479_0_supp_4498569_shhnyr.pdf)

Re: Comment 1.3b regarding ethnicity and variation in age of diagnosis.

I appreciate the authors' clarification of what data on race and ethnicity was available in the various datasets reported and

that the genetic analyses (SPARK and iPSYCH) included only individuals of genetically inferred European ancestries. Recognizing the limitations in the ethnic diversity of available samples and that the study was not designed to examine ethnic variation in age of diagnosis, their addition about the need to extend the findings to individuals of other genetic ancestry in the Limitations section of the Discussion addressed the concern expressed in review of the original submission.

Re: Comment 1. 4 Further clarification is also needed regarding how the age of diagnosis was stratified to 'childhood vs adolescence'.

I appreciate the clarification regarding the rationale for the cut-offs for each cohort, and the sensitivity analyses demonstrating a general effect of age of diagnosis. The authors did not specifically address the points I had included in my comment; i.e., "inconsistency in the application of a decision rule to determine the cut-off – LSAC-B and LSAC-K include assessments at both 9 and 11 years, yet a 9-year cut-off was applied to the former and a 11-year cut-off to the latter. Relatedly, not clear why the authors did not apply the cut-off somewhat later (e.g., consistently from 11 years), if their intent was to identify a subgroup diagnosed in adolescence." For the latter point, the cut-off for the GUI cohort could be the 13-year visit. I suspect the analyses are robust to these decisions, but some clarification would still be helpful, especially the rationale for applying different cut-points to the LSAC-B and LSAC-K cohorts.

; Regarding Comment 1.5 about distinctive socio-behavioural trajectories.

In response to my suggestion that they consider their findings in relation to a recent scoping review on trajectory research in children with an autism diagnosis (Gentles et al., 2024), they noted that of the 55 studies cited in the review, only one study has investigated the trajectories of social, behavioural, emotional and other related autistic traits prospectively, prior to an autism diagnosis, which "prevents us from asking if changes in developmental trajectories are indeed associated with age at autism diagnosis". In reviewing these studies, they also conclude that "Taken together, ours is the first study to systematically investigate if changes in social, emotional behavioural difficulties are associated with age at autism diagnosis." It is important to distinguish between changes in trajectories from changes in difficulties. To assess changes in trajectories, one would need to identify a change in slope or other trajectory parameter linked in time to a specific event (such as age of diagnosis). What the authors report in the paper are differences in trajectories (ie how difficulties change over time) in the age of diagnosis groups as a whole, which is not fundamentally different than other longitudinal cohorts that include autism diagnosis at baseline. So while it is true that a unique feature of the current paper is reporting autism-related developmental trajectories in birth cohorts, it is not clear whether trajectories prior to diagnosis represent 'different pathways into diagnosis' or just different clinical profiles among the heterogeneous condition of autism. While a minor point, I don't fully agree with the authors' statement that "(Although developmental variation is well-characterised in autism, to our knowledge), no study has robustly investigated if this variation is associated with age at autism diagnosis in prospective cohorts". Other studies have reported how developmental trajectories vary by age of diagnosis within their inception cohorts, although this tends to be a narrower range than in the birth cohorts reported in this paper.

Overall, the authors have responded sufficiently to the points that I as well as the other reviewers raised in their reviews and I appreciate the sensitivity and scientific rigour that has guided the supplemental analyses and text revisions.

Referee #2

(Remarks to the Author)
Nature MS 2024-07-15689A-Z

Reviewer #2:

1. Early vs. Later autism vs. autism with/without psychiatric conditions.

The response and additional analyses and the model are helpful in further clarifying that the early vs. later autism diagnosis genetic architecture is not due to overlap of the latter with psychiatric/mental health (MH) conditions. I find the model (in Note 6 SM) helpful.

2. Preschool (early) diagnosis

The MCS diagnosed before 5 years and the LSAC-B LGCMS and Figures helpful in clarifying the 'stepped' pattern of MH SDQ scores in relation to age of autism diagnosis/recognition. This also provides some confirmation that the threshold of ~11 years used in the genetic cohorts might be meaningful.

3. Age threshold of 'early' vs. 'later' diagnosis

The slight change in terminology to 'earlier' vs. 'later' is helpful and I agree no meaningful biological or clinical (or even service/societal) rationale for a non-arbitrary distinction of what constitutes 'early' diagnosis. The authors note that the 9–11-year-old threshold is around puberty (as well as high school transition) and I wonder if some biological factors (whether in part genetic or not) might be important too and should be mentioned).

4. Lack of measure of autism severity in population cohorts.

SDQ might correlate fairly highly with autism questionnaire measures but I'm not sure in the population cohorts this helps as without separate autism severity and socio-emotional-behavioral measures one cannot account for the impact of autism

severity per se in age of diagnosis. Helpful in SPARK to test whether accounting for SCQ and RBR and parent-reported IQ relates to PGS for earlier vs. later diagnosed and age of diagnosis – in SPARK quite low (i.e. 8% for SCQ) but higher in some other cohorts (as in Inset in Extended Data Figure 2).

5. Lack of effect of IQ and sex in the early vs. later diagnosis differences.

The additional analysis incl. in response to R#1 of demographic factors, sex, IQ/cognitive ability socio-emotional problems and autism severity and % variance accounted for are important and add to contribution this paper makes in terms of quantifying vs. other factors (as far as they can be or are measured in the available datasets) to age of diagnosis in SPARK.

This is my main response to this revision. In the very detailed response to R#1 the authors and also in the very helpful FAQ section they clarify that “genetics explain only about 11% of the total variation in when someone receives an autism diagnosis”. This is explored in Extended Data Figure 2 and for SPARK in Table 1.1 and then in the Supplementary Table 9 for MCS and LSAC-B. There is significant variation in the % of variance in age of diagnosis across these 3 cohorts (and I note lower in SPARK than in MCS and LSAC-B though SPARK is a much larger sample of autistic individuals) but the message (again from the FAQ summary – which is a fair lay description of the pattern of findings) the authors contrast the 11% genetic contribution to the ~10% - 60% from these other factors. I would prefer this kind of quantitative comparison to be more ‘up front’ in the main paper and Abstract as it seems to me important in judging the importance (sic) of the novel analysis and finding here. It’s an effect size (as a non-geneticist) that I can get my head around. Across the various cohort/sample data hard to fairly summarize but ~11% is about the same as autism severity (in SPARK) and around the range (or less) across sex and sociodemographic factors and SDQ scores in MCS and LSAC-B (Supplementary Table 9) – which is an accessible ‘effect size’ estimate to my mind i.e. how important is the new ‘polygenic architecture’ factor presented in explaining variation in age of diagnosis.

Minor point but in Extended Data Figure 2 I would order (to-to-bottom) the variable sets: sociodemographic, sex, autism severity, clinical factors, socioemotional (probably more appropriately termed socioemotional-behavioral given SDQ subscales). A further point to ensure is emphasized is that these datasets do not capture other family/service access/contextual factors that also impact on age of diagnosis in these different cohorts (particularly the population ones perhaps).

6. CIs for sex ratios in the cohorts since male:female ratio varies from 3.92 to 2.11 which is substantial (if not significant). Also, the $p > .05$ is less informative than the actual non-significant p-value – please replace.

CIs added. I still find $p > .05$ less informative than knowing the point estimation for a single p-value or minimum p-value for a set of values.

7. IQ/ ‘cognitive aptitude’ scores in the different cohorts

Helpful to clarify role of IQ and cognitive ability.

8. Females more likely to be diagnosed with autism later

As above, sex and IQ now more fully considered and (again as above) the addition of Table 1.1 – and Extended Data Figure 2. Is Table 1.1 in the response letter/rebuttal in the SM I cannot find it?

9. SPARK diagnosed before versus after 5 years.

SPARK before vs. after 6 years age of diagnosis helpful. In this response the authors make the point about ‘strength of gradients’ depending on the age of diagnosis – with perhaps an inflection around ~11 years – and I am not sure this comes across from in the main paper and I think it is helpful as any age threshold is arbitrary (or at least chronological age ones are).

10. Formal ID (intellectual disability) is or is not represented in the population and genetic cohorts.

Some additional analysis presented in MCS.

11. Contextual and service factors are very influential in the population cohorts where diagnosis relies on parent report of received diagnosis.

As above contextual factors now better addressed.

12. Genetic correlation between the factors as ‘moderate’ (0.38) not ‘low’ but that might not make sense to geneticists.

I am happy with the change (but I am not a geneticist).

Referee #3

(Remarks to the Author)

This is a revision of “The polygenic architecture 1 of autism varies by age at diagnosis” (previously titled: “An axis of genetic heterogeneity in autism is 1 indexed by age at diagnosis and is associated with varying developmental and mental health

profiles). I was one of the original reviewers, and am focusing this review both on the overall manuscript and on responsiveness to Reviews 3 and 4 from the original submission.

I am of two minds regarding this revised submission. On the one hand, while the manuscript has improved since the original submission, it is very dense- reporting a large number of analyses of many different datasets that are difficult to follow (both in terms of the text and in terms of the figures) and thematically connect without a great deal of re-reading and cross-checking information across different parts of the manuscript. Given my expertise in neurodevelopment, complex trait genetics, and statistical methods, I am the exact sort of reader who should find this manuscript tractable. So if I am having difficulty, I fear that the more general reader may be entirely lost. On the other hand, I do think there are some genuine findings and meaningful inferences reported that are important for the field. I think that it is interesting and important that the genetic etiology of early-onset ASD is highly dissociable from that of later diagnosed ASD, with distinct patterns of genetic correlates, and distinct trajectories of symptomology. I also believe that a GWAS of age of onset within ASD cases only case-control GWASes of ASD stratified by age of onset are of potentially high value to the field. I found myself wondering whether the dense nature of manuscript and difficulty to follow was attributable to the strict word/page limits required for a Nature submission, especially given the breadth of analyses reported. However, I also found the responses to reviewers to be difficult to follow and often not directly linked to the issues raised, many times enumerating a slew of tangential points for a given concern, rather than directly successinctly addressing the concern. This suggests that perhaps the authors have a more general difficulty clearly and directly articulating their work in an easily digestible way and makes me concerned that the manuscript would require several round of revisions and considerable feedback from reviewers at each juncture before it is ready for publication.

I am not able to fully detail the many points of difficulty that I had in reading the manuscript, but I will provide a few examples. One point is that there are many different datasets used, each of which is represented by a different acronym. Some of the datasets are small and used for growth mixture modeling, some are larger and used for case-control GWAS stratified by age of onset, some are larger and used for age of onset GWAS in cases only, others are larger and used for polygenic prediction. Each dataset has unique features, some of the most salient being the substantial differences in the distribution of ages at which variables are measured and in the distribution of age of ASD onset. These are all important features that must be taken into account, and the relevant information is reported. However, all too often datasets are simply referred to by their acronym without explicit reference to their sample size, analysis performed, or age (of measurement or onset) distribution, even though these features are key to the interpretation provided. For instance, take Figure 3, its associated caption and in-text description. Only after very carefully reading through all of this information am I able to keep in the fore of my mind that the SPARK discovery, SPARK replication, and iPSYCH datasets are used to conduct case-only GWAS of age of onset (which is presented in panel A), and that the covariate control analysis in panel B is conducted specifically with respect to this case-only age-of-onset analysis in the SPARK cohorts only, and that panel C reports genetic correlations between these different case only age-of-onset GWAS with different case-control analyses of ASD (stratified by data source, and in some cases age of onset of cases and sex). Ideally, this information should be so intuitively displayed in the figure that the reader can draw the key inferences without ever reading the caption or main text. Rather, the figures are entirely opaque without very careful readings of such sources.

Further, in the main text, the authors indicate that it is notable that in panel C of Figure 3, “we observed variable genetic correlations between age at autism diagnosis and different GWAS of autism,” however it is unclear why this heterogeneity occurs. Presumably it is because the different case-control GWAS of AD differ in the typical age of onset of the ascertained cases- but this is not made explicit (except for the iPSYCH after 10 and iPSYCH before 11, which I also not are unintuitive names- “10 and under” and “11 and older” would make more sense). This issue carries forward to the analyses presented in Figure 4. Why is it that PGC 2017 loads on the early onset factor and FinnGen loads on the late onset factor? This is interesting but the authors do not make clear whether this would be expected based on how ASD was ascertained in those two datasets. They leave it to the reader to put the pieces together.

The same reliance on the reader to “read between the lines” is apparent in the Discussion. For instance, the authors write: “This two-polygenic trait genetic model also explains the often contradictory patterns of genetic correlations between autism and various neurodevelopmental, and psychiatric phenotypes across different autism GWAS. The variation in genetic correlation between ADHD and autism stratified by age of diagnosis is particularly noteworthy. Older GWAS of autism (including the PGC-2017) were not significantly genetically correlated with ADHD45–47 whereas more recent GWAS for autism have moderate genetic correlations with ADHD48. Genetic correlation analyses (Figure 4) indicate that the genetic correlation with ADHD increases with a later diagnosis of autism.”

By older GWAS of autism, I gather that they are referring to GWAS published several years ago. At first I thought they were referring to GWAS of older age of onset- but that is clearly not the case given that PGC-2017 loads on the early onset factor in Figure 4. Even putting aside this point of confusion, the authors indicate that they have explained a contradictory pattern (differing patterns of genetic correlation between ADHD and ASD by GWAS) because the ADHD-ASD rG varies by age of onset- but they do not complete the thought for the reader because they never make explicit whether the “older” (in time, not age) GWAS indeed disproportionately included earlier age-of-onset cases. Again, the reader is left to put the pieces together.

I would anticipate that the authors may be frustrated by the above- perhaps they feel that the relevant information is indeed in the manuscript and can even point to where it can be found. I do not dispute this. Rather, my comment is that it is the responsibility of the authors to make the information accessible rather than relying on the reader to do the work to search through the manuscript, supplement, and figure captions to find it. Indeed, R4 had also previously written that “It is hard to keep track of the meandering storyline” and “I consider the findings presented in figure 4b as central and very insightful, but the way to this point in the paper is arduous.”

The authors report that “The age at diagnosis in the two cohorts were moderately genetically correlated with each other ($r_g = 0.51$, $s.e. = 0.19$, $P = 7.56 \times 10^{-3}$, between iPSYCH and meta-analysed SPARK) and had variable genetic correlation with other mental health phenotypes (Supplementary Table 11). This is likely due to the different distribution of age at autism diagnoses in the two cohorts (Extended Data Figure 5) because of differing ascertainment.” The statement that this is likely due to the age distributions is speculative and unsatisfying. For instance, what if the authors were to rerun the age of onset GWAS, attempting to better match the age of onset distributions, e.g. by removing all cases with onset after age 10 where Extended Data Figure 5 indicates they diverge most greatly? Reviewers had raised concern about this genetic correlation being rather low, and the response was that the case-control GWAS of early onset were strongly correlated ($\sim .8$) across datasets, as were the case-control GWAS of late onset. However, this is only partially helpful, as the concern about the age of onset GWAS concerns whether the onset phenotype itself in the cases only is a valid and useful analysis.

A technical concern: I was surprised to find the statement “For genomicSEM39 analyses, we first conducted genetic correlation analyses among different autism GWAS using LDSC. This included a multi-ancestry case pseudocontrol GWAS in SPARK93 (6,222 case-pseudocontrol pairs)...” Both LDSC and Genomic SEM can handle multiple ancestries one at a time, but cannot be validly applied to multi-ancestry GWAS. If this was indeed what was done, then the results reported cannot be trusted. Elsewhere the authors indicate that European ancestry LD scores were used, but it appears from the above statement that although European ancestry LD scores were used (this is the independent variable in LD score regression), multi-ancestry GWAS summary data from SPARK were used (this is the dependent variable in LD score regression). This is not appropriate. I do not think in the Discussion, there is the statement “Finally, our genetic analyses focused on genetically inferred European ancestries due to limited GWAS data from other populations.”

I will conclude by mentioning a few items that I found valuable and responsive. I found that Extended Data Figure 7 was a very nice way to demonstrate the differential patterns of r_G by age of onset are not driven by differences in cohorts. I thought Extended Data Figure 1 and Note 6, Figure 1 were nice ways of illustrating the distinct theoretical models of onset-stratified autism etiology. I thought that the analysis represented by Note 6, Figure 2 was a clever way to pragmatically address the issue of potential differences in the accuracy of differential diagnosis by age of onset.

Version 2:

Reviewer comments:

Referee #1

(Remarks to the Author)

I appreciate the authors' careful and thorough response to the previous set of reviews, both clarifying critical findings as well as conceptual issues. No further comments or suggestions, other than to commend the authors on this important work.

Lonnie Zwaigenbaum MD
University of Alberta

Referee #2

(Remarks to the Author)

The authors have been responsive again to Reviews and the manuscript is improved again. It is a fairer and clearer communication of what was done and what was found and their interpretation. I am not going to provide detailed comments on this version. I have reviewed the article and some of the Supplement etc. and will make some minor comments for final consideration.

1. The Abstract correctly describes the 4 independent birth cohort studies but I think it helpful to indicate to the reader here that the genetic analysis is on different samples.

2. Abstract - Is it possible to indicate that in addition to the 2nd (later) polygenic factor having moderate-to-high positive genetic correlations with ADHD and MH conditions that the genetic correlation for the 1st (earlier) polygenic factor is low. Appreciate this might be in part repetition and indirect implicit as stated but that is what I see when I look at Figure 6. Perhaps they can combine the moderate-to-high and the contrast in one sentence?

3. The most important change for me is the contextualisation of the 11% variance with other variance (that they have available) $\sim 15\%$ and this is in the Abstract. I do keep going back to Extended Data Figure 2 and Suppl Table 1 to look at the data and I would like to see these more prominent though appreciate this might not be possible. Extended Data Figure 2 is quite hard to read but I can see no easy solution...

Referee #3

(Remarks to the Author)

This is my third review of the submission that is now titled “The polygenic and developmental profiles of autism differ by age at diagnosis.” The manuscript has evolved considerably since the first submission, and the authors have put a great

deal of effort into updating analyses and revising the text and figures based on previous feedback. The updated figures are a major improvement in terms of clarity of communication, and Box 1 is a helpful new addition to aid the reader in accounting for all of the many datasets and their features. I do believe that this paper is innovative and important in establishing that autism diagnoses made very early in child development do not simply differ in their levels of severity, but are considerably etiologically distinct (and therefore probably also taxonomically distinct) from those made later in child development and adolescence/early adulthood. Not only does this result enhance scientific understanding, but it helps to resolve previous discrepancies in the literature, and it informs decisions on how data should be aggregated across individuals and cohorts in future research.

I had previously indicated that the manuscript was a long way from being clear and articulate in a way that did not impose on the reader to assemble the different pieces of information and results together in their own head in order to make sense of the conclusions drawn. I wrote in my last review that I was "concerned that the manuscript would require several round of revisions and considerable feedback from reviewers at each juncture before it is ready for publication." In these respects, I think that the manuscript is much improved. I continue to think that the manuscript (text and figures) would benefit from some additional revisions for clarity of communication, but I don't think that additional reviews are appropriate after this. For instance, in Figure 5, panel B it is still not clear without cross-referencing with other panels that PGC (2017) has a very low median age (3.5) at diagnosis and FinnGen has a very high (22.66) age at diagnosis. Yes, the information is there in panel A, but it would be so much more intuitive if the median age for each of the 6 GWAS samples in panel B were also printed right under the sample name. Similarly, Note 6, Figure 1 is fantastic and the exact sort of analysis that I had suggested. It really drives home the inference that the genetic correlation between different age at diagnosis GWAS depends on the developmental period under consideration, and directly supports the inference that the moderate genetic correlation between the SPARK and iPSYCH age at autism diagnosis GWASs can be attributed to the differing median age at diagnosis. This is really stunning, and could even be included in the main text. However, The Y axis of the plot in panel A refers to subsets but never makes clear that it is subsets of SPARK (even though the parallel plot in panel B does indicate that the subsets on the Y axis are for iPsych). It is also not clear what the numbers in parentheses on the X and Y axis tick labels in both panels are without digging into the figure note (they are the median age and median absolute deviation in age from the median). I just don't see why the reader should have to do work to make sense of this figure when some clear labelling would make the figure extremely intuitive to understand without even having to read the caption. If the manuscript is accepted, I would hope that copy editors and journal staff would help with these further issues, so as to optimize clarity and readability. Like I indicated in my previous review, it's not that the information is missing. It's about presenting it clearly and accessibly.

I have been asked to comment on whether the language around describing "autistic individuals" (and similar) is appropriate - or would the use of "individuals with autism" (and similar) be more appropriate? The most appropriate language of course shifts with time, and varies according to specific diagnosis and the preferences of the communities of affected individuals. My understanding is that while there is not a consensus among all individuals within the community, there has been a recent movement toward identity-first language in the case of autism. In other words, many individuals in the autism community prefer the term "autistic person" over "person with autism." However, others still prefer person-first language (i.e. "person with autism"). I do not think that there is a clearly correct or more appropriate term, but I do believe that identify-first is becoming increasingly popular, and I am in support the language as it is currently used in the manuscript, if that is the preference of the authors (who themselves have considerable experience working in this research area).

Response to reviewer's comments

Referee #1 (Remarks to the Author):	2
Referee #2 (Remarks to the Author):	23
Referee #3 (Remarks to the Author):	40
Referee #4 (Remarks to the Author):	50
References	62

Referee #1 (Remarks to the Author):

Summary of key results:

The authors utilized four birth cohorts from different areas of the world (UK, Ireland and Australia) to examine 'socio-behavioural trajectories'; that is, change over time in 'Strengths and Difficulties Questionnaire' scores, as assessed using Latent Growth Curve Models. They assessed whether trajectories differed based on whether a child had ever been diagnosed with autism, and whether there were differences in trajectories among autistic children, related to age of autism diagnosis, dichotomized at age 9 or 11, depending on cohort. Sensitivity analyses indicated that these trajectories were relatively specific to autism (vs ADHD) and did not reflect sex differences in age of diagnosis. SCQ trajectory membership was also associated with mental health outcomes; specifically, the 'late childhood emerging' trajectory was associated with higher rates, of anxiety, depression, self-harm and suicidal ideation. In a separate US-based cohort ('SPARK') with available genetic data, the authors reported that age of diagnosis showed significant heritability (based on SNP data), not accounted for by early developmental milestones (e.g. age of walking and first words), intellectual disability nor by parental socio-economic status. The authors also reported SNP-based heritability of age of diagnosis in the Danish iPsych cohort, replicating this main finding, although the mean age of diagnosis markedly differed with the SPARK cohort (mean age 10.98 yr and 4.79 yr for iPsych and SPARK, respectively). The authors concluded that 'age at autism diagnosis is partly heritable and is associated with two moderately correlated autism polygenic factors. One of these factors is associated with earlier diagnosis of autism, lower social and communication abilities in early childhood. The second factor is associated with later autism diagnosis, increased socio-emotional difficulties in adolescence, and has moderate to high positive genetic correlations with Attention-Deficit/Hyperactivity Disorder, mental health conditions, and trauma,' as reported in the Abstract.

Study strengths include a large longitudinal sample, comprising 4 national birth cohorts with a relevant common measure and robust ascertainment for autism, sensitivity analyses to examine for potential effects of sex and co-occurring ADHD, and reporting of mental health profiles. Genetic modeling also involves large (albeit not population) cohorts, and in-depth analysis of potential covariates and confounders that might account for the age of diagnosis with SNP-based and polygenic indices of genetic variation.

Overall, the findings are more robust for the SCQ-based socio-behavioural trajectories derived from the longitudinal population-based cohorts, whereas there are some questions regarding the interpretation of polygenic factors associated with age-of-diagnosis, linked to the cross-sectional cohorts (see below). The justification for integrating the two sets of findings is also somewhat underdeveloped (i.e., since ultimately they come from independent samples) and could benefit from further elaboration.

We thank the reviewer for their appraisal of the study and provide our responses to their concerns below. We have conducted several additional analyses to demonstrate the robustness of the genetic analyses, and better articulated key theoretical models to help understand our findings. In the Discussion, we have further discussed the concordant results between the genetic findings and the birth cohort findings to better integrate the two results. Please find our detailed response below.

Validity

1.1. The main weakness of the manuscript is insufficient attention to the fundamental question of whether ‘age of diagnosis’ is a biological construct that can be heritable (vs familial) and directly influenced by specific genetic mechanisms. ‘Age of diagnosis’, as the authors acknowledge in their selection of covariates to assess mediating and moderating relationships, likely reflects the complex interplay among multiple factors. These include the child’s clinical profile and other individual factors such as sex, parents/caregivers’ actions related to sharing concerns, seeking specialised assessment and access to supports and services, and health systems factors including intensity and accuracy of surveillance for features of autism, accessibility of diagnostic assessment and potential structural barriers to system access and navigation that interact with family-level factors such as race/ethnicity and socio-economic status in ways that can influence age of diagnosis independent of child-level factors. The conclusion that ‘age at autism diagnosis is (partly) heritable’ implies that the timing of diagnosis is independent of that broader environmental/health systems context, and reflects only the child’s genetics, or perhaps, familial genetic factors (although the authors do not explore that latter).

Thank you for raising this important point. We address this concern in detail below.

Part1: Age at autism diagnosis is complex.

We fully agree that age at autism diagnosis is complex, and multifactorial. Indeed, we clearly mention this in the Discussion:

“There is substantial variation across the datasets explored, highlighting that age at diagnosis of autism is immensely complex, varying across geography and time. Local cultural factors, access to healthcare, gender bias, stigma, ethnicity, and camouflaging, likely have an impact on who receives a diagnosis and when.”

And in the FAQs:

“What is the impact of factors like camouflaging, waiting time, stigma on age at autism diagnosis?”

The study did not directly measure the impact of factors like camouflaging autistic traits, delays in assessment/diagnosis waitlists, or stigma on age at autism diagnosis. However, we acknowledge that these likely play a role, in addition to the genetic, some demographic, and developmental factors that we have studied.

It is important to note that in this study, genetics explain only about 11% of the total variation in when someone receives an autism diagnosis. Similarly, developmental and some demographic factors explain between 10 - 60% of the variance in age at autism diagnosis across cohorts, with considerable variation among the cohorts studied, likely due to geographical and ascertainment differences across cohorts. Taken together, it is clear that there are several other unmeasured factors that contribute to when someone receives an autism diagnosis.

More research is needed to understand how much these social/environmental variables influence the timing of when someone receives an autism diagnosis.”

Changes made: However, we further address this complexity in two ways. First, we conducted a review of the variance in the age at autism diagnosis explained by various sociodemographic and clinical factors, including sex and autism severity. Second in SPARK, we investigated the variance explained in age at autism diagnosis by multiple other variables. Third, we have rewritten the introduction and abstract to focus the reader on the primary finding - that earlier and later diagnosed autism are modestly genetically correlated.

We explain this in the Methods:

“Variance explained: literature review and analyses of the SPARK cohort

To contextualise the SNP-based heritability and the variance explained by SDQ total and subscale scores, we conducted a review of the variance in the age at autism diagnosis explained by various sociodemographic and clinical factors, including sex and autism severity.

We systematically searched for studies that investigated the factors contributing to the age at diagnosis of autism. Using Google Scholar and PubMed, we searched for studies published between 1998 and December 10, 2024. The search included combinations of the following terms in the title or abstract: "age at diagnosis" AND "autism", "autism" AND "age," and "diagnosis age" AND "autism". To ensure comprehensiveness, we also used these search terms with alternative terminology for autism, including "autism spectrum condition," "autism spectrum disorder," and "ASD". This initial search resulted in over 1,700 studies.

To refine this pool, we manually reviewed the identified studies to determine whether they explored factors that contribute to or affect the age at receiving an autism diagnosis. 184 studies met these criteria. Among these, we further narrowed our focus to studies that quantitatively assessed the variance in age at diagnosis using statistical measures such as R^2 or η^2 . Only 13 studies meeting these quantitative criteria were included in our final analysis (see **Supplementary Table 1** for a summary of the studies and samples).

In SPARK, we used the v9 release of the phenotypic data. We focussed on the variance explained by sociodemographic factors (sex, reported race, household income, mother’s education, and father’s education), cognitive and developmental factors (reported IQ score, reported intellectual disability, age at walking independently, age at first words, language regression, other regression), and autism severity (scores on the Social Communication Questionnaire and Repetitive Behaviour Scale-Revised); as these factors have been shown in our review to influence the variance of the age at autism diagnosis and were available in our current dataset. We analysed data from 5,773 autistic individuals diagnosed before age 22 (excluding any cases with missing data). To understand what factors influence the timing of autism diagnosis, we used a relative importance analysis (LMG method)¹ to calculate how much each variable contributed to explaining differences in diagnosis age. We performed

this analysis using the *relaimpo* package in R, which allowed us to examine all variables' contributions simultaneously."

What is clear is that most factors individually explain typically less than 15% of the variance in age at autism diagnosis. This is provided in **Extended Figure 2**.

Furthermore, in SPARK, we demonstrate that, similar to previous literature, each individual factor explains less than 10% of the variance (**Table 1.1**).

Extended Data Figure 2: Variance explained (R^2 or η^2) in autism diagnosis by clinical and socio-demographic factors, identified from the review of literature (1998-2023). Variables are grouped into socio-demographic (MAD, SES), socioemotional (SDQ scores), sex, clinical (e.g., IQ, regression, language ability), and autism severity (e.g., SCQ, ADOS, ADI-R, RBS-R) categories. Point size represents sample size, with larger circles indicating larger cohorts. Greyscale shading indicates publication year, with darker shades representing more recent studies. Colored points denote variables analyzed in the current study. Inset shows studies where the variance explained (R^2) > 10%. **Abbreviations:** MAD, Maternal Age at Delivery; IQ, Intelligence Quotient; SES, Socio-economic Status; ADOS, Autism Diagnostic Observation Schedule; ADI-R, Autism Diagnostic Interview-Revised; RBS-R, Repetitive Behavior Scale-Revised; SCQ, Social Communication Questionnaire; SDQ, Strengths and Difficulties Questionnaire.

Table 1.1: Variance in age at autism diagnosis explained by various sociodemographic and clinical factors in the SPARK cohort (N = 5,773)

Category	Subcategory	%variance explained
Sociodemographic factors	Race/ethnicity	0.01
Sociodemographic factors	Family income	0.15
Sociodemographic factors	Mother's highest education	0.07
Sociodemographic factors	Father's highest education	0.41
Sociodemographic factors	Sex	1.55
All sociodemographic factors		2.19%
Clinical Factors	Reported cognitive test score	2.95
Clinical Factors	Reported cognitive impairment	0.67
Clinical Factors	SCQ total score	8.31
Clinical Factors	RBS-R total score	0.60
Clinical Factors	Age at walking independently	0.12
Clinical Factors	Age at using words	2.88
Clinical Factors	Language regression	5.36
Clinical Factors	Other regression	0.18
All clinical factors		21.07%
Total variance (sociodemographic + clinical)		23.27%
SNP-based heritability		11%
Total variance (sociodemographic + clinical + SNP-based heritability)		34.00%

We also refer to these findings in the text with these two changes:

“We examined how latent class membership, socio-demographic factors, and cognitive aptitude related to age at autism diagnosis. The full model explained 17.4% (LSAC-B) to 35.0% (MCS) of variance in diagnosis age (**Supplementary Table 9**). The GMM latent classes for SDQ total and subscales alone explained 9.9 - 30.3% of variance in age at autism diagnosis. In MCS-imputed data, these latent classes explained greater variance: 59.8% and 56.6% respectively (**Supplementary Table 3**). Overall, socio-demographic variables explained 1.9 - 3.7% of the total variance in different cohorts, and this was in line with previous estimates (**Extended Data Figure 2**). Socio-demographic effects on diagnosis age were independent of SDQ latent classes, with no mediation observed (**Supplementary Table 9, Supplementary Note 5**).”

and

“Under Model 1, we may expect this heritability to attenuate when accounting for several clinical factors, including scores on measures of core autism features. However, this SNP-based heritability did not significantly attenuate after accounting for several of the child’s developmental and clinical phenotypes and parental socio-demographic phenotypes (**Figure 3B, Supplementary Table 10**). Furthermore, the variance in age at autism diagnosis explained by SNPs is larger than the variance explained by several other clinical and sociodemographic factors tested (**Extended Data Figure 2**). Taken together, this suggests that none of the observed clinical factors mediate the SNP-based heritability of age at autism diagnosis.”

Part 2: Age at autism diagnosis is only partly heritable and examining what this heritability means using two theoretical models.

We do apologise for any misunderstanding, but we do not suggest that age at diagnosis is *per se* a biological construct. Rather, as provided in our original title, we suggest that age at diagnosis is indexing genetic heterogeneity in autism. To this end, we respectfully disagree with the reviewer’s statement: “The conclusion that ‘age at autism diagnosis is (partly) heritable’ implies that the timing of diagnosis is independent of that broader environmental/health systems context, and reflects only the child’s genetics, or perhaps, familial genetic factors (although the authors do not explore that latter).” As stated in the manuscript, the SNP heritability of age at autism diagnosis is about 11%. This suggests that a large fraction of the variance in age at autism diagnosis is attributable to several other factors. This would include broader environmental factors (access to diagnostic services), familial factors (parental awareness, socio-economic status), and child-based factors (child’s developmental trajectories).

What might this SNP heritability for age at autism diagnosis reflect? One aetiological explanation for later age at diagnosis is that a less penetrant version of the same polygenic propensity for earlier diagnosed autism, resulting in a less prominent manifestation of behavioral traits (Model 1). An alternate model (model 2), suggests that there are at least two correlated polygenic aetiologies that are developmentally different with earlier and later differentially associated with the two different polygenic aetiologies. Subsequently, the SNP heritability of age at autism diagnosis reflects a shift from one polygenic aetiology to another. Please see our **Extended Data Figure 1** which we provide below.

Extended Data Figure 1: Two aetiological polygenic models of autism

In Model 1, we assume a single liability threshold polygenic model. In this model, autism emerges from a unitary polygenic aetiology. Autistic individuals diagnosed later have lower polygenic predisposition than individuals diagnosed earlier. In Model 2, we model two correlated age-dependent polygenic liabilities.

Under Model 1, we will expect to observe three key results: First, there will be a strong negative genetic correlation between all GWAS of autism and (increasing) age at autism diagnosis. Second, there will be high genetic correlation between different age at diagnosis stratified GWAS. Third, the SNP-heritability of age at autism diagnosis will be substantially attenuated once features of severity are accounted for (e.g., scores on the Social Communication Questionnaire, Developmental Milestones). Across our analyses, we observe none of the following, and, in reality, we observe low genetic correlation between earlier and later diagnosed autism even within the same cohort, providing evidence for the second Model.

Through a series of further analyses (detailed below), we clearly demonstrate that this SNP-based heritability is not attenuated when controlling for a range of child's clinical factors (developmental milestones, cognitive test score, language regression, level of support needed) or measures of autistic feature scores (SCQ and RBS-R), or other broad indicators of access - parental socio-economic status and area deprivation (**Figure 3A and B**). Furthermore, Together, this does not support Model 1. Rather, this SNP heritability is reflecting the fact that there are two different but correlated underlying polygenic latent traits that are correlated with age at diagnosis. This will generate a significant heritability for age at autism diagnosis. We further demonstrate this using factor analyses of GWAS for autism later on.

Figure 3: Heritability of age at autism diagnosis, and genetic correlations with various autism GWAS. A. SNP-based heritability for age at autism diagnosis in the SPARK (calculated using GCTA-GREML) and iPSYCH (calculated using LDSC). **B.** SNP-based heritability (GCTA-GREML) in the SPARK cohorts after accounting for various clinical and sociodemographic factors. ‘+’ indicates the baseline model in addition to the specified covariates. The x-axis has been truncated at 0 and 0.25. For A and B, points represent SNP-based heritability estimates and whiskers indicate 95% confidence intervals. **C.** Genetic correlation between age at autism GWAS in SPARK (meta-analysed from discovery and replication cohorts) and iPSYCH and various autism GWAS. Points represent genetic correlation estimates and whiskers indicate 95% confidence intervals.

We have included these analyses in the Results - please see the section “Age at autism diagnosis is partly heritable”.

We have also introduced the two models more clearly in the introduction.

Finally, we have changed the title of the manuscript to: “The polygenic architecture of autism varies by age at diagnosis” This change explicitly mentions our key finding.

Part 3: Empirical data supporting Model 2

A GWAS typically examines genetic differences between groups by comparing allele frequencies across the genome. In the case of age at autism diagnosis, we observe significant SNP heritability - but what does this actually mean? This heritability emerges from two distinct but correlated polygenic liabilities that influence when an autism diagnosis occurs.

The first polygenic liability influences earlier diagnosis, potentially through genetic variants affecting early neurodevelopmental trajectories, more pronounced early features, or earlier-emerging behavioral differences. The second polygenic liability influences later diagnosis, possibly through different presentation patterns, or later-emerging features. These two sets of genetic influences are moderately correlated ($r \approx 0.38$ (0.07)), suggesting that there are different alleles influencing these two traits.

When we conduct a GWAS on age at diagnosis, we are effectively capturing the variance in timing of diagnosis that can be explained by these two polygenic influences. Some genetic variants will systematically associate with earlier diagnosis ages, while others will associate with later diagnosis ages (**Please see Extended Data Figure 1 above**). The observed SNP heritability represents the difference in allelic effect of both sets of variants on diagnosis timing.

Importantly, this dual polygenic architecture explains why the SNP heritability remains significant even after controlling for clinical features, developmental milestones, and family factors. Rather than reflecting a single continuous genetic influence on age at diagnosis, the heritability captures how these two distinct but correlated polygenic traits systematically influence when autism becomes clinically apparent and is diagnosed.

This can be conceptualized as analogous to how height has different genetic influences acting at different developmental stages - some genes influence early growth while others affect pubertal growth spurts, yet together they create heritability in final adult height. Similarly, these two autism-related polygenic traits act at different developmental times but together create heritability in age at diagnosis.

In response to this comment, we have made several key changes throughout the manuscript that we do not reproduce here. Specifically, we have better articulated the two Models in the Introduction, and produced a schematic figure illustrating the two Models. In Section 2 of the Results, in particular, we have contextualised the findings based on whether they are consistent with Model 1 or Model 2. We have also run additional sensitivity analyses (detailed below).

1.2a There are at least two potential influences on age of diagnosis that would imply a more complex relationship than a direct effect of the child's polygenetic indices, both of which the authors explored, but arguably, the available data are insufficient to fully resolve.

The first is that specific clinical features (e.g., language and cognitive level, severity of autistic features) mediate the relationship between polygenetic indices and age of diagnosis. There is extensive evidence that such factors influence timing of diagnosis in community studies (reviewed by Daniels and Mandell, 2014 <https://www.ncbi.nlm.nih.gov/pmc/articles/PMC4775077/>; also see Zwaigenbaum et al, 2019 <https://www.ncbi.nlm.nih.gov/pmc/articles/PMC6376294/>) and in studies of increased likelihood cohorts Ozonoff et al., 2015 <https://www.ncbi.nlm.nih.gov/pmc/articles/PMC4532646/>). It is more biologically plausible that polygenetic indices influence such individual-level clinical features, which then influence timing of identification/diagnosis, rather than genetic indices directly influencing timing of diagnosis.

Thank you for raising this important critical point. We agree, and as the reviewer would have noted, had conducted additional analyses investigating if some clinical variables affect the SNP-based heritability (age at walking and first words, and a binary indicator of ID). We have now conducted further analyses to supplement existing analyses that the heritability that we are observing is independent of the aforementioned clinical features. We find no meaningful attenuation of the SNP-based heritability from the baseline (sex, ID, and genetic principal components) after accounting for the following covariates: parent-reported IQ scores, age at walking independently, age at first words, language regression, any other regression, SCQ total scores, RBS-R total scores, parental education and family income (SES), area deprivation. Please see **Figure 3** which we reproduce below:

Figure 3: Heritability of age at autism diagnosis, and genetic correlations with various autism GWAS. A. SNP-based heritability for age at autism diagnosis in the SPARK (calculated using GCTA-GREML) and iPSYCH (calculated using LDSC). B. SNP-based heritability (GCTA-GREML) in the SPARK cohorts after accounting for various clinical and sociodemographic factors. ‘+’ indicates the baseline model in addition to the specified covariates. The x-axis has been truncated at 0 and 0.25. For A and B, points represent SNP-based heritability estimates and whiskers indicate 95% confidence intervals. C. Genetic correlation between age at autism GWAS in SPARK (meta-analysed from discovery and replication cohorts) and iPSYCH and various autism GWAS. Points represent genetic correlation estimates and whiskers indicate 95% confidence intervals.

Second, in the existing paper, we use polygenic scores to demonstrate that PGS for later diagnosed autism is associated with increasing age at autism diagnosis in SPARK. Now, we extend this analysis to include a range of other clinical variables as covariates: Binary indicator of ID and Reported IQ scores (as measures of cognition), SCQ and RBS-R total scores (as measures of autism severity), age at first words, age at walking independently, any

language regression, and any other regression (as measures of developmental delay and regression). We run additional models after accounting for parental SES, and area deprivation index, ADHD (to account for diagnostic overshadowing), trio status (accounting for potential participation bias), and DSM diagnosis (DSM IV vs DSM 5, to account for changes to the diagnostic criteria). Across all of this, the iPSYCH after 10 PGS was still associated with a later age at autism diagnosis. Please see **Extended Data Figure 6B** which we reproduce below.

Extended Data Figure 6: PGS association with age at autism diagnosis

A. Over-transmission of PGS for iPSYCH_{before11} and iPSYCH_{after10} from parents to autistic children in the SPARK cohort (meta-analysed estimates). Estimates provided for unstratified, sex-stratified, and ID-stratified analyses. Children's PGS have been standardised to parental mean PGS, with the line at zero indicating no over-transmission. **B.** Association between age at autism diagnosis PGS for iPSYCH_{before11} and iPSYCH_{after10} in the SPARK cohort (meta-analysed estimates). Estimates provided after correcting for ID, sex and PCs (baseline), and additionally: Model 2: developmental (dev.) milestones, Model 3: dev. Milestones, dev. Regression, SCQ, RBS-R and IQ scores; Model 4: dev. Milestones and parental socio-economic status (SES); and Model 5: dev. milestones, parental socioeconomic, and area deprivation. In Model 7, we controlled for ADHD, in Model 8 DSM-status (DSM-IV vs DSM-5), and in Model 9, trio status. + indicates that baseline covariates were also included. **C.** Association between autism diagnosed in childhood and adolescence and PGS for iPSYCH_{before11} and iPSYCH_{after10} GWAS in the MCS cohort. For all plots, points indicate the estimate, whiskers indicate 95% confidence intervals.

We note that we are unable to control for ADOS and ADI-R as there are a limited number of participants in SPARK who have completed them. However, there is reasonably high correlation (~ 0.7) between SCQ and ADI-R². Similarly, we are unable to control for IQ scores derived from standardised tests for similar reasons, but we observe high correlation between parent reported IQ scores (reported in IQ bins) and fSIQ scores from standardised tests (Please see **Figure 1.2a** below).

Figure 1.2a: Parent reported cognitive test scores against IQ scores measured on standardised tests among autistic individuals from the SPARK cohort. Median values are given along with the interquartile ranges.

In addition, as mentioned in our response to point 1.1, we quantify the variance explained by the clinical factors both in the wider literature and SPARK, and demonstrate that each individually typically explains less than 10% of the variance. Finally, we also more clearly explain the theoretical framework using two alternate models in the main manuscript, demonstrating that Model 2 can explain many of the observed results, as explained in our response to 1.1. In sum, we show that the association between later diagnosed autism PGS is robust to the inclusion of any of these covariates, suggesting that the association between PGS and age at autism diagnosis is not mediated through these variables.

1.2b. In the body of the paper, it is noted that ‘child’s cognitive aptitude’ was incorporated by summary scores derived from principal components analysis (PCA) to capture information measured by diverse scales (pg 23, line 42-43), and the reader is referred to the supplementary materials. The Supplementary Notes provide a general description of the four birth cohorts. Given that these are general population cohorts, it is not surprising that they do not include observational measures of language and intellectual functioning. However, it would be helpful to have further details about how ‘development was tracked’; no specific measures were specified.

Apologies for not providing this more clearly. In the birth cohorts, intellectual disability was measured as scoring two standard deviations below the mean on the summary measure derived from principal component analysis (PC1). We have updated the Methods in response to this:

“In MCS, we identified multiple variables linked to cognitive ability, SES, and area deprivation. Similarly, in LSAC-B, we identified multiple measures linked to SES, although there were no measures with sufficient sample size linked to cognitive ability or area deprivation. Subsequently, we conducted principal component analysis (PCA) for cognitive abilities, SES, and area deprivation in MCS, and for SES in LSAC-B. PCA was performed using a wide range of measures collected across sweeps (**Supplementary Table 27**), with one variable excluded from any pair showing a correlation coefficient greater than 0.70 to address multicollinearity. The first principal component (PC1) explained more than 40% of the variance for each corresponding factor, while subsequent components contributed substantially less, supporting the use of the respective PC1 as the summary measure for cognitive ability, SES, and area deprivation (**Supplementary Table 27**).

Intellectual disability (ID) was defined as scoring two standard deviations below the mean on the PC1 of the cognitive aptitude factor, consistent with prior studies. No autistic children in the MCS, GUI, or LSAC cohorts who had measures of cognitive aptitude met the criteria for ID, likely due to participation bias. All PCA analyses were conducted in R using the *prcomp()* function³.”

For the LSAC-B cohort, the IQ/cognitive ability summary score was not included in regression or mediation models, as only 58 autistic children (46.4% of the autistic cohort) had complete records for cognitive ability measures over time. Among these 58 children, no cases of intellectual disability, as defined above, were identified. However, it is important to note that the remaining 67 children without complete records may include cases of intellectual disability, which could have contributed to the missing data. This potential bias due to missing data should be considered when interpreting the results. However, there was no significant difference in the mean age at diagnosis between children with complete cognitive ability measures (mean = 10.61 years) and those without (mean = 10.79 years, Welch’s t-test: $t(115.76) = .45, p = .651, 95\% \text{ CI } [-0.61, 0.97]$: Please see figure below).

Regarding “Tracking of Development”, we are unsure exactly what the Reviewer is requesting. As these are birth cohorts, we have limited information on many autism specific measures, although we have run several additional analyses in the SPARK cohort including the variance in age at autism diagnosis explained by several factors as provided in our response to point 1.1. Second, it is worth noting that our primary aim was not to comprehensively track the development of cognitive abilities or other areas beyond general developmental differences. Rather, we used SDQ as a broad measure to capture differences in developmental trajectories among children diagnosed at varying ages, followed by more in-depth genetic analyses.

1.3a Section 3 of the Supplement on ‘Clinical factors: Intellectual Disability and co-occurring developmental delays’ largely discounts the possibility that delays contributed to age in diagnosis. However, the authors also note that ‘none of the autistic participants had intellectual disability’, which raises questions about both the sensitivity of the ascertainment process (which presumably relied on parent report and whether participants had actually received a cognitive assessment), and of how variation in intellectual function was determined. It is surprising that none of the autistic participants had an intellectual disability (ID) given the known association, which suggests under-ascertainment. Data on developmental milestones is insufficient to address these important issues. The authors provide additional sensitivity analyses from the SPARK and iPSYCH dataset in the Supplement (“we find similar over-transmission of the polygenic scores (PGS) for iPSYCHbefore11 and iPSYCHafter10 among autistic individuals with and without ID” and “In SPARK, PGS for iPSYCHafter10 has a similar positive association with age at autism diagnosis even after excluding individuals with ID and limited verbal ability”). However, again, there is little detail about how these details were ascertained, and dichotomizing intellectual and verbal abilities (ie ID or not, limited vs some verbal ability) may have obscured associations. Further clarity is needed regarding what data were available, and consideration of whether these data were insufficient to exclude the possibility that such clinical indices mediated the relationship between polygenic scores and age of diagnosis.

Thank you for raising this point. It's helpful to distinguish the findings from the longitudinal birth cohorts from that of SPARK. As mentioned above, none of the autistic participants with complete cognitive ability measures in the birth cohorts had any reported intellectual disability. This could be due to participation bias in the cohort. We have provided this in the updated Methods section.

"Intellectual disability (ID) was defined as scoring two standard deviations below the mean on the PC1 of the cognitive aptitude factor, consistent with prior studies. No autistic children in the MCS, GUI, or LSAC cohorts who had measures of cognitive aptitude met the criteria for ID, likely due to participation bias. All PCA analyses were conducted in R using the *prcomp()* function³."

In SPARK, there certainly are individuals with intellectual disability. 16.34% of individuals in SPARK have an intellectual disability (Now included in the methods: "The baseline model included ID (16.34% had caregiver reported ID), sex, and the first 10 genetic principal components as covariates."). In further sensitivity analyses as requested by the reviewer, we find accounting for parental reported cognitive ability bins did not attenuate the SNP heritability for age at autism diagnosis, nor the association between later diagnosed autism PGS and age at autism diagnosis. We note that SPARK has only measured IQ in fewer than 5000 individuals. However, among these individuals, we find roughly similar distribution of IQ measured using standardised measures and parent reported IQ bins as provided in response to point 1.2. We ran additional heritability and PGS-based association analyses for other measures relating to IQ and language (language regression, ID, age at first words) as outlined in 1.2a, and observed similarly robust results. Although there may be other fine-grained measures of intellectual ability, we think it's unlikely that they will explain any substantial variance in age at autism diagnosis.

Finally, it is important to note that the mediation by additional variables (either environmental or clinical) does not explain why the genetic correlation between earlier and later diagnosed autism is only moderate, as explained in our response to point 1.1.

1.3b Second, prior research suggests that there are racial disparities in identification/diagnosis of autism. For example, Black (African-American) children in the US are diagnosed less frequently, at later ages, and with apparently increased rates of intellectual disability, suggesting increased diagnostic threshold by some clinicians (Constantino et al (2020). Timing of the diagnosis of autism in African American children. *Pediatrics*, 146(3), e20193629. <https://doi.org/10.1542/peds.2019-3629>; Durkin et al. (2017) Autism spectrum disorder among US children (2002–2010): socioeconomic, racial, and ethnic disparities. *American Journal of Public Health*, 107(11), 1818–1826. <https://doi.org/10.2105/AJPH.2017.304032>; Habayeb et al. (2022). Still Left Behind: Fewer Black School-Aged Youth Receive ASD Diagnoses Compared to White Youth. *J Autism Dev Disord* 52, 2274–2283. <https://doi.org/10.1007/s10803-021-05118-1>. There is insufficient granularity in race/ethnicity data reported in the manuscript, particularly for SPARK and iPSYCH cohorts which provide the polygenic scores data. Ethnicity is measured as 'ethnic minority status' (presumably white vs non-white), which lacks attention to the potential that specific ethnic groups face systemic barriers to access to diagnosis, or that clinical biases

may favour identification of autistic diagnoses earlier vs later in some groups. Compounding these potential challenges is that it was not possible to discern the percentage of participants with 'ethnic minority status' in the genetic cohorts (ie the US SPARK and Denmark iPsych cohorts) despite review of the primary manuscript and supplemental materials (notably, Supplementary Figures but not Supplementary Tables referred to in the Supplement were included). 'Ethnic Minority' is listed in the Excel Datafile only in reference to regression analyses (tab SD3); data for the 'birth cohorts used' (tab SD1) is limited to sex and age of diagnosis for the longitudinal cohorts; no descriptive data is provided for the cross-sectional cohorts. The SPARK website <https://www.sfari.org/spark-demographic-and-clinical-information/> (accessed Sep 12, 2024) indicates that 73.1% of participants are white (this may or may not be reflected in the cohort reported in the current manuscript). Notably, the non-white group are diverse and reflected in multiple race/ethnic categories, including 7.1% African American. It is concerning to attribute age-of-diagnosis to genetic differences without sufficient attention to how race/ethnicity may contribute to disparities in access to appropriate assessments and/or clinical biases. This needs further consideration in the manuscript.

Thank you for raising this point. Again, it is helpful to separate the longitudinal analyses from the genetic analyses.

In the longitudinal birth cohorts, we used the term ethnic minority status as we were limited by sample size, and in Australia and Ireland no specific info was available. Furthermore, by including income and deprivation status, this could partially cover the access to diagnosis/services bit. In these cohorts, ethnic minority status explained between 3% (LSAC-B) to 0.1% of the variance in age at autism diagnosis. We compared this to SPARK, in analyses not conducted using any genetic variable. Here, ethnicity/race explained about 1% of the variance in age at autism diagnosis. We scanned existing literature, and we identified race/ethnicity explained less than 4% of the variance in age at autism diagnosis (**Extended Data Figure 1**, provided in our response to 1.1). So, in sum, our findings from the birth cohort are largely consistent with what we observe in SPARK and what has been previously reported.

For the genetic analyses (SPARK and iPSYCH), we have focussed only on individuals of genetically inferred European ancestries to avoid false positives and minimise population stratification. Population stratification occurs when systematic differences in allele frequencies between subpopulations in a study coincide with phenotypic differences, leading to false-positive associations (Price et al., 2006). Additionally, linkage disequilibrium (LD) patterns and minor allele frequencies vary substantially across ancestral groups, affecting the statistical power to detect true associations (Martin et al., 2017). So all genetic analyses have been done within a genetically inferred ancestral subgroup. In SPARK, where we had access to the data, only 1.00% have indicated non-white race. Please find the Table below.

Category	Count	Percentage
White only*	22969	79.1
White and other race**	2064	7.11

Non-White	290	1.00
No Race indicated	3649	12.57

*Includes those who have identified as “White” and any other race.

**Non-white includes the following races: “Asian”, “Hispanic”, “African American”, “Native Hawaiian”, and “Native American”

We also wish to highlight that the association with race/ethnicity is not consistent across studies as identified by a critical review by Daniels and Mandel⁴. Finally, we wish to reiterate that the aim of our study was not to identify all factors associated with age at autism diagnosis, but rather if there are different developmental trajectories and polygenic profiles underlying earlier and later diagnosed autism. We have made this more clear now throughout the manuscript.

We have mentioned the need to extend the findings to individuals of other genetic ancestry in the Limitations section of the Discussion: “Finally, our genetic analyses focused on genetically inferred European ancestries due to limited GWAS data from other populations, highlighting the critical need for future research to examine the transferability of these findings across diverse genetic ancestries.”

1. 4 Further clarification is also needed regarding how the age of diagnosis was stratified to ‘childhood vs adolescence’. Cut-off was at age 9 vs 11 years, depending on cohort. The authors noted that ‘specific cut off age was cohort-dependent, as different birth cohorts collected information at different ages’. However, Extended Figure 1 suggests some inconsistency in the application of a decision rule to determine the cut-off – LSAC-B and LSAC-K include assessments at both 9 and 11 years, yet a 9-year cut-off was applied to the former and a 11-year cut-off to the latter. Relatedly, not clear why the authors did not apply the cut-off somewhat later (e.g., consistently from 11 years), if their intent was to identify a subgroup diagnosed in adolescence. That said, the authors report SNP heritability for age of diagnosis in the iPsych dataset which indicates a similar pattern whether the cut point is 9 or 11 years (pg 10-11).

Thank you for this query and apologies for the confusion. It is helpful to address this point using **Extended Data Figure 3**. In this figure the box represents all the time points when a report of autism diagnosis was available. So, based on this, in LSAC-K no autism diagnosis was recorded before 11. Whereas in LSAC-B, the earliest autism diagnosis reported was at age 9. The next report of autism diagnosis was age 13. Across all four cohorts, there is an autism diagnosis report at either age 9 (GUI and LSAC-B) or age 11 (LSAC-K and MCS). We thought this was a natural window to specify a cutoff as this coincides with the onset of puberty, and the transition from primary to secondary school. We were unable to choose an earlier cutoff as only MCS (at ages 5, and 7), and GUI (at age 7) have a record of an autism diagnosis before this window. We were also unable to choose a later cutoff as there are no records of an autism diagnosis in MCS after age 14.

Extended Data Figure 3: Schematic diagram of the birth cohorts included in the study

Schematic diagram of the cohorts included in the study and the ages when data was collected for SDQ scores (dots) and autism diagnosis (in boxes). So, reports of autism diagnosis were available at ages: MCS - 5,7,11,14; GUI - 9,13, 17; LSAC-B - 7,9,11,13,15; and LSAC-K: 11,13,15, and 17. MCS = Millennium Cohort Study; GUI = Growing up in Ireland (cohort '98); LSAC-B = Longitudinal Study of Australian Children (Birth cohort); LSAC-K = Longitudinal Study of Australian Children (Kindergarten cohort). Sample sizes and the year of initial SDQ data collection for each cohort are shown on the ordinate axis. The age cutoff used in the Latent Growth Curve Models for each cohort is indicated by a red line.

Please note, in response to reviewer #4's comments about the readability of the manuscript, we have now moved the LGCM model into the **Supplementary Note 4**. However, to further address this point we have:

1. Ran sensitivity analyses using a range of different cutoffs and demonstrated that the slope of the curve increases and the intercept decreases with a later age at diagnosis in all cohorts. This addition was clearly addressed in the Methods.

“To address this limitation and better capture the relationship between diagnostic timing and socioemotional outcomes, we conducted additional LGCMs for autistic children using stepwise groupings by age at diagnosis, in the most abundant autistic cohort, MCS.” Additional information is available in **Supplementary Note 4** and an extra **Supplementary Figure 8**.

2. Explained the choice of cutoff more clearly in the Methods.

“Across the cohorts, the earliest autism diagnoses were reported at ages 9 (GUI, LSAC-B) or 11 (LSAC-K, MCS). We chose this 9-11 age window as our cutoff since it aligns with puberty onset and the primary-to-secondary school transition, because there is epidemiological evidence showing increased autism incidence among females during this window⁵ and because trajectory analyses have identified increasing autism-related traits in a subset of individuals after this period⁶. An earlier cutoff was not feasible as only MCS

(ages 5, 7) and GUI (age 7) recorded autism diagnoses before this window. A later cutoff was not possible due to no autism diagnoses in MCS after age 14. To further capture the relationship between diagnostic timing and socioemotional outcomes, we conducted additional LGCMs for autistic children using stepwise groupings by age at diagnosis in MCS and LSAC-B (see **Supplementary Note 4** and **Supplementary Figure 10**)."

3. Included a line in **Extended Data Figure 3** that clearly explains when reports of an autism diagnosis were available.

"So, reports of autism diagnosis were available at ages: MCS - 5,7,11,14; GUI - 9,13, 17; LSAC-B - 7,9,11,13,15; and LSAC-K: 11,13,15, and 17."

Originality and significance:

1.5 Perhaps because the primary focus was on the genetic analyses, the finding of distinctive socio-behavioural trajectories received less attention in the Discussion. Distinct trajectories of co-occurring internal/externalizing behavioural features have been reported extensively in longitudinal autistic cohorts (see recent review by Gentles et al (2024). Trajectory research in children with an autism diagnosis: A scoping review. *Autism*. 2024 Mar;28(3):540-564. doi: 10.1177/136236132311702800. The originality and significance of the current findings could be discussed further in that context. Overall, the findings of the manuscript will be of broad interest to clinicians and researchers, with the caveat of concerns regarding the presentation and interpretation of polygenic scores in relation to age of diagnosis, as discussed earlier.

Thank you for raising this. Of the studies cited in the review, the 55 studies belong to two broad categories that only one study has investigated the trajectories of social, behavioural, emotional and other related autistic traits prospectively, prior to an autism diagnosis⁷.

All other studies have investigated the longitudinal trajectories in individuals with familial risk or diagnosis of autism at baseline, or investigated trajectories of traits related to autism in the general population. This prevents us from asking if changes in developmental trajectories are indeed associated with age at autism diagnosis.

These 55 studies belong to one of two broad categories:

1. Sibling studies that have carefully characterised younger siblings of autistic individuals^{8,9,10,11}. This design is different from ours in three ways. First, we focus on children in birth cohorts who may not have a family history of autism. Second, we fit longitudinal trajectory models whilst all sibling studies primarily compare scores between two timepoints. Third, most published sibling studies have followed up infants and toddlers, only up until early childhood^{8,9,12}.
2. Population based studies. To our knowledge, there are two studies that have investigated trajectories of social, emotional and behavioural differences among autistic individuals. Mandy and colleagues (2022) used the Millennium Cohort Study to fit a Growth Curve Model on SDQ scores for autistic children diagnosed early (7 or younger) or later (8 -11). Second, May and colleagues used data from LSAC to understand diagnostic stability of autism, and compared SDQ trajectories using Generalized Estimating Equations among those with late, persisting, and desisting

autism diagnosis. This study was primarily focussed on understanding diagnostic stability and did not use Latent Growth Curve Models/Growth Mixture Models to assess SDQ trajectories based on age of diagnosis. Furthermore, neither of these studies directly compared their findings across countries, and did not further investigate if these differences in socio-emotional behaviours are partly due to differing genetics.

Simply put, all other studies have investigated the longitudinal trajectories in individuals with familial risk or diagnosis of autism at baseline, or investigated trajectories of traits related to autism in the general population. This prevents us from asking if changes in developmental trajectories are indeed associated with age at autism diagnosis. Taken together, ours is the first study to systematically investigate if changes in social, emotional behavioural difficulties are associated with age at autism diagnosis. We also assess the variance explained in age at autism diagnosis by these trajectories, and if these trajectories mediate the effects of socio-demographic variables on age at autism diagnosis. We are also the first study to investigate the genetic correlates of age at autism diagnosis and demonstrate that earlier (typically in early childhood) and later diagnosed (late childhood and afterwards) autistic individuals have differing common genetic variant profiles.

Changes to the manuscript: We have now discussed the findings from the trajectory analyses and contextualised them in greater detail.

“Although developmental variation is well-characterised in autism¹³, to our knowledge no study has robustly investigated if this variation is associated with age at autism diagnosis in prospective cohorts. Our trajectory analyses across multiple birth cohorts converge with the genetic findings to suggest different developmental pathways to autism diagnosis. The identification of early and later emergent latent classes aligns with the differential effects of age-stratified polygenic scores. Despite the relatively modest sample sizes of the cohorts analysed, we find robust and converging findings across multiple cohorts and different analytical methods. These convergent evidence from both phenotypic trajectories and genetic analyses suggest that the timing of autism diagnosis may partly reflect genuine aetiological differences in developmental pathways, rather than purely environmental or diagnostic factors.”

Data & methodology:

1.6 The manuscript has many strengths with respect to available data and the extensive analyses reported. Concerns regarding the analyses are covered in the earlier comments. The Figures are excellent overall, but error bars are not consistently defined.

Apologies for this. All error bars represent 95% confidence intervals. We will carefully define this in all figures.

Conclusions:

1.7 Concerns regarding conclusions (especially related to the genetic findings) were discussed earlier.

We hope our response addresses these concerns.

Suggested improvements:

1.8 With regards to concerns regarding limitations in the data, I suspect that more detailed data on child-level clinical factors that might influence age of diagnosis are not available. More granular data on race/ethnicity are available for at least the SPARK cohort that could be incorporated into the age-of-diagnoses analyses. While I realize that an evaluation of the nuanced multi-layered factors that may influence age-of-diagnosis is beyond the scope of the current manuscript, more explicit acknowledgement of the limitations in the data and the interpretation of the polygenic analyses could strengthen the papers.

Thank you for this. We want to first clarify that our aim is not to understand all factors associated with age at diagnosis. Rather, what our primary findings demonstrate is that there are at least two modestly correlated polygenic factors underlying autism. These factors have an impact on the developmental trajectories underlying autism, which is associated with when someone is diagnosed as autistic. However, we have now included additional clinical variables in SPARK and tested their association with age at autism diagnosis, as highlighted in our response to the previous comments.

References:

See earlier comments

Clarity and context:

1.9 The Abstract is clear and accessible, but revisions should be aligned with revisions related to earlier comments; likewise, the introduction and conclusions, both in terms of the relevance of the socio-behavioural trajectory findings independent of genetic modelling, and the complexity of factors that may influence age of diagnosis, including those not sufficiently covered in the available datasets/study measurement model.

Thank you. We have updated the abstract in line with the revisions.

Inflammatory material:

Although the language within the manuscript is respectful and appropriate, the notion of age-of-diagnosis being genetically determined may be offensive to some (and this is also relevant to Springer Nature's commitment to EDI).

Thank you for raising this. We have taken substantial care to communicate our findings by also including an FAQ and summary aimed at the general public. We have worked with members of the autistic community to obtain feedback on this work and also review the FAQs and summary. Please note, that we have now rewritten portions of the manuscript such that the focus is not on the message that age at diagnosis is heritable, but that there are differences in polygenic architecture of autism based on age at diagnosis. This is also reflected in the new title of the manuscript.

Referee #2 (Remarks to the Author):

This is an impressive and potentially important paper in the autism field. It utilizes data from multiple longitudinal population cohorts on age of autism diagnosis and mental health traits/symptoms (assessed using the SDQ) and then genetic data from two large cohorts iPSYCH and SPARK (and PGC and FinGEN). It is cleverly and competently done and very ambitious and also broad in scope with potential impact on both the science and practice of autism. My expertise relates to clinical phenotypic data, and I will restrict most of my comments to this topic.

Main points:

2.1 The concept and findings are potentially important to help explain the variability in developmental trajectories of individuals who have autism. In some ways the findings from the genetic association shown in Figure 5 corroborate what we know from clinical studies – That individuals with later diagnosed autism are more likely to have additional co occurring mental health difficulties including ADHD, anorexia, depression and PTSD with associated phenomena such as self-harm and childhood maltreatment. One could alternatively characterize the two autism PGS factors not as early versus later diagnosed autism but as autism with or without the presence of these mental health conditions (at least in adolescences when some at least are more routinely identified by CAMHS services). It is the case in the population cohorts that we know that in many communities, including those from which these longitudinal population cohorts are drawn, autism is under-recognized and often only comes to the attention of carers and clinical services because of early to later adolescent onset of emotional and behavioral difficulties. This may be an equally important way of parsing or characterizing heterogeneity in the developmental presentation of autism (in terms of etiological understanding, developmental pathways and mechanisms, and potential indicators for appropriate targeted intervention support at different ages - which is not mentioned in the manuscript), but it would be perhaps less revolutionary than the idea that the authors put forward.

Thank you for raising this important point. In responding to this alternate hypothesis, it's helpful to separate genetic effects from observed clinical effects.

Genetic findings

Firstly, one framework to characterise the genetics of later diagnosed autism is that it is autism + mental health conditions. In other words, later diagnosed autism would reflect the additive genetic effects of autism (indexed by earlier diagnosed autism) and a series of other mental health conditions.

Figure 2.1 illustrates three potential models of autism's genetic architecture.

Model 1 (analogous to the first glass filled entirely with green beads) represents autism as emerging from a single, uniform polygenic liability. In this model, earlier and later diagnosed individuals share the same fundamental genetic basis, with later diagnoses potentially resulting from having fewer autism-associated genetic variants (i.e., less "severe") or from non-genetic factors such as limited access to diagnostic services.

Model 2 (similar to the second glass, half green beads and half mixed colors) suggests that autism's genetic architecture comprises two distinct components: a core autism polygenic component plus various other mental health-related polygenic factors. Under this model, later diagnosed individuals might either have mental health conditions misdiagnosed as autistic or a combination of lower autism polygenic liability with higher genetic predisposition to other mental health conditions to necessitate a clinical diagnosis.

Model 3 (represented by the third glass with its three distinct layers) proposes a more complex structure: two related but distinct genetic architectures for earlier and later diagnosed autism (shown by the green and purple beads), along with additional genetic factors related to other mental health conditions (the mixed-color beads). Unlike Model 2, this model suggests that autism's genetic architecture cannot be fully explained by combining early autism genetics with other mental health conditions.

Figure 2.1: Illustrative example of the three models

In other words, is the polygenic architecture of later diagnosed autism merely the polygenic effect of autism + polygenic effects of other mental health conditions misclassified as autism? This framework is exactly what we have tested and reported in **Supplementary Note Section 3.2** in the original manuscript. We have now taken the opportunity to spell it out as a separate note (**Supplementary Note 6**) and further extend this using the expanded age-at-diagnosis stratified GWAS from SPARK. Please see below the findings.

In the first model we use the iPSYCH autism GWAS (which consists of both earlier and later diagnosed autism), we sequentially subtract the genetic effects of earlier diagnosed autism (using the PGC autism GWAS in Model1 or SPARK_before6 GWAS in Model2) and other

mental health conditions (ADHD, schizophrenia, PTSD, anorexia nervosa, and bipolar disorder) using genomicSEM. In both models with SPARK_before6 and PGC, we find that only the genetic signal for earlier diagnosed and ADHD explain the variance in iPSYCH autism GWAS, but there is significant residual genetic variance. This suggests that the genetic architecture of autism captured in the iPSYCH cohort contains components that are distinct from both early-diagnosed autism and other psychiatric conditions.

In line with our findings, we hypothesise that this residual genetic variance likely represents unique genetic factors associated with later-diagnosed autism presentations. We test this by including SPARK_after10 autism GWAS in the models as a measure of later diagnosed autism, and PGC (Model 3) or SPARK_before6 (Model 4) as a measure of earlier diagnosed autism. In both models, we find that SPARK_after10 significantly explains the residual variance in the iPSYCH autism GWAS ($p = 0.0193$ and $p = 0.0365$ for PGC and SPARK_before6 models respectively). This confirms our hypothesis that the previously identified residual genetic variance is indeed associated with later-diagnosed autism presentations. Notably, in both models, ADHD genetic effects now fail to reach statistical significance ($p = 0.620$ and $p = 0.939$) after including SPARK_after10, suggesting that the genetic architecture of later-diagnosed autism is distinct from ADHD. The genetic effects of other psychiatric conditions also remain non-significant. These findings provide strong evidence for distinct genetic architectures between early and later-diagnosed autism, and that later autism diagnosis is not merely autism + the genetic effects of other mental health conditions.

Supplementary Note 6, Figure 2: Path diagrams representing results from genomic multiple regression analyses using GenomicSEM. The genetic effects of the iPSYCH autism GWAS were regressed on the genetic effects of the PGC-2017 autism GWAS (Models 1 and 3) or SPARK before 6 autism GWAS (Models 2 and 4), attention-deficit/hyperactivity disorder (ADHD), major depressive disorder (MDD), posttraumatic stress disorder (PTSD), schizophrenia (SCZ), anorexia nervosa (Anorexia), and bipolar disorder (Bipolar) simultaneously. In Models 3 and 4, we additionally included SPARK after 10 GWAS. Single-headed arrows indicate conditional

genetic associations between the explanatory variables and the iPSYCH autism GWAS. Numbers represent standardised correlation coefficients, with standard errors in parentheses. Genetic associations between explanatory factors were accounted for in the analyses but are not shown on the graph for simplicity. The two-headed arrows connecting the genetic component of the iPSYCH autism GWAS to itself represent the residual genetic variance unexplained by the genetic influence of any of the GWAS included in the black circles. Solid lines indicate significant genetic associations, while dashed lines indicate non-significant associations. For simplicity, only the significant estimates and standard errors are shown, with the exception of the residual variance (and standard errors) in Models 3 and 4.

Secondly, in separate work from our group, we investigated if autistic individuals with and without co-occurring mental health conditions are genetically different by assessing the heritability of these co-occurring health conditions. This is described in detail in a preprint available here: <https://www.medrxiv.org/content/10.1101/2024.11.11.24317097v1> and the key figure is provided in **Table 2.1** below.

Table 2.1: SNP-based heritability (liability scale) for autism with co-occurring conditions.

PHENOTYPE	N cases	N controls	h2SNP (General Population; reference)	h2SNP (SE): GREML-SC	h2SNP (SE): LDSC
ADHD	7409	10173	0.222 (Demontis et al, 2019)	0.177 (0.04)	0.139 (0.04)
DISRUPTIVE BEHAVIOUR DISORDERS	1741	15841	Unavailable	0.519 (0.09)	0.421(0.09)
DEPRESSION	2539	15043	0.057-0.167 (Als et al, 2023)	0.065 (0.08)	NA
ANXIETY DISORDERS	5751	11831	0.26 (Purves et al, 2019)	0.068 (0.04)	0.048 (0.04)
SCHIZOPHRENIA	253	17329	0.24 (Trubetskoy et al, 2022)	0.173 (0.51)	0.276 (0.32)
BIPOLAR DISORDER	661	16291	0.186 (Mullins et al, 2021)	0.614 (0.24)	0.07 (0.14)

Briefly, we find that, with the exception of ADHD and Disruptive Behavioural Disorders, other co-occurring mental health conditions do not have a significant SNP-based heritability in a sample of up to 17,582 autistic individuals (**Figure 2.1.2**). Second, neither PGS analyses nor genetic correlation studies identify shared genetics between autism and co-occurring mental health phenotypes specifically among autistic individuals. This suggests that stratifying autistic individuals by the presence of anxiety, depression, schizophrenia, and bipolar disorder do not provide meaningful axes of characterising genetic heterogeneity in autism.

Taken together, these analyses indicate that the genetics of later diagnosed autism cannot be characterised as autism and other mental health conditions. Furthermore, only some co-occurring mental health conditions are heritable among autistic individuals from the SPARK cohort.

Epidemiological and qualitative findings

We next turn to the epidemiological findings. We agree with the reviewer that later diagnosed autistic individuals have elevated mental health conditions compared to earlier diagnosed autistic individuals. However, compared to non-autistic individuals, mental health conditions are elevated even among individuals diagnosed early (e.g., before age 6)^{14,15}, suggesting that age at diagnosis and the presence of mental health are not equivalent axes.

Importantly, there is a rich body of literature that frames mental health conditions as a consequence of later diagnosis rather than a cause. Later diagnosis is associated with lack of important early support, increased adverse life experience (e.g., bullying, low self-esteem, peer rejection) leading to distress and mental health conditions¹⁶⁻¹⁸.

Taken together, although we agree with the reviewer that parsing heterogeneity through the lens of co-occurring mental health conditions is important (and we have done just that in our preprint), we do not agree that this framing is interchangeable with our current framing from genetic, epidemiological, and qualitative perspectives.

2.2 In the 4 population cohorts (MCS, GUI, LSAC-B, LSAC-K) as shown in Extended Figure 1 age of diagnosis data is not captured (and from Supplementary Table 1 would be rare in these cohorts) in the preschool years. This contrasts to the very opening sentence of the paper referring to autism not necessarily being manifest by age 3 years and the age of autism diagnosis in the volunteer US SPARK cohort shown in Extended Figure 2 (where the mean is ~ 5 years).

Thank you for this point. First, we agree that the opening sentence contrasts with what we can test in these cohorts. Our aim was to highlight that in sibling cohorts where there is detailed, longitudinal assessment for autism, there are some children who do not initially meet the criteria for autism but later do. In other words, this is due to the developmentally variable emergence of autism features rather than later community diagnosis due to various other factors. That said, we have now rewritten the Introduction to more clearly articulate that we are testing the idea that autism is developmentally heterogeneous.

We next tested this in MCS. In sensitivity analyses, we ran LGCMs with stepwise age cutoffs, including the before (N = 27) and after (N = 161) age 5 grouping to reflect the transition into school, to assess differences in slopes and intercepts by age at diagnosis across SDQ total and subscales (see **Supplementary Note 4** and **Supplementary Figure 10**, also below). This approach extended our original dichotomous classification of early versus late diagnosis, allowing for a more granular analysis of developmental patterns across subgroups. The results consistently showed that groups including children diagnosed at later ages had lower intercepts across SDQ subscales, indicating fewer initial difficulties, but exhibited higher slopes, reflecting a steeper increase in difficulties over time. Specifically, significant differences were shown in slopes and intercepts for SDQ total scores and every subscale

score (**Supplementary Figure 10**, also below). For consistency, we conduct similar analyses with LSAC-B. Notably, estimation issues, including negative latent variances or a non-positive definite latent covariance matrix, emerged in models with smaller group sizes (dashed lines in the figure below), further supporting our choice of a dichotomous age cut for diagnosis grouping. Also, the confidence interval for LSAC-B was notably wide, despite the overall trend remaining consistent. We did not conduct a similar analysis in LSAC-K as the earliest age at diagnosis is age 11.

Furthermore, as mentioned in the text, we believe that any age at diagnosis related threshold is inherently arbitrary, which is why we have supplemented the analyses with Growth Mixture Models, which identify latent groups based on trajectories. We do not believe the limited number of participants with data before age 5 unduly affects our primary conclusions that the polygenic architecture of autism varies substantially with age at autism diagnosis.

LGCM estimates across SDQ total and subscales by stepwise age at autism diagnosis grouping in MCS - main: Intercept (top) and slope (bottom) estimates (with 95% confidence interval bars) in LGCM for SDQ total and subscale scores in autistic children grouped by age at diagnosis (and no autism diagnosis status) in MCS - main.

LGCM estimates across SDQ total and subscales by stepwise age at autism diagnosis grouping: Intercept (top) and slope (bottom) estimates (with 95% confidence interval bars) in LGCM for SDQ total and subscale scores in autistic children grouped by age at diagnosis (and no autism diagnosis status) in LSAC-B. Dashed lines flag estimates with negative latent variances or non-positive definite latent covariance matrices, likely due to small sample sizes.

2.3 This highlights a perhaps critical limitation of bringing the population cohorts and genetic cohorts together in that ‘early diagnosis’ means different things in different cohorts (e.g. different in iPSYCH than SPARK in Extended Figure 2 and thus more like the population cohorts in Extended Figure 1 and Supplementary Table 1). The authors are correct that any age threshold of ‘early’ vs. ‘later’ diagnosis is arbitrary but their analysis does not tell us about the preschool vs. later (mid-childhood onwards) age of diagnosis that has been more clearly realized in the clinical field (as they recognize in References #7 to #14). This point compliments my alternative characterization of the genetic factors they identify as being about (in one view) autism with/without mental health difficulties rather than early versus later diagnosis.

Thank you for raising this point. Although sibling studies have typically focussed on preschool and mid/late childhood years, we note that there is no clear consensus within the literature as to what late diagnosis refers to, with studies having cutoffs ranging from 2 - 55 as detailed in this recent review of the problem.¹⁹ Indeed, in one particularly illuminating paragraph (quoted below), this study demonstrated that different research groups studying different aspects of autism defined late diagnosis differently. We quote the relevant section

“For research groups working with infants, a “late” diagnosis was often described as anything later than the first-identifiable behavioral diagnosis at 18–24 months, with late thresholds often around 2–3 years (Cox et al., 1999; Mishaal et al., 2014; Nitzan et al., 2023). For children, more cutoffs emerged around school-age entry, such as 5–6 years (Berg et al., 2018; Johnson-Taylor, 1987; J onsd ottir et al., 2011; Santos et al., 2017). In adolescence, many thresholds set by researchers centered around access to individualized education programming and aging out of those resources, which centered around 12–13 years of age (Harrop et al., 2024). As research participants aged, so did the consideration of a “late” diagnosis, with adult studies frequently reporting an adult diagnosis (at 18 years) as the cutoff for a late diagnosis (Fombonne et al., 2022; Frank et al., 2018; Ghanouni & Seaker, 2023).”

We were aware of the lack of consensus in the field and consequently, used the terms “earlier” and “later” to underscore the relative nature of age at autism diagnosis. The authors of the previously cited systematic review further suggest that we contextualise the conceptualisation of late diagnosis, which is precisely what we have done in the text.

For the birth cohorts, we note: “We chose this 9-11 age window as our cutoff since it aligns with puberty onset and the primary-to-secondary school transition, there is epidemiological evidence showing increased autism incidence among females during this window⁵ and because trajectory analyses have identified increasing autism-related traits in a subset of individuals after this period ⁶.”

For the iPSYCH stratification we note: “We chose this age cutoff to divide the iPSYCH cohort into two subgroups with similar sample sizes and because age coincided with the window in which we observe an increase in SDQ scores in the birth cohorts, and which is associated with an increase in diagnosis of females in epidemiological samples⁵. “

We further note that the main aim of our study was not to necessarily investigate differences between preschool vs early childhood. This is because all our diagnosis is based on community diagnosis, and as such, there can be variable delay in age at diagnosis after the first signs have been noticed. This means that there need not be a clear, discrete cutoff point during which there is a shift in the developmental trajectory and polygenic architecture. This is evidenced by Figures and new analyses provided in response to query 2.2 - there is a clear gradient in the slope and intercept of the SDQ total and subscale scores in MCS and LSAC-B. Whilst we agree that it would be nice to have a clear atlas of developmental and polygenic differences by different age at onset, this is impossible in cohorts with community diagnoses as there will be several systematic and idiosyncratic differences between a clear(er) age at onset and age at diagnosis. Indeed, a strength of this study is that we are finding quite clear differences based on age at diagnosis despite this obvious limitation underlying community diagnosis. Our findings can inform future research in longitudinal sibling cohorts where age at onset is more clearly measured, such as in the BASIS cohort.

We also do not think that earlier vs later diagnosis mean different things in different cohorts. In both iPSYCH and the birth cohorts, we have stratified autistic individuals based on a diagnosis of autism at age 10 (iPSYCH-based genetic analyses) or in the 9 - 11 age window (LGCM models in the birth cohorts). In addition, recognising the arbitrary nature of this cutoff we have conducted analyses using data-driven approaches - Growth Mixture Models in the birth cohort and genomicSEM in the genetic analyses.

Given the lack of consensus of what threshold constitutes late vs early diagnosis, the robust results using different age thresholds and methods, and a similar threshold adopted for the genetics and longitudinal cohorts, we do not think this is a 'critical limitation' - our results are robust to these considerations.

We have further included the following lines in the Discussion to reflect this discussion in the Manuscript:

"Fourth, we use 'earlier' and 'later' diagnosed autism as relative terms, reflecting that developmental and polygenic differences represent a gradient rather than discrete categories, particularly given the lack of consensus on age thresholds for early versus late diagnosis¹⁹. Fifth, since the autism diagnoses in this study rely on community-based caregiver or self-reporting rather than standardised clinical assessments, there may be varying delays between when autism features first appear and when a formal diagnosis is received. However, our findings can guide future research using longitudinal cohorts, particularly sibling studies, that systematically track the emergence of autism features over time."

2.4 Furthermore, what is missing in these analyses (of both the population and the genetic cohorts) is any measure of autism severity, aside from categorical presence or absence of diagnosis. There is some evidence at least from the ALSPAC study that it is not consistently the case across developmental time that those who receive an earlier diagnosis have the

most severe autistic symptoms. However, in other clinical cohorts they do. It is something that is missing from consideration in the current manuscript. I partly know what measures on autism severity the different cohorts have (I think all have some) and this should be considered by the authors.

Thank you for raising this point. None of the birth cohorts have any measure of severity of autistic traits/autism as these are not autism specific cohorts. However, the SDQ has moderate correlations with many measures of autistic traits including SRS^{20,21}, SCDC²², SCQ²³, and other autism trait measures²⁴. So it is likely that aspects of autism severity among these autistic children will be correlated with their scores on the SDQ.

With regard to genetics, in our previous work, we did investigate if stratifying autism by severity using parent reported SCQ and RBS-R scores modifies the SNP heritability²⁵. We find that stratifying by severity does not improve SNP heritability, and therefore has no impact on genetic heterogeneity.

We tested if accounting for severity modifies the results in two ways. Please see our response to Reviewer 1.2 for further details. We provide brief responses below.

First, in the SPARK cohort we investigated if the association between age at autism diagnosis and PGS for earlier and later diagnosed autism statistically differ after accounting for total SCQ score, total RBS-R scores, and parental-reported cognitive test scores. We do not find any statistical difference even after accounting for this.

Second we investigated if the SNP-based heritability for age at autism diagnosis attenuates after accounting for these measures. Yet again, we do not observe any significant attenuation.

This can be explained by a theoretical model by using a polygenic liability threshold model. Under an infinitesimal polygenic liability model, we would expect the genetic correlation between extreme ends and middle ranges of the phenotypic distribution to be reasonably high and positive. This occurs because it is the same alleles that contribute across the phenotypic distribution.

So from that perspective, if one were to assume that severity of autism and an autism diagnosis (regardless of age) emerge from the same underlying polygenic liability distribution, then you would expect that accounting for severity would attenuate any genetic signal for age at autism diagnosis. Furthermore, the genetic correlation between earlier and later diagnosed autism will be high if this is primarily explained by severity (Model 1).

However, our key finding is that there are two different, although correlated, polygenic distributions underlying autism, which vary by age at diagnosis. In other words, the genetics of earlier and later diagnosed autism are not merely attributable to differences in severity but rather to different underlying but correlated polygenic distributions.

2.5 Relatedly, I was surprised by the lack of effect (as far as I understand it) of IQ and sex in the early vs. later diagnosis differences. One relevant point (a limitation not a flaw) is the

modest size of the population cohorts with a diagnosis and from Supplementary Table 1 the sex ratios at different ages are limited by low sample sizes – and I do think the reader needs to be alerted to this point overall as they might not look-see it in this Table – OK it is on p.4 but should be noted in the Limitations).

Thank you for raising this. Although, on average, females are diagnosed later than males, there is substantial variation even within sex, suggesting that variance within the group is quite large. There is a small body of literature which identifies factors contributing to age at autism diagnosis. We conducted a review of the literature and summarised the findings. Please see **Extended Data Figure 2**, and our response to reviewer 1 (1.1) where we show that IQ/language regressions explain typically less than 15% and sex explains typically less than 6% of the variance in age at autism diagnosis. One previous systematic review has identified that 13 out of 17 studies have found no association between sex and age at autism diagnosis. The same study also highlighted the mixed association between intellectual disability and age at autism diagnosis⁴. In sum, there is variation in the association between these factors and age at autism diagnosis, and the variance explained is not particularly large, especially for sex.

Furthermore, we investigated this in SPARK in a subset of 5,773 autistic individuals with complete information on several clinical and demographic variables, sex explained about 1.5% of the variance and language and ID related variables explained about 11% of the variance in age at autism diagnosis. Together, the variables we investigated explained about 23.27% of the variance explained. In comparison, the SNP-based heritability for age at autism diagnosis is about 11% and is independent of these variables.

In the longitudinal population based cohorts, sex explains 1 - 2% of the variance in age at diagnosis. This seems largely similar to SPARK and other studies available. So, although, on average, females are diagnosed later than males, there is substantial variation within each sex, that sex per se explains a small fraction of the variance in age at autism diagnosis. Please see **Table 1.1** in our response to Reviewer 1 (1.1) for additional details on the variance explained.

With ID, we agree that this primarily reflects ascertainment bias. Longitudinal population cohorts are likely to attract and retain individuals who are, on average, healthier. Subsequently, there are likely to be fewer individuals with autism with complex support needs in these cohorts compared to cohorts ascertained for autism. Ideally, a well-phenotyped epidemiological sample is needed to assess the true proportion of variance explained by ID and language delays.

We do make note about the relatively small size of the birth cohorts, and have now clearly mentioned this across the Discussion. We have added this line: “Although the sample sizes of participants included in the trajectory analyses are relatively small, these findings are robust across multiple cohorts and different analytical methods.”

We have also clearly stated the sample sizes in the text now in the result section.

“We first investigated the association between variable developmental trajectories and age at autism diagnosis using four birth cohorts (N = 89 to 188).”

“In MCS, these results were largely consistent: (1) in an expanded sample after including individuals with co-occurring ADHD and inconsistent autism diagnoses (N = 238, **Supplementary Table 3**); (2) after imputation (N = 623, **Supplementary Table 3, Supplementary Note 2 and 3**); and (3) when restricting to only males (N = 136, **Supplementary Table 3**).”

“In contrast, although we also obtained two latent classes based on SDQ and subscale trajectories among individuals with a diagnosis of ADHD but not autism (N = 89), these latent classes were not significantly associated with age at ADHD diagnosis (**Supplementary Table 6, Supplementary Figures 7 and 8**). We still obtained two latent classes for SDQ total scores and most subscales in an imputed sample of children with only ADHD but no autism (N = 325).”

2.6 On this point I was not sure why cell sizes are given as <10 when there must be a number?

In several of these cohorts, we are restricted (by the data providers) from publishing numbers fewer than 10 as this may aid identifiability.

More minor points as I read through the manuscript:

2.7. p.7 – Can we see CIs for sex ratios in the cohorts since male:female ratio varies from 3.92 to 2.11 which is substantial (if not significant). Also, the $p > .05$ is less informative than the actual non-significant p-value – please replace.

Yes, we have now provided odds ratio and CI for samples with non-masking values (i.e., over 10 children in each category) in **Supplementary Table 3b**. We also reported ORs instead of count ratios in main text:

“male: female odds ratio = 1.08 - 1.50; $P > 0.05$, chi-square tests, **Supplementary Table 3**”

2.8. p.7 and throughout – I was not clear even with some checking again what the ‘cognitive aptitude’ scores was in the different cohorts – primarily a direct IQ / cognitive assessment or something else?

Thank you. In MCS, we used a Principal Component of multiple measures as a measure of cognitive aptitude. We have now provided this in the Methods as well as **Supplementary Table 87**.

“In MCS, we identified multiple variables linked to cognitive ability, SES, and area deprivation. Similarly, in LSAC-B, we identified multiple measures linked to SES, although there were no measures with sufficient sample size linked to cognitive ability or area deprivation. Subsequently, we conducted principal component analysis (PCA) for cognitive abilities, SES, and area deprivation in MCS, and for SES in LSAC-B. PCA was performed using a wide range of measures collected across sweeps (**Supplementary Table 27**), with one variable excluded from any pair showing a correlation coefficient greater than 0.70 to

address multicollinearity. The first principal component (PC1) explained more than 40% of the variance for each corresponding factor, while subsequent components contributed substantially less, supporting the use of the respective PC1 as the summary measure for cognitive ability, SES, and area deprivation (**Supplementary Table 27**)”.

We did not use cognitive aptitude measures in other birth cohorts due to the low sample size.

In SPARK, we used caregiver-reported reports of cognitive impairment i.e., intellectual disability. In sensitivity analyses of the SNP-based heritability and PGS association, we also used parental reported measures of IQ score. We mention this in the Methods.

“We focussed on the variance explained by sociodemographic factors (sex, reported race, household income, mother’s education, and father’s education), cognitive and developmental factors (reported IQ score, reported intellectual disability, age at walking independently, age at first words, language regression, other regression), and autism severity (scores on the Social Communication Questionnaire and Repetitive Behaviour Scale-Revised).”

In iPSYCH ID was based on Electronic Health Records.

2.9. p.9, line 17 – Females more likely to be diagnosed with autism later – in other cohorts yes but was this the case in the 4 population cohorts and main 2 genetic cohorts. Later, the authors also say that mental health problems also higher in autistic females vs. males (p.20, line 24) but the Ref #64 is not a strong one and there is good SR/MA data on this. This relates to my comment above that I do not fully follow (this may be my own limitation) quite how the key clinical factors that might affect age of recognition of both autism and mental health problems – IQ and sex – do not seem to be involved in the early vs. late distinction here – is this true both for the population and genetic cohorts and how does this sit with the broader clinical literature (that does not directly consider genetic information)?

Thank you for this. We have now provided two references for this study now:

1. <https://pubmed.ncbi.nlm.nih.gov/34228813/> - which is from Denmark and uses the Danish registries
2. <https://pubmed.ncbi.nlm.nih.gov/36287538/> - a similar study from Sweden, using the Swedish registries.

Apologies, but we are unable to find a SR/MA that explicitly provides information about sex differences in mental health conditions among autistic individuals.

We hope our response to your previous query (2.5) addresses the issue of sex and IQ. We hope our response to reviewer comment 1.1 addresses the comment about how this is not picked up by genetic factors.

2.10. On pages 10 and 11 the authors present and test different thresholds for early vs. later diagnosis is in iPSYCH but relating to my point about 10/11 years being a late threshold in some ways for late diagnosis and looking again at Extended Figure 2 for age of diagnosis in

iPSYCH and SPARK I wondered why they did not also explore at least in SPARK those diagnosed before versus after 5 years (they cannot do this in iPSYCH probably due to the low numbers). It might inform the point I make above.

In our manuscript, we primarily focussed on the 9 - 11 window for three reasons as mentioned in the Methods. First, prior data from epidemiological literature and analyses of trajectories had identified this window as one that saw both an increase in the incidence of autism diagnosis among girls, as well as an increase in socio-emotional behaviours in the general population. This also coincided with the transition from primary to secondary school transition period and the onset of puberty. This information, along with the availability of data in the birth cohorts informed our choice of cut-off threshold. This threshold was also convenient for genetic analyses as approximately half of the iPSYCH population were diagnosed after age 10 and half before.

We did not focus on before age 5 and after, although this is an age at diagnosis threshold identified in some literature, because we did not have sufficient data in most birth cohorts, and, there were insufficient number of participants in the iPSYCH cohort who were diagnosed before age 5 to conduct a GWAS.

We have now conducted a GWAS of autism diagnosed after the age 6 (SPARK_after6) and calculated the genetic correlation with the existing GWAS of those before the age 6 (SPARK_before6) in the Discovery cohort. The genetic correlation is 0.36 (0.20), which is similarly modest in line with the genetic correlation between the earlier and later diagnosed autism GWAS. Nevertheless, this genetic correlation is higher than the genetic correlation between SPARK_before6 and SPARK_after10 (0.02, s.e.= 0.13), underscoring our previous point that this is a gradient, especially given we only have access to community based diagnosis (and so, there may be some participants who had onset of features before age 6 but would have been diagnosed only later). Given these limitations, we choose not to focus on using age 6 as a cut-off window. We further choose not to report this specific genetic correlation finding in the main text to make the manuscript easier to read based on feedback from Reviewer 4.

2.11. Can the authors comment (for me) more clearly on how formal ID (intellectual disability) is or is not represented in the population and genetic cohorts. It is one of the key diagnostic overshadowing considerations (in addition to ADHD and perhaps oppositionality and anxiety disorders) and there is a mention in the SM p.13 that 'in the birth cohorts, none of the autistic participants has ID' which cannot in fact be true but might represent the data available (ID still is present in between 25% and 33% of children with autism in population samples).

Thank you. We have clarified this in the Methods section. In the epidemiological cohorts used in this study, intellectual disability (ID) was assessed based on the first principal component (PC1) derived from cognitive ability measures.

For the Millennium Cohort Study (MCS), we used a comprehensive set of cognitive aptitude measures collected between the ages of 3 and 11, as detailed in **Supplementary Table 27a**. We excluded one variable from any pair with a correlation coefficient greater than 0.70 to address multicollinearity before performing principal component analysis (PCA). The results

of PCA indicated that PC1 accounted for a substantial proportion of the variance (40%), with a notable drop in the variance explained by subsequent components. This supports the decision to use only the PC1 score as the summary measure of cognitive ability²⁶.

Consistent with prior studies on the MCS, intellectual disability was defined as scoring two standard deviations below the mean on the summary measure (PC1)^{27,28}. In this study, no autistic children in the MCS cohort who had measures of cognitive aptitude were classified as having intellectual disability based on this criteria.

A similar approach was applied to cognitive ability measures available in the Growing Up in Ireland (GUI) and Longitudinal Study of Australian Children (LSAC) cohorts. Using the same criterion for ID, no autistic children in these cohorts were identified as having intellectual disability.

Note, due to the limited number of autistic children with complete cognitive ability profiles in the LSAC cohorts, we did not include these cohorts in the downstream regression analyses. And for GUI, there was no downstream regression or mediation analyses, as no latent trajectories were identified in GMM analysis.

We have now clarified this in the text: “In MCS, we identified multiple variables linked to cognitive ability, SES, and area deprivation. Similarly, in LSAC-B, we identified multiple measures linked to SES, although there were no measures with sufficient sample size linked to cognitive ability or area deprivation. Subsequently, we conducted principal component analysis (PCA) for cognitive abilities, SES, and area deprivation in MCS, and for SES in LSAC-B. PCA was performed using a wide range of measures collected across sweeps (**Supplementary Table 27**), with one variable excluded from any pair showing a correlation coefficient greater than 0.70 to address multicollinearity. The first principal component (PC1) explained more than 40% of the variance for each corresponding factor, while subsequent components contributed substantially less, supporting the use of the respective PC1 as the summary measure for cognitive ability, SES, and area deprivation (**Supplementary Table 27**).”

2.12. I recognize that the authors include where available SES and some family (including parent) characteristics and do a ‘country analysis’ at one point but contextual and service factors are very influential in the population cohorts where diagnosis relies on parent report of received diagnosis (and this needs to be made clear – it is not a direct study acquired clinical ascertainment) which is dependent of course on services. This applies differently to the genetic cohorts, perhaps especially SPARK as a volunteer family/parent recruited sample with many young diagnosed children.

We agree that contextual and service factors are influential. We do indicate this in our Discussion of our original manuscript.

“Notably, there is substantial variation across the datasets explored, highlighting that age at diagnosis of autism is immensely complex, varying across geography and time. Local cultural factors, access to healthcare, gender bias, stigma, and camouflaging likely have an impact on who receives a diagnosis and when.”

We have also addressed this further in our FAQs in the original manuscript - please see our response to this question in the FAQ: "What is the impact of factors like camouflaging, waiting time, stigma on age at autism diagnosis?"

We have retained these points in the revised manuscript. We have also added the following lines in the Discussion to highlight this point: "Fifth, since the autism diagnoses in this study rely on community-based caregiver or self-reporting rather than standardised clinical assessments, there may be varying delays between when autism features first appear and when a formal diagnosis is received. However, our findings can guide future research using longitudinal cohorts, particularly sibling studies, that systematically track the emergence of autism features over time."

However, that said, these factors cannot explain the modest genetic correlation between the two autism polygenic factors. We also note that the aim of our study is not to identify all factors that affect age at autism diagnosis, but rather to investigate if earlier and later diagnosed autistic individuals have similar polygenic etiology.

2.13. p.19, line 20 – I would characterize the genetic correlation between the factors as 'moderate' (0.38) not 'low' but that might not make sense to geneticists.

Thank you for raising this point. This is contextual - it is low if you expect to see a high genetic correlation, which one would expect within the same diagnosis. However, we have now used the term "modest" throughout to refer to this genetic correlation.

2.14. p.20, line 36 – The paragraph on the camouflaging concept is interesting but for me almost entirely speculative with respect to the data presented. Personally, I remain to be convinced that females camouflage more than males (though the authors state this) not necessarily have more mental health problems (the sampling in this area is often biased) but they do not find sex effects on the two groups they identify and have no direct or indirect measures of the concept. Alternative clinical issues to mention would be about (1) what services need to know and (2) does this inform ages and targets for intervention and support of both impairing autism characteristics and mental health difficulties.

We have removed the paragraph about camouflaging. We have addressed this in the FAQs as some readers may be interested in the role of camouflaging. Indeed, one theory is that later diagnosis is primarily due to camouflaging of autistic features.

We do not think it is appropriate to discuss either services or intervention as we believe that it is premature based on our findings. This manuscript primarily demonstrates that earlier and later diagnosed autism differ genetically, and this is associated with differences in developmental trajectories.

Referee #3 (Remarks to the Author):

3.1 This paper reports a series of analyses that explore the differentiation of socio-behavioral trajectories and genetic profiles associated with age at Autism diagnosis. Overall, I found the paper to be interesting but rather exploratory. At times, the results seemed somewhat circular in the sense that the efforts to validate the differentiation of early vs. late diagnostic groups were oftentimes simply tests of whether these groups differed in the timing of their symptom presentations.

We thank the reviewer for their interest in our work. The fundamental aim of our work was to understand heterogeneity within autism through the lens of age at diagnosis. This is an important but complex issue, and has potential clinical consequences. So the main thrust of this paper was to triangulate the evidence using multiple different genetic and longitudinal methods and datasets that earlier and later diagnosed autism differ in their developmental and polygenic profiles. We have provided more detailed comments to the comments below and made substantial changes to the manuscript to address these points.

3.2 One salient feature of the paper, although perhaps stylistic, had a large impact on my reading. The authors frequently use the phrase, “we wondered” when describing the impetus for their analyses (examples below). This gave a strong impression that the manuscript reports a series of exploratory analyses that lead to one another in an unplanned and somewhat arbitrary fashion. I do think that simple declarative sentences could give a very different impression. For instance, rather than stating “given result X, we wondered whether results would persist after including covariate Z,” one could write “We tested the whether results would persist after including covariate Z.” That said, I do believe that the phrasing chosen may be revealing about the scientific process, and I am therefore generally concerned about how much of this manuscript is a report of a series of serendipitous results that stem from a much larger set of exploratory analyses. A pre registration would have been valuable for combatting this possibility, especially given the wide range of analyses reported, the relatively small sample sizes used for the growth mixture models, and the potential for flexibility when defining age groups using cut points.

“Given the significant association between age at autism diagnosis and the GMM latent class membership in LSAC-B and MCS, we wondered if these socio-behavioural trajectories explain any variance in age at autism diagnosis after accounting for socio demographic factors”

“Consequently, we wondered if the associations between ADHD and educational attainment (EA) PGS on age at autism diagnosis were due to parental indirect genetic effects”

“We wondered if the genetic signal for later diagnosed autism can be explained by diagnostic misclassification.”

“Given increasing number of autistic individuals being diagnosed in adolescence, we wondered if there are broad differences in the trajectories”

In the revised manuscript, we have substantially changed the writing style. We note that the previous choice of writing was simply a stylistic choice. We highlight that in our original submission, all our analyses have been conducted in at least two cohorts. Furthermore, in our original submission, we have triangulated the evidence using multiple methods, and run several sensitivity analyses to assess the robustness of these findings throughout. Specifically:

- For the longitudinal analyses, we have run LGCM in four cohorts, and GMM in three cohorts. We are unaware of other birth cohorts which have the information needed to further replicate these findings. GMM and LGCM find consistent results, which are in line with results from sibling studies. This includes the association between SDQ trajectories and mental health traits.
- All genetic analyses conducted in autistic individuals were primarily done in the SPARK cohort and the replicability was assessed in the iPSYCH cohort.
- For genomicSEM analyses, we used all available GWAS available to us and tested several alternate models.
- We conducted genetic correlation analyses using both individual cohort specific data stratified by age at diagnosis in SPARK and iPSYCH as well as using genomicSEM based approaches.
- For analysing if the effect of PGS on SDQ scores changes over time we used both MCS and ALSPAC. We are not aware of other large-scale longitudinal datasets that have SDQ information from childhood to adolescence.

Although we did not pre-register our initial analyses, we have now tested key genetic findings in another group of autistic (9,231) and family controls from SPARK whose genetic data was not available at the time of our original analyses but is available now (SPARK iWES3, released in August 2024). We refer to the original SPARK cohort as SPARK Discovery cohort, and the new cohort as the SPARK Replication cohort. We observe highly consistent results between Discovery and Replication cohorts. Due to the volume of the results, we do not reproduce the findings here.

We understand the authors' concerns about age cutoff points for Latent Growth Curve Models. It is precisely to allay this that we have run Growth Mixture Models, which do not require an age at diagnosis cut off point. However, we have now run additional sensitivity analyses by using other age at diagnosis cut off points in MCS and yet again find consistent results. Please see our response to Reviewer 2, comment 2 (2.2) that addresses this.

In sum, we hope the replication in a new subset of individuals from SPARK, the numerous sensitivity analyses, and a change in the writing style addresses this reviewers' concerns.

3.3 I found that the focus on SDQ trajectories in relation to early vs. late Autism diagnosis to be somewhat circular. The analysis finds that pre-adolescent diagnoses are associated with difficulties early on that tended to persist, and the adolescent and later diagnoses were associated with fewer difficulties early on, but an increase in difficulties over development. This doesn't seem to be surprising or especially validating of the onset groupings- I would expect that any disorder that is stratified by age at diagnosis would show early elevations in relevant symptoms for the early onset group and increases in relevant symptoms for the later onset group.

Although this is likely true of other psychiatric conditions, there are still many researchers and clinicians who work on autism who would challenge such a framework. Autism has always been conceptualised as a lifelong neurodevelopmental condition, with the core autism features emerging in early childhood, typically before age three. It is only relatively recently, over the last decade, that a small number of sibling studies have clearly demonstrated that some autistic individuals genuinely have a later emergence of autistic features (after age 3)^{29,30}.

Second, as mentioned in our response to reviewer 1, although several studies have documented variability in trajectories of autistic and related traits among those diagnosed as autistic, there has been no study that has demonstrated in prospective cohorts that diagnostic trajectories are associated with age at diagnosis. This, to our knowledge, is a critical and important gap in the literature to address, which we do so here. Indeed the prevailing hypothesis is that later diagnosis of autism is due to several other social factors, clinical severity, diagnostic overshadowing or simply not recognising the features early on.

Third, we present the longitudinal trajectory analyses alongside analyses of genetic differences to demonstrate that there are differences in both developmental trajectories and genetic profiles associated with age at autism diagnosis. We also observe key convergence between these two sets of analyses. PGS for later diagnosed autism has increasingly larger effects on SDQ total scores and peer relationship problems with age in two cohorts. Similarly, both trajectory and genetic analyses identify larger overlap with mental health issues among the later-emergent trajectory class and with later-diagnosed autism genetic factors.

In sum, we think the trajectory analyses are important as: (1) this is still an area of active debate where additional evidence is welcome; (2) to our knowledge, no study has assessed if developmental trajectories are associated with age at autism diagnosis in the general population; and (3) these analyses complement and inform the genetic analyses presented in the manuscript.

3.4 The authors go on to fit growth mixture models to the data from individuals with autism diagnoses only (it seems that, despite the data analyzed being from birth cohorts, these analyses are confined to individuals with an autism diagnosis). When they extract two latent trajectory classes, one corresponding to early difficulties and the other corresponding to increased difficulties over development, they find that these groups correspond well with the timing of autism diagnosis. Again, I'm not sure that there is anything particularly surprising about this finding: I would expect that the timing of symptom elevations to correspond well to the timing of diagnoses... I should also say that- given the nonspecific nature of the SDQ questionnaire- if the growth mixture model was fit in the full sample, there would likely be non-specificity with respect to the mapping of trajectory classes to Autism (as opposed to other childhood psychiatric disorders). The sensitivity test using ADHD diagnoses in place of Autism diagnoses seems rather weak.

We hope our response to our previous comment sufficiently addresses the issue of why this may be surprising to some readers, and is an important point to empirically demonstrate.

We focus our attention below to the issue of specificity and the choice of SDQ as a measure for the analysis.

We think there are several advantages to the SDQ that makes it a helpful measure for this study. First, it is widely used, allowing us to obtain different cohorts from different countries with this measure. This would have been exceedingly difficult, if not impossible, for a more autism-specific measure. Second, the SDQ is invariant across both age and genders³¹⁻³³. This is extremely useful for a study like this, where invariance across age and gender is needed to obtain interpretable trajectories and compare these trajectories against gender. Age-related invariance for commonly used autistic trait measures such as the SCQ and SRS is either unknown or minimally assessed³⁴. Third, linked to this, some studies have demonstrated that commonly used autism screening tools have lower accuracy in diagnosing autism in adolescents than in children, likely because these were primarily developed based on features observed in children, which may be different in autistic adolescents^{35,36}. This ties in to the point that measuring autistic traits among adolescents is not straightforward. Fourth, the availability of subscale measures in the SDQ allows us to compare the relative specificity of the findings in autism against ADHD, another neurodevelopmental condition, which would have been impossible with some of the standard autism measures as these do not measure features that are pertinent to ADHD. Fifth and finally, there is some advantage in using a broader measure for these analyses as many autistic individuals report a range of issues across adolescence. For example, internalising issues are common among many autistic individuals diagnosed later in life. It is unclear whether this is a feature of autism or due to co-occurring conditions. This may be incompletely captured by several traditional autism measures, but can be captured, to an extent, by the SDQ³⁷⁻³⁹. Taken together, using an autism-specific measure would create a methodological challenge: such measures are typically unavailable for non-autistic children or those diagnosed later from an early age. This results in a "catch-22" scenario, where it's impossible to study the developmental trajectory of autistic traits prior to diagnosis. By leveraging the SDQ, we bypass this limitation and can explore socioemotional outcomes in several datasets. To reflect these discussions, we have included the following line in the Limitation: "Second, our trajectory models were built using only the SDQ, which can capture a wide range of neurodevelopmental and mental health traits."

We selected ADHD as a comparator as this is another neurodevelopmental condition that emerges in childhood with diagnostic and etiological overlap with ADHD. Although the sample size of our initial analyses was relatively small, we have now imputed data for children with ADHD only (i.e., no autism) in the MCS dataset which increases the sample size to 325 individuals. This is larger than all autism-only cohorts in the longitudinal trajectory analyses except for the autism MCS-imputed cohort. We continue to observe differences in the trajectory models of autism and ADHD. In GMM analyses, although two latent classes emerged for ADHD, typically 10 - 11% of individuals with ADHD were assigned to the second class. This in contrast to autism where a more even split was observed across all cohorts and analyses (55%/45%). Furthermore, as expected, the association between age at ADHD diagnosis and the latent classes were driven by hyperactivity/inattention and conduct problem subscales. This is in contrast to autism, where peer relationship problems and emotional difficulties also contributed to age at diagnosis in addition to the SDQ total scores.

We have added these findings in the Main Text: “We still obtained two latent classes for SDQ total scores and most subscales in an imputed sample of children with only ADHD but no autism (N = 325). However, in contrast to the more even split across the two latent classes for autism, ADHD showed more uneven distributions, and associations with age at diagnosis were driven by hyperactivity/inattention and conduct problem subscales (**Supplementary Table 6**).”

We acknowledge that the nonspecific nature of the SDQ could lead to some overlap with other childhood psychiatric disorders if a growth mixture model were fit to the full sample. In addition to the sensitivity analyses on ADHD, it is also important to note that we are not using SDQ differences to distinguish autism from other childhood psychiatric disorders. Instead, we are employing the SDQ as a tool to understand how heterogeneity in broad developmental trajectories correlates age at diagnosis of autism, which, to our knowledge, has not been done before.

Finally, although we find different effects between ADHD and autism in our analyses, a lack of specificity is not necessarily an inherent limitation. Developmental differences, such as those reflected in the trajectory classes, may indeed overlap with other conditions. This overlap could itself be an important finding, contributing to a broader understanding of shared developmental mechanisms across neurodevelopmental and psychiatric conditions, which is an important ongoing debate in the field.

3.5 The GWAS of age of onset is interesting and valuable. This GWAS requires restricting the sample to cases which does make some of the interpretation ambiguous, especially with respect to genetic correlations with traits measured in the general population and case-control traits. That the r_g across samples for this GWAS was only .5 is somewhat concerning. The authors are forthcoming about the fact that the different distributions of age at diagnosis are very different across the two samples but don't seem to explain why (different ascertainment strategies?).

Re. the point: “This GWAS requires restricting the sample to cases which does make some of the interpretation ambiguous, especially with respect to genetic correlations with traits measured in the general population and case-control traits.” We agree. This is why we have conducted age at diagnosis stratified GWAS of autism. We have now more clearly articulated this in the text.

Indeed, the difference in genetic correlation between iPSYCH and SPARK primarily reflects ascertainment. SPARK is active ascertainment - individuals need to participate in SPARK. The iPSYCH is passive ascertainment, and based on blood spots obtained from all children born in Denmark and diagnosis based on health records. We have added the following line in the Results to contextualise this.

“The age at diagnosis in the two cohorts were moderately genetically correlated with each other ($r_g = 0.51$, $s.e.m = 0.19$, $P = 7.56 \times 10^{-3}$, between iPSYCH and meta-analysed SPARK) and had variable genetic correlation with other mental health phenotypes (**Supplementary Table 11**), and this is likely due to the different distribution of age at autism diagnoses in the two cohorts (**Extended Data Figure 5**) because of differing ascertainment.”

We agree that the r_g is only moderate, and believe this is due to the different distributions of age at autism diagnosis in the two cohorts. Encouragingly, we observe high genetic correlation when stratifying by age at autism diagnosis. For example the genetic correlation between SPARK_after10 and iPSYCH_after10 is 0.81 (s.e. = 0.13). Similarly, the genetic correlation between SPARK_before11 and iPSYCH_before 11 is 0.80 (s.e. = 0.10).

3.6 The genetic correlation between age of diagnosis and the case control autism trait is variably negative (whereas the authors note that for schizophrenia and depression, the age at diagnosis/onset is largely negatively genetically correlated with the phenotype itself, indicating that earlier diagnosis/onset is associated with greater polygenic propensity for the condition). This, like many of the results reported in this manuscript, is an intriguing observation, but does not give much in the way of decisive answers. The authors do conduct an analysis that splits the case control phenotype into early vs. late onset cases, and this appears to provide more consistent results, but such an analysis (like the growth mixture analysis) simply confirms that different ways of stratifying analyses by age at diagnosis agree at least moderately.

We agree that the variable genetic correlation (note, we obtain some positive genetic correlation - so not all genetic correlation with age at diagnosis are negative) is intriguing but doesn't provide clear answers. It is precisely for this reason that we conduct age of diagnosis stratified GWAS.

A preconception is that all autism emerges from one underlying polygenic latent trait, and later diagnosed autism merely has lower polygenic propensity. This model would predict variably negative genetic correlation, with earlier diagnosed autism being less negative. However, a positive genetic correlation would not be an outcome of this model, simply because even later diagnosed autistic individuals would have greater polygenic propensity than controls (and subsequently, one would expect to observe a negative genetic correlation with age at autism diagnosis).

An alternate model is that earlier and later diagnosed autism emerge from different, albeit correlated polygenic latent traits. This is what we test and find support for in the age of diagnosis stratified GWAS. An age of diagnosis GWAS does not allow us to convincingly separate the two models as identified by the reviewer. Consequently, age-of-diagnosis-stratified GWAS is not just a different way of confirming the results obtained from the age of diagnosis GWAS, but helps us distinguish between the two models.

We appreciate that although we had alluded to this theoretical framework in the introduction, we had not made this explicitly clear. We have re-written the introduction to introduce the two theoretical frameworks, and provided a Figure explaining the two models.

3.7 The genetic correlation analysis and Genomic SEM analyses (see Figures 4 and 5) are interesting and persuasive that the genetic architecture of early vs. late autisms diagnosis are partly separable, with the early diagnosis polygenic profiles being more characteristic of male cases and later diagnosis polygenic profiles being more characteristic of female cases. There are some concerns, such as the fact that the GWAS phenotypes themselves overlap in

their content (the sample overlap is not an issue for the estimators- it's more that the actual definitions of the cases e.g. ipsych before 11 and ipsych before 9 overlap substantially), which ensures clusters of very high rGs.

We understand this concern. We note that we do not use any overlapping GWAS (either in content or participants) for genomicSEM. We have now conducted sensitivity analyses after removing GWAS with overlapping phenotypes (iPSYCH_before10 and SPARK_before10, retaining iPSYCH_before9 and SPARK_before6), and observed consistent clustering (**Figure 3.7** below).

Figure 3.7: Clustering based on genetic correlation after excluding GWAS with overlapping phenotypes.

3.8 I did not find the analysis that takes into account genetic sharing with ADHD to be especially helpful in ruling out the potential for diagnostic misclassification. The genetic correlations presented in panel 5B indicate more substantial genetic sharing between later diagnosed autism (compared to early diagnosed autism) and a range of disorders, indicating that later diagnoses are far less genetically specific. This could result from genuine differences in genetic architecture by disorder onset, but it could also potentially stem from the differential diagnosis being more challenging and thus later diagnoses being less accurate (and hence more likely to contain individuals who present with autism-like features caused by non-autism psychiatric conditions). This is heightened by the fact that females are more likely to be diagnosed later and are also more likely to have less clear presentations of autism (such that differential diagnosis is especially challenging for them).

Thank you for raising this important point. We agree that later diagnosis of autism is indeed complex, and oftentimes challenging for a few reasons - the presentation is not quite clear, obtaining developmental history is complex, and it is not easy to rule out the presence of other mental health conditions.

We had indeed tested this using the genomicSEM analyses in the previously submitted version in **Supplementary Note Section 3**, but this was likely buried due to the sheer volume of the material. We have now refined the analyses, clarified the text in more detail, and written up these findings more clearly in **Supplementary Note 6**. We have responded in a lot more detail to a similar query to Reviewer comment 2.1, and we ask you to kindly refer to that response for details. However, we provide a brief response below.

In this study, we were concerned if the polygenic effects of later diagnosed autism can be attributable to a mix of the polygenic effects of autism and other mental health conditions. This is illustrated in the figure below. In the first cup, autism emerges from a single polygenic distribution. In the second cup, autism emerges from a single polygenic distribution, and later diagnosed autism is a combination of polygenic effects of autism and other mental health conditions. In the third cup, there are differential polygenic effects of earlier and later autism and additionally (and minimally), some other genetic influences.

Note 6, Figure 1

Illustrative example of the three models

We used genomicSEM to model whether the variance iPSYCH GWAS (where half the population are diagnosed after age 10, and so is a mixture of earlier and later diagnosis) can be explained using a combination of an earlier diagnosed autism GWAS (PGC/SPARK_before6) and other psychiatric conditions (ADHD, schizophrenia, PTSD, anorexia, bipolar) (**Note 6, Figure 2**). However, we find significant residual genetic variance. In other words, the polygenic effects of the iPSYCH autism GWAS cannot be characterised as the polygenic effects of earlier diagnosed autism and other mental health conditions.

This residual variance is specifically explained by later-diagnosed autism (SPARK_after10, $p < 0.04$). In other words, when we use a model that includes both early autism genetics and other psychiatric conditions, we still see a distinct genetic signal specific to later diagnosed autism that cannot be explained by either early autism or psychiatric conditions. But when also include a GWAS of later diagnosed autism, we observe: (1) a significant genetic correlation between both earlier and later diagnosed autism and the iPSYCH autism GWAS even after conditioning on the genetic effects of other mental health conditions; (2) no

significant genetic correlation between other mental health conditions and the autism GWAS; and (3) no significant residual variance in the iPSYCH autism GWAS.

Taken together, our analysis does not support the hypothesis that later diagnosed autism is likely a mixture of polygenic effects of autism and other mental health conditions.

Note 6, Figure 2

Path diagrams representing results from genomic multiple regression analyses using genomicSEM. The genetic effects of the iPSYCH autism GWAS were regressed on the genetic effects of the PGC-2017 autism GWAS (Models 1 and 2) or SPARK before 6 autism GWAS (Models 2 and 4), attention-deficit/hyperactivity disorder (ADHD), major depressive disorder (MDD), posttraumatic stress disorder (PTSD), schizophrenia (SCZ), anorexia nervosa (Anorexia), and bipolar disorder (Bipolar) simultaneously. In Models 3 and 4, we additionally included SPARK after 10 GWAS. Single-headed arrows indicate conditional genetic associations between the explanatory variables and the iPSYCH autism GWAS. Numbers represent standardized correlation coefficients, with standard errors in parentheses. Genetic associations between explanatory factors were accounted for in the analyses but are not shown on the graph for simplicity. The two-headed arrows connecting the genetic component of the iPSYCH autism GWAS to itself represent the residual genetic variance unexplained by the genetic influence of any of the GWAS included in the black circles. Solid lines indicate significant genetic associations, while dashed lines indicate non-significant associations. For simplicity, only the significant estimates and standard errors are shown, with the exception of the residual variance (and standard errors) in Models 3 and 4.

3.9 The figures presenting results of PGS analyses often seem to invite the reader to compare the magnitude (not simply direction) of effects for different PGSs. This is highly problematic, because the magnitude of a PGS effect relates to the power of the discovery GWAS on which it is based (which depends on sample size, heritability and polygenicity). Thus, PGSs based on different GWASs should only be compared with respect to direction of effect, rather than magnitude. In contrast, genetic correlations can be directly compared.

Thank you for raising this point. All PGS analyses have been conducted using the iPSYCH after 10 and before 11 GWAS, which has roughly similar sample size ($iPSYCH_{before11}$, $N_{autistic} = 9,500$; $iPSYCH_{after10}$, $N_{autistic} = 9,231$, and non-autistic individuals = 36,667) and are drawn from the same cohort. We have now revised the figures to avoid this potential misinterpretation. In the manuscript, the association with PGS is reported in three figures:

1. In 5B, we provide the association between PGS for iPSYCH after 10 and before 11 with two measures of early social communication in children from ALSPAC. We do note in the text that there is no statistical difference in association between the two sets of PGS.
2. In 5C, we report the age*PGS interaction estimate with SDQ total scores. In this figure, we are primarily reporting that the effect size of the PGS changes with age. Following Dudbridge, 2013, the underlying parameter driving this change in effect over age (and the differences between two sets of PGS) is the covariance between autism PGS and SDQ at specific time points. Everything else is either shared within a PGS across time points (e.g., training sample size, training sample heritability, number of markers in genotype model) or between the two sets of PGS (e.g., heritability of the phenotype in the testing sample, testing sample size). Furthermore, the interaction effects are in the opposite direction and statistically significant, which is the primary finding of this analysis. This primary finding seems unlikely to be influenced by differences in the heritability, sample size or polygenicity of the underlying GWASs. We provide the same plots for the SDQ total scores and subscales in **Extended Data Figure 8**.
3. In **Extended Data Figure 6**, we provide the association of PGS with both age at diagnosis stratified autism GWAS from iPSYCH and age at autism diagnosis. In **Extended Data Figure 6A**, the main point we are making is that both sets of PGS are over transmitted to autistic children from their parents, demonstrating that they are both associated with autism. In **Extended Data Figure 6B**, we demonstrate that they have opposite effects on age at autism diagnosis even after accounting for various clinical and socio-demographic factors. In this plot, as the reviewer suggests, the effects are in the opposite direction to both sets of PGS. Finally in **Extended Data Figure 6C**, we test the association between both sets of PGS and autism stratified by age at diagnosis in MCS, to primarily demonstrate that the effect of iPSYCH before 11 changes based on age at autism diagnosis.

In sum, we have revised several key figures and clarified the interpretation of the figures to ensure that the reader is not misled into misinterpreting the figures.

3.10 A small note: I'm not sure what s.e.m. (used when reporting nearly all effects) stands for. I believe "standard error of measurement," but I believe it should just be standard error, as standard error of measurement is a term that refers to uncertainty in an individual score whereas standard error refers to uncertainty in a parameter estimate.

Thank you. We have changed it to s.e. Throughout.

Referee #4 (Remarks to the Author):

4.1 Zhang et al present a compelling case of genetic heterogeneity in autism that is associated with the age of onset/diagnosis and subsequent differences in the trajectory of mental health profiles. The study is original and highly significant. The paper presents each step systematically, but following the narrative becomes a challenging task since the study involves too many steps. It is hard to keep track of the meandering storyline, a single overview (figure) would greatly facilitate the messaging.

We thank the reviewer for suggesting that the findings are original and significant, and apologise for the difficulty in following the work. We have now provided a single overview figure (**Figure 1**, please see below), and made substantial edits to the manuscript to make it shorter and improve the flow. Specifically, the manuscript has been reduced from 5,362 to 4,377 words, and we have moved several analyses to the Supplementary Note to make it easier to read and follow. Key changes that we have made are:

1. Moved the latent growth curve models to the Supplementary Note to focus on the data-driven growth mixture models.
2. Shifted results on the genetic correlation with age at autism diagnosis and other phenotypes and the SNP-based heritability analyses stratified by age at autism diagnosis to Supplementary Notes.
3. Generated new figures to make the findings easier to follow.
4. Moved the PGS association with SDQ trajectories to the final section to make it easier to follow.
5. Replicated the key findings by including a new cohort of autistic individuals from SPARK (SPARK replication, N = 9,701).

Aim 1: Investigate the association between developmental trajectories of socio-emotional behaviours and age at autism diagnosis in birth cohorts.

Key finding: Developmental trajectories are associated with age at autism diagnosis.

Aim 2: Quantify the SNP-based heritability of age at autism diagnosis and test if clinical and demographic factors reduce this heritability.

Key finding: Age at autism diagnosis is partly heritable and not explained by known clinical or demographic factors.

Aim 3: Investigate whether two or more polygenic factors influence age of autism diagnosis and if this partly explains genetic heterogeneity in autism.

Key finding: Two underlying polygenic factors linked to differences in age at autism diagnosis.

Aim 4: Characterise the genetic relationship between the two autism polygenic traits and other mental health conditions, developmental phenotypes and trajectories.

Key finding: Two autism polygenic traits have differing genetic relationship between mental health conditions and developmental trajectories.

Figure 1: Schematic of the study. *The study consists of four linked aims to understand if the polygenic etiology of autism differs by age at diagnosis. In the first aim, we modelled socio-behavioural trajectories among autistic individuals in birth cohorts and investigated their association with age at autism diagnosis. In the second aim, we investigated the SNP-based heritability of age at autism diagnosis in over 44,000 autistic individuals from three cohorts and investigated if the SNP-based heritability attenuates when accounting for various clinical and demographic factors. In the third aim, we investigated if the SNP-based heritability is better explained by considering as two or more polygenic factors correlated with age at autism diagnosis by using genetic correlation and genomicSEM based analyses. Finally, in the fourth aim, we investigated the genetic relationship between the two autism polygenic factors and mental health and developmental phenotypes, and developmental trajectories.*

4.2 First, the authors observe in four birth cohorts that SDQ scores follow different trajectories that are linked to age at autism diagnosis. The number of autism cases in each birth cohorts are small but the observation (with SDQ scores) is consistent in all four. The birth cohorts are from the same time-period, which I consider a positive. One of the four cohorts quietly disappears in the story (not shown in Figure 1) because the follow-up GMM analysis does not identify a two-trajectory model in this one.

The GUI cohort is not included in the GMM analyses as there are too few time points (3) to accurately identify two trajectories. In general, it is recommended having at least four time points to run a GMM (<http://www.statmodel.com/discussion/messages/14/20.html>). We explain this in the text. “The exception was the GUI cohort, where a one-trajectory model was optimal, likely due to fewer sweeps (three) for SDQ scores.”

4.3 A sensitivity analysis included a male-only analysis but why is there no mentioning of a female-only analysis?

We are unable to run a female-only analysis as there is insufficient sample size to conduct this. There are fewer than 100 females in each cohort. We have now mentioned this in the text. “We were able to run equivalent female-only analyses due to the low sample sizes.”

4.4 In the text (page 7 line 12ff) results are presented of a higher rate of depressive symptoms or anxiety but beyond significant p-values, it remains unclear how much of a difference is observed between the two latent classes.

In the **Supplementary Table 8**, we have provided the effect sizes (Regression coefficients), the standard errors and p-values for all mental health conditions. We do not provide effect sizes directly in the main manuscript to keep it easy to read.

4.5 The focus of the story subsequently changes to the genetic evaluation of age at diagnosis of autism. SPARK is used to perform a SNP heritability measurement showing an estimate of $h^2=0.11$. SPARK is an interesting cohort collected by the Simons Foundation and different from the Simon Simplex Collection (SSC). The latter involved deep and consistent phenotyping while SPARK is a community-based effort with more ambivalent data. I wonder why SSC is completely absent from this study.

We agree with the reviewer, and would have very much liked to have included the SSC. However, the SSC has no information on age at diagnosis, so we were unable to include it in the analyses. That said, iPSYCH is based on electronic health records which provides a different source of autism diagnosis compared to SPARK.

4.6 The other concern is that SPARK's range of age of onset is very much skewed toward <10 (extended Fig 2) which is one of the boundaries used in the paper at a later stage. In other words, the variation in age of onset driving the h2 signal in SPARK is lacking measurements of the >10 range. How does this affect the downstream observations?

We agree with the reviewer that there is variation in age at autism diagnosis. We are not entirely clear which specific analyses the reviewer is referring to, and so have touched upon a broad range of analyses.

- Age at diagnosis GWAS: The low relative percentage of those diagnosed after 10 in SPARK will affect this GWAS. However, we complement the SPARK age at diagnosis analyses with iPSYCH age at diagnosis and compare the findings. Although the genetic correlation between the age at diagnosis GWAS between SPARK and iPSYCH is moderate, this is primarily due to the distribution of the participants' age at diagnosis as age at diagnosis stratified GWAS have high genetic correlations between the cohorts. Encouragingly, the genetic correlation between SPARK after 10 and iPSYCH after 10 is high ($r_g = 0.81$, $se = 0.13$).
- Age at diagnosis stratified GWAS: We have expanded the SPARK cohort to include participants whose genotypes data were not available during the initial analyses ($N_{\text{replication}} = 9,701$). We refer to this cohort at SPARK Replication, and the previous SPARK cohort as SPARK Discovery. This increases the absolute number of autistic individuals in SPARK who were diagnosed after age 10 ($N = 6,243$), although the relative percentage is still at $\sim 10\%$. Although $N = 6,243$ is not large enough to likely identify significant GWAS loci, this is sufficiently large enough to conduct genetic correlations - the genetic correlations with SPARK after 10 and other autism GWAS has a median standard error of 0.13, which is not very high.
- The genomicSEM analysis is even better powered, as using genomicSEM we conducted a weighted meta-analysis using SPARK after10, iPSYCH after 10, and the FinnGen GWAS. We complement the genomicSEM based analyses using within cohort analyses stratified by age at autism diagnosis in SPARK and iPSYCH. Importantly, despite the demographic and ascertainment differences between these iPSYCH and SPARK, when participants are stratified by the same age at diagnosis (>10 years), the pattern of genetic correlations with mental health conditions is remarkably similar for most psychiatric and behavioral traits (**Figure 4.6** below).

Figure 4.6. Genetic correlation between SPARK after 10 and iPSYCH after 10 and other mental health and cognition related phenotypes. Estimates are genetic correlations and the error bars represent 95% confidence intervals.

In sum, we do not believe any of our findings are affected by the relatively small sample size of those diagnosed after age 10 in the SPARK cohort.

4.7 A little later, another population cohort (iPSYCH) is added to the analyses. Even though phenotype data may not be as deep as for other cohorts, the population-based design of this Danish birth cohort is outstanding. The age of onset distribution is very different from SPARK and the genetic correlation between the onset traits of SPARK and iPSYCH suggests that we are dealing with a different trait, much like major depression is different from schizophrenia (okay a little better) - it explains the different genetic correlation findings presented in figure 2.

We have now rewritten this to improve the flow and readability. We now introduce SPARK and iPSYCH at the same point in the text. We agree that the distribution of age at diagnosis differs substantially between the two cohorts, and partly explains the moderate genetic correlation and differing patterns of genetic correlations between the two GWAS and the psychiatric conditions. As mentioned earlier, encouragingly, when stratifying by age at diagnosis, there are high genetic correlations between similar/corresponding age at diagnosis stratified GWAS between the two cohorts (e.g., SPARK_after10 and iPSYCH_after10: $r_g = 0.81 (0.12)$). This demonstrates that it is the relative distribution of ages at autism diagnosis between the two cohorts that's the primary drive of the moderate genetic correlation between the two ages at diagnosis GWAS.

4.7 A further characterization of the genetic relationship between age at diagnosis and autism is next (page 10). First, some references are listed for patterns observed in schizophrenia and depression, but no mention of a bipolar study that was published 1-2

years ago. To my knowledge, genetic correlations between age at onset and any of the serious mental illnesses have not yielded consistent results.

We apologise, and we have now included the Bipolar GWAS. Although there is no negative genetic association between bipolar PGS and age at onset of bipolar disorder in the reference study, the authors speculate that this may be due to the low power of the bipolar disorder GWAS used. From our review of the literature, most studies report a negative genetic correlation/association between the condition and age at onset/diagnosis. Please see the table below for key studies in the field of mental health.

However, we have modified the text to acknowledge that this is not a universal pattern. The new text now reads: "Across several common mental health phenotypes, the age at diagnosis/onset tends to be negatively genetically correlated with the phenotype itself⁴⁰⁻⁴³, with bipolar disorder being an exception⁴⁴." We have also cited the relevant publications.

Condition	Key finding	Authors	DOI
Anorexia	Individuals in the lowest quartile of anorexia PGS had a later mean age of onset of anorexia.	Watson et al., 2021	10.1016/j.bpsgos.2021.09.001
Schizophrenia	Genetic correlation not conducted, but PGS for schizophrenia is negatively associated with age at onset.	Sada-Fuente et al., 2023	https://doi.org/10.1038/s41398-023-02508-0
Depression	Negative genetic correlation between age at onset/diagnosis of depression and depression [-0.49 (0.04)].	Harder et al., 2022	https://doi.org/10.1038/s41398-022-01888-z
Bipolar disorder	No relationship between bipolar PGS and age at onset of bipolar. The authors suggest that this may be due to the relatively low statistical power of the bipolar GWAS used.	Kalman et al., 2021	https://doi.org/10.1192/bjp.2021.102
Other mental health conditions	Consistent negative genetic correlation between age at onset and GWAS of anxiety, mood disorders, alcohol and substance use disorders, and other mental health conditions.	Chen et al., 2020	https://doi.org/10.1101/2020.11.20.20234302

Table 4.7: Key papers testing the association between age at diagnosis/onset and mental health conditions.

4.8 Around this time, the study shifts fully toward the iPSYCH data for a GWAS of autism separated by age at diagnosis. Yet, the genetic findings presented on page 11, lines 22-25, are very convincing. I am less impressed by the statement (on page 13) that "stratifying by age at diagnosis identifies a similar SNP-based heritability between iPSYCH, SPARK and PGC2017". Why is this so important? A similar heritability estimate says nothing about the underlying biology being similar.

We agree that this says nothing about biology per se. However, within the field of autism genetics, there has been discussions about the low SNP heritability of autism (despite the high twin heritability). One active area of discussion is the relatively low SNP-based heritability in iPSYCH compared to the primarily US-based cohorts included in the PGC and SPARK GWAS. In parallel, efforts, including by our team, to stratify autistic individuals into subgroups based on cross-sectional measures core autism features have not yielded any significant increase in SNP heritability, the sole exception being when we stratify by co-occurring ID (Warrier et al., 2022, Nature Genetics). The point we make is that by stratifying based on age at autism diagnosis, we increase SNP-based heritability among those with an earlier autism diagnosis.

However, to keep the manuscript easy to read, we have minimally mentioned the SNP-based heritability, and moved the key figure to **Supplementary Figure 12**.

“We further investigated predictions from Model 2 using two age of diagnosis stratified GWAS from iPSYCH with similar sample sizes: autism diagnosed before age 11 (iPSYCH_{before11}, N_{autistic} = 9,500) and autism diagnosed at age 11 or later (iPSYCH_{after10}, N_{autistic} = 9,231) (**Extended Data Figure 4**). The two GWAS were moderately correlated with each other ($r_g = 0.70$, s.e.m = 0.06) with a moderately higher SNP-based heritability for iPSYCH_{before11} ($h^2 = 0.17$, s.e.m = 0.01), compared to iPSYCH_{after10} ($h^2 = 0.13$, s.e.m = 0.01, **Supplementary Figure 11, Supplementary Table 14**).”

4.9 The paper now swerves back to phenotype measures and SDQ scores in the MCS cohort but this time not limited to autism cases. The iPSYCH-after11 PGS is significantly associated with increasing emotional behaviors, SDQ scores and other measurements. The effect was less pronounced in another cohort with SDQ scores, ALSPAC. The subtle switch from SDQ scores in autism cases to SDQ scores in the overall cohorts is easily missed.

We apologise for this. We have now moved this association with SDQ scores later in the manuscript (Aim 4), alongside the association with other mental health and developmental traits. This section consists of genetic correlation and PGS analyses from the general population - we use the term loosely to describe non-autistic populations. So, by placing this here we think it will improve the overall readability and flow of the manuscript, and not mislead readers into believing that the analysis was conducted only among autistic individuals. We have also clearly stated that we are testing the association in the general population, rather than focussing only on autistic individuals.

“Given the cross-sectional nature of the above analyses, we further investigated whether genetics from age-stratified GWAS supported trajectory modelling findings in the general population using the MCS cohort (N = 6,142 - 5,135) and ALSPAC (N = 7,172 - 4,977).”

4.10 The study moves back to SPARK again and an age-specific GWAS is performed, this time (i) before age 6, (ii) before age 11, and (iii) after age 10. How much overlap is there between (i) and (ii)? Are these groups exclusive or overlapping? Another issue is that another age category is introduced (namely the before age 6) that has no bearing on the previous analyses and why is there no effort made to perform a GWAS in iPSYCH with age <6? I

consider the findings presented in figure 4b as central and very insightful, but the way to this point in the paper is arduous.

We do apologise for this. We agree that this is the central point of this paper. We have rewritten it to introduce all the age at diagnosis stratified GWAS together in the paper. There is no overlap between SPARK before age 11 and after age 10. There is substantial overlap between before age 6 and before age 11 groups. Approximately 75% of all autistic individuals in the SPARK before 11 age group are also there in the SPARK before 6 age group. We have now generated a genetic correlation heatmap restricting it to GWAS without any overlapping participants, and observed similar clustering based on age at autism diagnosis (please see our response to Reviewer comment #3.7 and the accompanying figure). As LDSC accounts for participant overlap, we do not think the genetic correlation heatmap is biased by the sample overlap. However, the genomicSEM analyses, which formalises the two-factor model, have been conducted using GWAS with no overlap.

The SPARK before 6 GWAS was introduced because in some literature all autism diagnoses after this age are termed as a later diagnosis. Indeed, reviewer #2 suggests this in their comment #6. So we think it is important to stratify based on this threshold. With iPSYCH, stratifying it to autism before age 6 is underpowered (3.8K autistic individuals). Subsequently, we used the lowest age threshold where there are at least 5K autistic individuals to conduct a GWAS. This was critical as it was important to obtain a non-US centric GWAS of earlier diagnosed autism to distinguish age of diagnosis related effects from Geography related (US vs Europe) effects.

However, that said, we have now removed the iPSYCH diagnosed after age 12 GWAS as this has a large overlap (both participant overlap and genetic correlation) with iPSYCH autism diagnosed after age 10. We have now included the following lines in the manuscript to justify the inclusion of SPARK before age 6 and iPSYCH before age 9.

“These findings suggest that the age at autism diagnosis reflects a mixture of different age-dependent polygenic traits (**Model 2**) rather than a unitary polygenic trait (**Model 1**). To test this, we generated similar age-stratified GWAS in the SPARK cohort (discovery and replication cohort meta-analysed) as in the iPSYCH: autism diagnosed before age 11 ($SPARK_{before11}$) and autism diagnosed at age 11 or later ($SPARK_{after10}$). In addition, to provide further age-stratified resolution we generated a GWAS of autism diagnosed before age 6 ($SPARK_{before6}$). We chose this age threshold because previous research has often categorised autism diagnosis after age 6 as 'late diagnoses'. In iPSYCH, given the low sample size, we generated a GWAS of autism diagnosed before age 9 ($iPSYCH_{before9}$) (**Extended Data Figure 4**).”

We have also generated a schematic diagram (**Extended Data Figure 4**) that helps to orient the reader to the different GWASs generated and the relationship among them.

Extended Data Figure 4: Schematic diagram of age at autism diagnosis GWAS and age stratified autism GWAS

Schematic diagram illustrating the main GWAS conducted in the study using the SPARK and iPSYCH cohorts. We conducted two age at autism diagnosis GWAS. In addition, we conducted six case-control GWAS, where autistic individuals were stratified based on their age at autism diagnosis.

From Figure 4b we can conclude:

4.11 1. GWAS_SPARK_before6 is only weakly correlated with GWAS_SPARK_after10, suggesting a different genetic architecture - or is this largely a power effect with the latter group being too small?

Sample size will not bias the estimate of the genetic correlation but will decrease the precision. The standard error is sufficiently small to indicate that the genetic correlation is significantly less than 1. We find similarly low genetic correlation when we increase the sample size in the revised manuscript after meta-analysing the SPARK discovery and replication cohorts.

4.12 2. there remain modest genetic correlations between iPSYCH and SPARK, but the distribution of correlation suggests again that age at diagnosis categories (e.g. early vs later) show the lowest correlation levels;

We agree with this.

4.13 3. However, the genetic correlation of iPSYCH_before9 and iPSYCH_after10 or 11 remains high - as if iPSYCH data tells a different story than SPARK.

This is likely due to the distribution of the data (please see **Extended Data Figure 5** below). In iPSYCH, the median age at diagnosis is 10, with 70% of the participants having a diagnosis between ages 6 to 15. This is in contrast to the SPARK data where the distribution is

different. It is likely that the higher genetic correlation between iPSYCH_before9/iPSYCH_after10 ($rg = 0.7$, $s.e = 0.06$) is driven by a subset of the participants, diagnosed as autistic in mid-childhood (ages 7,8,9). Supporting this, in the genomicSEM model, iPSYCH_before 9 loads onto both earlier and later diagnosed autism factors. We discuss this in the main article: “The cross loading of iPSYCH_{before9} suggests that Factor 2 may impact behaviours in mid/late childhood as well, leading to a diagnosis before age nine.”

Furthermore, although SPARK_before6 and SPARK_after10 have low genetic correlation, they do have moderate/high genetic correlation with iPSYCH_before9 (SPARK_before6: $rg = 0.73$, $se = 0.10$, SPARK_after10: $rg = 0.51$, $se = 0.12$). The relatively low number of participants in iPSYCH diagnosed before age 6 precludes us from stratifying iPSYCH at this age.

Extended Data Figure 5: Distribution of age at autism diagnosis in SPARK and iPSYCH

Frequency histograms of age at autism diagnosis in iPSYCH and SPARK. Median and median absolute deviation for age at diagnosis, and sample sizes have been provided.

4.14 4. SPARK_{after10} shows weak to modest correlations with all other GWAS options.

Apologies, but this is untrue. SPARK_{after10} has high genetic correlations with other GWAS with a later median age at diagnosis both in the original manuscript as well as in the revision, when we meta-analyse the SPARK discovery and replication datasets, many of which are not statistically different from 1. Below, we present the results from the revised manuscript.

Phenotype 1	Phenotype 2	Rg	SE	P
-------------	-------------	----	----	---

SPARK_after10	iPSYCH_after10	0.8101*	0.1288	3.23E-10
SPARK_after10	iPSYCH_after11	0.7775*	0.1306	2.66E-09
SPARK_after10	iPSYCH_males	0.8566*	0.1366	3.55E-10
SPARK_after10	iPSYCH_females	0.6577	0.1644	6.31E-05

Table 4.14.4: Genetic correlation between SPARK and selected GWAS from iPSYCH. *Indicates genetic correlations not statistically different from 1.

4.15 The narrative continues with a genomicSEM analysis that suggests that two genetic latent classes best explain the heterogeneity. I am also left with the idea that the different cohort studies (SPARK, iPSYCH, FinnGen) represent different traits with only modest genetic overlap. I am therefore less impressed (and not convinced) with the findings that earlier and later diagnosed autism genetic factors are associated with different mental health profiles.

We agree that there will be cohort-level heterogeneity which will be captured when running the genetic correlation analyses using the earlier vs later diagnosed autism polygenic traits. In the original paper, we had conducted additional sensitivity analyses demonstrating that these effects are observed even within individual cohorts, but this was likely buried in the Supplementary Note. We have now taken the opportunity to revise these analyses using the expanded SPARK cohort, create better figures, and provide the revised figure as **Extended Data Figure 7**, which we provide below.

Furthermore, when comparing late-diagnosed autism (age >10) between SPARK and iPSYCH, we observe largely consistent genetic correlations across most phenotypes. However, there are notable differences for cognitive aptitude (SPARK $rg=0.45$, iPSYCH $rg=0.15$) and educational attainment (SPARK $rg=0.29$, iPSYCH $rg=0.00$). These differences likely reflect participation bias in SPARK. Importantly, despite the demographic and ascertainment differences between these iPSYCH and SPARK, when participants are stratified by the same age at diagnosis (>10 years), the pattern of genetic correlations with mental health conditions is remarkably similar for most psychiatric and behavioral traits (**Figure 4.15** below).

Finally, we agree that there are likely other sources of genetic heterogeneity which are not fully captured by age at autism diagnosis. We state this in the Discussion: “It is likely that other dimensions contribute to heterogeneity in autism, including potentially further genetic differences based on age at diagnosis. For example, a significant proportion of the variation in the FinnGen autism GWAS was not explained by either of the two factors (**Figure 5A**).”

Extended Data Figure 7: Within-cohort genetic correlation between age at diagnosis stratified autism GWAS and mental health and cognition related phenotypes

Genetic correlation between age at autism stratified GWAS in SPARK (meta-analysed from discovery and replication cohorts) and iPSYCH and other mental health and cognition related phenotypes. Points represent genetic correlation estimates and whiskers indicate 95% confidence intervals. Green represents the earlier diagnosed autism GWAS (iPSYCH before 9 and SPARK before 6), and purple represents later diagnosed autism GWAS (iPSYCH and SPARK after 10). Asterisk () indicates significantly different genetic correlation between the earlier and later diagnosed GWAS ($P < 0.05$, two-tailed Z test).*

Figure 4.15. *Genetic correlation between SPARK after 10 and iPSYCH after 10 and other mental health and cognition related phenotypes. Estimates are genetic correlations and the error bars represent 95% confidence intervals.*

4.16 Overall, I agree with the authors that "there is substantial variation across the datasets explored" (page 20, line 30/31). The authors conclude that age at diagnosis is an important axis of heterogeneity of autism. However, they present a convoluted narrative with diverse cohorts, different types of analyses, and subtle changes that may indeed support their claim. The study certainly highlights the heterogeneity of autism spectrum disorder.

We hope that the changes made throughout the manuscript, including substantially editing the article, additional analyses to assess replicability of the key findings and sensitivity analyses helps address these reviewers' concerns.

References

1. Groemping, U. Relative Importance for Linear Regression in R: The Package relaimpo. *J. Stat. Soft.* **17**, 1–27 (2007).
2. Berument, S. K., Rutter, M., Lord, C., Pickles, A. & Bailey, A. Autism screening questionnaire: Diagnostic validity. *The British Journal of Psychiatry* **175**, 444–451 (1999).
3. Sigg, C. nsprcomp: Non-Negative and Sparse PCA. *CRAN: Contributed Packages* The R Foundation <https://doi.org/10.32614/cran.package.nsprcomp> (2013).
4. Daniels, A. M. & Mandell, D. S. Explaining differences in age at autism spectrum disorder diagnosis: a critical review. *Autism : the international journal of research and practice* **18**, (2014).
5. Dalsgaard, S. *et al.* Incidence Rates and Cumulative Incidences of the Full Spectrum of Diagnosed Mental Disorders in Childhood and Adolescence. *JAMA Psychiatry* **77**, 155–164 (2020).
6. Pender, R., Fearon, P., St Pourcain, B., Heron, J. & Mandy, W. Developmental trajectories of autistic social traits in the general population. *Psychol. Med.* 1–9 (2021).
7. May, T., Brignell, A. & Williams, K. Parent-reported Autism Diagnostic Stability and Trajectories in the Longitudinal Study of Australian Children. *Autism Res.* **14**, 773–786 (2021).
8. Estes, A. *et al.* Behavioral, cognitive, and adaptive development in infants with autism spectrum disorder in the first 2 years of life. *J Neurodev Disord* **7**, 24 (2015).
9. McDonald, N. M. *et al.* Developmental Trajectories of Infants With Multiplex Family Risk for Autism: A Baby Siblings Research Consortium Study. *JAMA Neurol* **77**, 73–81 (2020).
10. Choi, B., Leech, K. A., Tager-Flusberg, H. & Nelson, C. A. Development of fine motor skills is associated with expressive language outcomes in infants at high and low risk for autism spectrum disorder. *J. Neurodev. Disord.* **10**, 14 (2018).
11. Ozonoff, S. *et al.* Onset patterns in autism: Variation across informants, methods, and timing. *Autism Res.* **11**, 788–797 (2018).
12. Campbell, S. B., Moore, E. L., Northrup, J. & Brownell, C. A. Developmental Changes in Empathic

- Concern and Self-Understanding in Toddlers at Genetic Risk for Autism Spectrum Disorder. *J. Autism Dev. Disord.* **47**, 2690–2702 (2017).
13. Gentles, S. J. *et al.* Trajectory research in children with an autism diagnosis: A scoping review. *Autism* (2023) doi:10.1177/13623613231170280.
14. Risk of psychiatric comorbidity with autism spectrum disorder and its association with diagnosis timing using a nationally representative cohort. *Research in Autism Spectrum Disorders* **104**, 102134 (2023).
15. Kerns, C. M., Jessica E. Rast, M. P. H. & Shattuck, P. T. Prevalence and Correlates of Caregiver-Reported Mental Health Conditions in Youth With Autism Spectrum Disorder in the United States. *Psychiatrist.com* (2020).
16. Hosozawa, M., Sacker, A. & Cable, N. Timing of diagnosis, depression and self-harm in adolescents with autism spectrum disorder. *Autism* (2020) doi:10.1177/1362361320945540.
17. Harmens, M., Sedgewick, F. & Hobson, H. The Quest for Acceptance: A Blog-Based Study of Autistic Women’s Experiences and Well-Being During Autism Identification and Diagnosis. *Autism in Adulthood* (2022) doi:10.1089/aut.2021.0016.
18. Leedham, A., Thompson, A. R., Smith, R. & Freeth, M. ‘I was exhausted trying to figure it out’: The experiences of females receiving an autism diagnosis in middle to late adulthood. *Autism* (2019) doi:10.1177/1362361319853442.
19. Russell, A. S. *et al.* Who, when, where, and why: A systematic review of ‘late diagnosis’ in autism. *Autism Res* (2024) doi:10.1002/aur.3278.
20. The psychometric properties of the Norwegian version of the social responsiveness scale in a neuropsychiatric sample. *Research in Autism Spectrum Disorders* **95**, 101973 (2022).
21. Marinopoulou, M., Billstedt, E., Wessman, C., Bornehag, C.-G. & Hallerback, M. U. Association Between Intellectual Functioning and Autistic Traits in the General Population of Children. *Child Psychiatry & Human Development* 1–12 (2023).
22. Skuse, D. H. *et al.* Social communication competence and functional adaptation in a general

- population of children: preliminary evidence for sex-by-verbal IQ differential risk. *Journal of the American Academy of Child and Adolescent Psychiatry* **48**, (2009).
23. Hastings, S. E., Hastings, R. P., Swales, M. A. & Hughes, J. C. Emotional and behavioural problems of children with autism spectrum disorder attending mainstream schools. *International journal of developmental disabilities* **68**, (2021).
24. Adachi, M. *et al.* Adaptation of the Autism Spectrum Screening Questionnaire (ASSQ) to preschool children. *PLOS ONE* **13**, e0199590 (2018).
25. Warriar, V. *et al.* Genetic correlates of phenotypic heterogeneity in autism. *Nat. Genet.* **54**, 1293–1304 (2022).
26. von Luxburg, U. A tutorial on spectral clustering. *Statistics and Computing* **17**, 395–416 (2007).
27. Totsika, V., Hastings, R. P., Emerson, E. & Hatton, C. Early Years Parenting Mediates Early Adversity Effects on Problem Behaviors in Intellectual Disability. *Child Development* **91**, e649–e664 (2020).
28. The role of physical environmental characteristics and intellectual disability in conduct problem trajectories across childhood: A population-based Cohort study. *Environmental Research* **209**, 112837 (2022).
29. Ozonoff, S. *et al.* Diagnosis of Autism Spectrum Disorder After Age 5 in Children Evaluated Longitudinally Since Infancy. *J. Am. Acad. Child Adolesc. Psychiatry* **57**, 849–857.e2 (2018).
30. Bazelmans, T. *et al.* Mid-childhood autism sibling recurrence in infants with a family history of autism. *Autism Res.* (2024) doi:10.1002/aur.3182.
31. Speyer, L. G., Auyeung, B. & Murray, A. L. Longitudinal Invariance of the Strengths and Difficulties Questionnaire Across Ages 4 to 16 in the ALSPAC Sample. *Assessment* **30**, 1884–1894 (2023).
32. Murray, A. L., Speyer, L. G., Hall, H. A., Valdebenito, S. & Hughes, C. A Longitudinal and Gender Invariance Analysis of the Strengths and Difficulties Questionnaire Across Ages 3, 5, 7, 11, 14, and 17 in a Large U.K.-Representative Sample. *Assessment* **29**, 1248–1261 (2022).

33. Woerner, W. *et al.* The Strengths and Difficulties Questionnaire overseas: Evaluations and applications of the SDQ beyond Europe. *Eur. Child Adolesc. Psychiatry* **13**, ii47–ii54 (2004).
34. Frazier, T. W. *et al.* Confirmatory factor analytic structure and measurement invariance of quantitative autistic traits measured by the Social Responsiveness Scale-2. *Autism* (2013) doi:10.1177/1362361313500382.
35. Chesnut, S. R., Wei, T., Barnard-Brak, L. & Richman, D. M. A meta-analysis of the social communication questionnaire: Screening for autism spectrum disorder. *Autism : the international journal of research and practice* **21**, (2017).
36. Hollocks, M. J. *et al.* Brief Report: An Evaluation of the Social Communication Questionnaire as a Screening Tool for Autism Spectrum Disorder in Young People Referred to Child & Adolescent Mental Health Services. *Journal of Autism and Developmental Disorders* **49**, 2618–2623 (2019).
37. Mandy, W. *et al.* Mental health and social difficulties of late-diagnosed autistic children, across childhood and adolescence. *Journal of Child Psychology and Psychiatry, and Allied Disciplines* **63**, 1405 (2022).
38. Cremone, I. M. *et al.* Measuring Social Camouflaging in Individuals with High Functioning Autism: A Literature Review. *Brain Sciences* **13**, 469 (2023).
39. Milner, V., Mandy, W., Happé, F. & Colvert, E. Sex differences in predictors and outcomes of camouflaging: Comparing diagnosed autistic, high autistic trait and low autistic trait young adults. *Autism* **27**, 402 (2022).
40. Watson, H. J. *et al.* Common Genetic Variation and Age of Onset of Anorexia Nervosa. *Biol Psychiatry Glob Open Sci* **2**, 368–378 (2022).
41. Sada-Fuente, E. *et al.* Common genetic variants contribute to heritability of age at onset of schizophrenia. *Translational Psychiatry* **13**, 1–9 (2023).
42. Harder, A. *et al.* Genetics of age-at-onset in major depression. *Translational Psychiatry* **12**, 1–7 (2022).
43. Feng, Y.-C. A. *et al.* Findings and insights from the genetic investigation of age of first reported

occurrence for complex disorders in the UK Biobank and FinnGen. *bioRxiv* (2020)

doi:10.1101/2020.11.20.20234302.

44. Kalman, J. L. *et al.* Characterisation of age and polarity at onset in bipolar disorder. *The British Journal of Psychiatry* **219**, 659–669 (2021).

Referees' comments

Referee #1:

I appreciated the thorough and comprehensive response from Zhang and co-authors to the initial set of reviews. The manuscript is considerably strengthened the greater clarity of the focus, from the revised title to the schematic summary of the 4 aims and focus on contrasting specific theoretical models. I will focus my review of the revision on their response to feedback as Reviewer 1, with additional comments on the revised manuscript as a whole.

Thank you! We are pleased that you find the revised manuscript clearer and of interest.

Comment 1.1 that (a) weakness of the manuscript is insufficient attention to the fundamental question of whether 'age of diagnosis' is a biological construct that can be heritable (vs familial) and directly influenced by specific genetic mechanisms. 'Age of diagnosis', as the authors acknowledge in their selection of covariates to assess mediating and moderating relationships, likely reflects the complex interplay among multiple factors.

The authors now acknowledge in the Discussion, 'There is substantial variation across the datasets explored, highlighting that age at diagnosis of autism is immensely complex, varying across geography and time. Local cultural factors, access to healthcare, gender bias, stigma, ethnicity, and camouflaging, likely have an impact on who receives a diagnosis and when.' And in the FAQs, 'It is important to note that in this study, genetics explain only about 11% of the total variation in when someone receives an autism diagnosis'. They added a detailed analysis of Variance in age at autism diagnosis explained by various sociodemographic and clinical factors in the SPARK cohort (Table 1.1) They also found that 'SNP-based heritability did not significantly attenuate after accounting for several of the child's developmental and clinical phenotypes and parental socio-demographic phenotypes... None of the observed clinical factors mediated the SNP-based heritability of age at autism diagnosis.' They present two theoretical models that could account for SNP heritability for age at diagnosis; ie how age at diagnosis is indexing genetic heterogeneity in autism. This could result from a less penetrant version of the same polygenic propensity for earlier diagnosed autism, resulting in a less prominent manifestation of behavioral traits, or from at least two correlated polygenic aetiologies that are developmentally different with earlier and later differentially associated with the two different polygenic aetiologies. Evidence from further analyses (SNP-based heritability is not attenuated when controlling for a range of child's clinical factors) supports the 2nd model. The authors elaborate and anchor the study analyses and findings as contrasting these two models. This clarification is helpful and for the most part addresses my previous concern. I also appreciate the change in Title to "The polygenic architecture of autism varies by age at diagnosis", which a more straightforward description of the study findings and focus than the original.

Thank you.

Comments 1.2a and 1.2b.

The supplemental analyses are compelling. That said, whether polygenic heritability genuinely influenced age of diagnosis independent of clinical factors is still uncertain. The authors report that there is no meaningful attenuation of the SNP-based heritability from the baseline (sex, ID,

and genetic principal components) after accounting for parent-reported IQ scores, age at walking independently, age at first words, language regression, any other regression, SCQ total scores, RBS-R total scores. However, they do not have continuous data on language skills, nor directly assessed cognitive level or autistic symptoms that have previously been reported to correlate with age of diagnosis. The authors refer to developmental phenotypes (e.g., in Aims 1 and 4, but the data in the available cohorts is mainly limited to milestones, presence/absence of intellectual disability and ‘cognitive aptitude’ derived by factor analysis of available, mainly parent-report measure. **This limitation reflects the available data but could be acknowledged more explicitly and represents an important caveat to their conclusion that ‘(observed) clinical factors did not mediate the SNP-based heritability of age at autism diagnosis’.**

Indeed, in clarifying ‘What might this SNP heritability for age at autism diagnosis reflect?’ (pg 7 of their Response) they note that the contrasting theoretical models would lead to different patterns and distributions of clinical manifestations at different ages. **Would the authors disagree that age of diagnosis ultimately is a proxy for these differences in onset and expression? How else could age of diagnosis be influenced by genetic factors?** So it is important to consider the possibility that the lack of evidence of mediation is related to limitations in the available phenotypic data.

Yes, we agree that “age of diagnosis is ultimately a proxy for these differences in onset and expression”. We have now amended the manuscript to reflect this, including in the title, which now reads: “The polygenic and developmental profiles of autism differ by age at diagnosis”. In the results section, where we test whether the SNP-based heritability is attenuated after including various covariates, we now add: “We acknowledge that this lack of attenuation may be a result of having imperfectly measured clinical phenotypes.” Furthermore, in the Discussion, we have added the following line: “We find that the genetic effects on age at autism diagnosis are not mediated by several of these measured developmental and demographic factors, but we acknowledge that there may be other unmeasured developmental and demographic factors.”

Our original intention was to flag that this cannot simply be explained by co-occurring developmental delays in major motor or language milestones, or co-occurring intellectual disability. In other words, these findings do not primarily reflect autistic children with and without global developmental delay. But rather, as the reviewer points out, these findings reflect other developmental changes ultimately leading to changes in onset and expression.

Re: the authors comment in the rebuttal regarding my original comment 1b (‘Regarding “Tracking of Development”, we are unsure exactly what the Reviewer is requesting’, pg 15).

This is a specific reference to the original Supplemental material which did refer to developmental tracking/follow-up of the longitudinal cohorts. Specifically, “The GUI study aims to describe and understand the lives of children in Ireland, with Cohort 98’ tracking their development from childhood to adulthood” (pg 3, 487479_0_supp_4498569_shhnyr.pdf)

Thank you for the clarification. We have now edited the text to respond to the original comment and no longer use the term “Tracking of development”. The edited paragraph now reads:

“The GUI study aims to describe and understand the lives of children in Ireland, with Cohort 98’ following the children from childhood to adulthood and identifying key factors associated with their well-being. It also seeks to examine the effects of early childhood experiences on later life

outcomes, map variations in children's lives, gather children's perspectives, and provide data to inform the development of effective policies and services for children and families.”

Comment 1.3b regarding ethnicity and variation in age of diagnosis.

I appreciate the authors' clarification of what data on race and ethnicity was available in the various datasets reported and that the genetic analyses (SPARK and iPSYCH) included only individuals of genetically inferred European ancestries. Recognizing the limitations in the ethnic diversity of available samples and that the study was not designed to examine ethnic variation in age of diagnosis, their addition about the need to extend the findings to individuals of other genetic ancestry in the Limitations section of the Discussion addressed the concern expressed in review of the original submission.

Thank you.

Comment 1. 4 Further clarification is also needed regarding how the age of diagnosis was stratified to 'childhood vs adolescence'.

I appreciate the clarification regarding the rationale for the cut-offs for each cohort, and the sensitivity analyses demonstrating a general effect of age of diagnosis. The authors did not specifically address the points I had included in my comment; i.e., “inconsistency in the application of a decision rule to determine the cut-off – LSAC-B and LSAC-K include assessments at both 9 and 11 years, yet a 9-year cut-off was applied to the former and a 11-year cut-off to the latter. Relatedly, not clear why the authors did not apply the cut-off somewhat later (e.g., consistently from 11 years), if their intent was to identify a subgroup diagnosed in adolescence.” For the latter point, the cut-off for the GUI cohort could be the 13-year visit. I suspect the analyses are robust to these decisions, but some clarification would still be helpful, especially the rationale for applying different cut-points to the LSAC-B and LSAC-K cohorts.

Thank you for this query. We focussed on the 9 - 11 window as this coincides with the increase in prevalence of autism in girls reported in epidemiological studies in Denmark. This has been mentioned both in Box 1: “This cutoff period was chosen to reflect the cutoff used in the Latent Growth Curve models, and represents a time window characterised by the onset of puberty, transition from primary to secondary school, and an increase in the number of autistic girls being diagnosed. ” and in further detail, in the Methods:

“Childhood diagnosed (diagnosed before ages 9 - 11, depending on the cohort), and adolescent diagnosed (diagnosed after the ages of 9 - 11, depending on the cohort) were a priori defined (see Extended Data Figure 3). We chose this 9 - 11 age window as our cutoff since it aligns with the onset of puberty, the transition from primary to secondary school, and aligns with epidemiological evidence showing increased autism incidence among females during this window³⁴. An earlier cutoff was not feasible as only MCS (ages 5, 7) and GUI (age 7) recorded autism diagnoses before this window. A later cutoff was not possible due to no autism diagnoses in MCS after age 14. To further examine the relationship between age at diagnosis and socioemotional and behavioural outcomes, we conducted additional Latent Growth Curve Models for autistic children using stepwise groupings by age at diagnosis in MCS and LSAC-B (see Supplementary Note 5 and Supplementary Figure 10).”

With regard to using ages 11 or 13 as thresholds, there was no assessment of autism diagnosis at age 11 in GUI, and the next available assessment was age 13. Furthermore, there are few children diagnosed as autistic at ages 13 and beyond in LSAC-B, making latent growth curve

models using a threshold of 13 underpowered (Please see Supplementary Figure 10c). However, as provided in Supplementary Figure 10 we demonstrate consistent results when varying the age at diagnosis cutoffs in MCS and LSAC-B.

We have provided further information in Supplementary Note 5. “While our primary analyses utilised cohort-specific thresholds corresponding to ages 9-11, we explored the robustness of this approach. In GUI, autism diagnosis assessment was unavailable at age 11, necessitating use of the next available assessment at age 13. Conversely, LSAC-B had insufficient numbers of children diagnosed at age 13 and beyond, making latent growth curve models using a higher threshold underpowered. Nevertheless, we conducted sensitivity analyses in MCS and LSAC-B using various age cutoffs. Consistent with previous analyses, groups with a higher proportion of later diagnoses exhibited lower baseline difficulties (intercepts) but steeper increases over time (slopes) across SDQ subscales. Models with smaller group sizes showed estimation issues in LSAC-B (e.g., negative latent variances for a parameter or a non-positive definite latent covariance matrix), supporting our choice of the age cut in dichotomous-group analysis. “

1.7 Regarding Comment 1.5 about distinctive socio-behavioural trajectories.

In response to my suggestion that they consider their findings in relation to a recent scoping review on trajectory research in children with an autism diagnosis (Gentles et al., 2024), they noted that of the 55 studies cited in the review, only one study has investigated the trajectories of social, behavioural, emotional and other related autistic traits prospectively, prior to an autism diagnosis, which “prevents us from asking if changes in developmental trajectories are indeed associated with age at autism diagnosis”. In reviewing these studies, they also conclude that “Taken together, ours is the first study to systematically investigate if changes in social, emotional behavioural difficulties are associated with age at autism diagnosis.” It is important to distinguish between changes in trajectories from changes in difficulties. To assess changes in trajectories, one would need to identify a change in slope or other trajectory parameter linked in time to a specific event (such as age of diagnosis). **What the authors report in the paper are differences in trajectories (ie how difficulties change over time) in the age of diagnosis groups as a whole, which is not fundamentally different than other longitudinal cohorts that include autism diagnosis at baseline. So while it is true that a unique feature of the current paper is reporting autism-related developmental trajectories in birth cohorts, it is not clear whether trajectories prior to diagnosis represent ‘different pathways into diagnosis’ or just different clinical profiles among the heterogeneous condition of autism.** While a minor point, I don’t fully agree with the authors’ statement that “(Although developmental variation is well-characterised in autism, to our knowledge), no study has robustly investigated if this variation is associated with age at autism diagnosis in prospective cohorts”. Other studies have reported how developmental trajectories vary by age of diagnosis within their inception cohorts, although this tends to be a narrower range than in the birth cohorts reported in this paper.

We thank you for this important semantic clarification (changes vs differences in trajectories), which we agree with.

Re. “it is not clear whether trajectories prior to diagnosis represent ‘different pathways into diagnosis’ or just different clinical profiles among the heterogeneous condition of autism.”. We agree. Unfortunately, this is not something our study can evaluate and we welcome further research that can help clarify this important issue.

Re. “ (Although developmental variation is well-characterised in autism, to our knowledge), no study has robustly investigated if this variation is associated with age at autism diagnosis in prospective cohorts”. We have now modified this line to read: “Consistent with the larger literature on developmental variation in autism²¹, our trajectory analyses across multiple birth cohorts converge with the genetic findings to suggest different developmental pathways to autism diagnosis.”

Overall, the authors have responded sufficiently to the points that I as well as the other reviewers raised in their reviews and I appreciate the sensitivity and scientific rigour that has guided the supplemental analyses and text revisions.

Thank you!

Referee #2

Nature MS 2024-07-15689A-Z

Reviewer #2:

2. 1. Early vs. Later autism vs. autism with/without psychiatric conditions. The response and additional analyses and the model are helpful in further clarifying that the early vs. later autism diagnosis genetic architecture is not due to overlap of the latter with psychiatric/mental health (MH) conditions. I find the model (in Note 6 SM) helpful.

Thank you!

2. 2. Preschool (early) diagnosis. The MCS diagnosed before 5 years and the LSAC-B LGCMs and Figures helpful in clarifying the 'stepped' pattern of MH SDQ scores in relation to age of autism diagnosis/recognition. This also provides some confirmation that the threshold of ~11 years used in the genetic cohorts might be meaningful.

Thank you!

3. Age threshold of 'early' vs. 'later' diagnosis. The slight change in terminology to 'earlier' vs. 'later' is helpful and I agree no meaningful biological or clinical (or even service/societal) rationale for a non-arbitrary distinction of what constitutes 'early' diagnosis. The authors note that the 9–11-year-old threshold is around puberty (as well as high school transition) and I wonder if some biological factors (whether in part genetic or not) might be important too and should be mentioned).

Thank you. We agree that other biological factors are likely to be important. It is not immediately clear whether this is puberty, or earlier (around the period of adrenarche), or a complex combination of multiple biological factors. We have now reviewed the literature on this, and found limited consistent and robust evidence linking altered puberty/adrenarche and age at autism diagnosis. Consequently, our opinion is that any discussion about this is speculative at this stage. Given this (and the word limit) we have decided not expand our Discussion further to other biological factors.

4. Lack of measure of autism severity in population cohorts. SDQ might correlate fairly highly with autism questionnaire measures but I'm not sure in the population cohorts this helps as without separate autism severity and socio-emotional-behavioral measures one cannot account for the impact of autism severity per se in age of diagnosis. Helpful in SPARK to test whether accounting for SCQ and RBR and parent-reported IQ relates to PGS for earlier vs. later diagnosed and age of diagnosis – in SPARK quite low (i.e. 8% for SCQ) but higher in some other cohorts (as in Inset in Extended Data Figure 2).

We agree. We have made the following changes to address this:

Results: “In Aim 1, we investigated whether autistic individuals have varying trajectories of socioemotional and behavioural trajectories and whether these are associated with age at autism diagnosis in three birth cohorts (N = 89 to 188 autistic individuals with recorded age at diagnosis between 5 to 17 years). These are the Millennium Cohort Study (MCS, participants born in 2000), and Longitudinal Study of Australian Children: Kindergarten cohort (LSAC-K, 1999) and Birth cohort (LSAC-B, 2003) (Supplementary Table 2, Extended Data Figure 3, Supplementary Note 1). All three cohorts collected longitudinal information on socioemotional and behavioural development using the caregiver-reported Strengths and Difficulties Questionnaire (SDQ)²³. The SDQ has five subscales (emotional, conduct, hyperactivity/inattention, peer problems, and prosocial behaviours), and the total score of difficulties (hereafter “total difficulties”) is the summed score of the first four subscales. The SDQ is widely used, has excellent psychometric properties^{24–26}, and is largely invariant across age, sex, and different populations^{27–29}, suggesting that it is measuring the same latent trait across these demographic variables. **In addition, the SDQ is moderately correlated with autism-specific measures^{30–33}, although it does not capture all of the core diagnostic features of autism.** Because not all cohorts recorded the exact age when children received their autism diagnosis, we used the child's age during the study data collection when caregivers first reported the diagnosis as an approximation of age at autism diagnosis.”

Discussion: “Second, our trajectory models (Figure 2) were built using only the SDQ, which measures a wide range of parent-reported neurodevelopmental and mental health traits. Although the SDQ is correlated with an autism diagnosis, it does not fully capture core autistic traits, and other measures of autistic traits were not available in the birth cohorts.”

5. Lack of effect of IQ and sex in the early vs. later diagnosis differences. The additional analysis incl. in response to R#1 of demographic factors, sex, IQ/cognitive ability socio-emotional problems and autism severity and % variance accounted for are important and add to contribution this paper makes in terms of quantifying vs. other factors (as far as they can be or are measured in the available datasets) to age of diagnosis in SPARK.

This is my main response to this revision. In the very detailed response to R#1 the authors and also in the very helpful FAQ section they clarify that “genetics explain only about 11% of the total variation in when someone receives an autism diagnosis”. This is explored in Extended Data Figure 2 and for SPARK in Table 1.1 and then in the Supplementary Table 9 for MCS and LSAC-B. There is significant variation in the % of variance in age of diagnosis across these 3 cohorts (and I note lower in SPARK than in MCS and LSAC-B though SPARK is a much larger sample of autistic individuals) but the message (again from the FAQ summary – which is a fair lay description of the pattern of findings) the authors contrast the 11% genetic contribution to the ~10% - 60% from these other factors. I would prefer this kind of quantitative comparison to be more ‘up front’ in the main paper and Abstract as it seems to me important in judging the importance (sic) of the novel analysis and finding here. It’s an effect size (as a non-geneticist) that I can get my head around. Across the various cohort/sample data hard to fairly summarize but ~11% is about the same as autism severity (in SPARK) and around the range (or less) across sex and sociodemographic factors and SDQ scores in MCS and LSAC-B (Supplementary Table 9) – which is an accessible ‘effect size’ estimate to my mind i.e. how important is the new ‘polygenic architecture’ factor presented in explaining variation in age of diagnosis.

We agree and have now made this change both in the abstract and the discussion. We note that the 60% is obtained after combining the variance explained by the SDQ-total and subscale scores in the MCS cohort after imputation. However, no individual subscale or total score explained greater than 15% of the variance. This is consistent with the literature review presented in Extended Data Figure 1, where, with the exception of three outliers, all other variables individually explained less than 15% of the variance. Subsequently, we have stated that individual sociodemographic and clinical factors typically explain less than 15% of the variance. We have made the following changes to the manuscript:

Abstract: “Common genetic variants account for approximately 11% of the variance in age at autism diagnosis, comparable to the contribution of individual sociodemographic and clinical factors, which typically explain less than 15% of this variance.”

Introduction: “Several social, demographic, and clinical factors have been linked to age at autism diagnosis. However, past studies show that individual clinical and sociodemographic factors explain only a modest proportion (typically less than 15%) of the variance in age at autism diagnosis (Extended Data Figure 2, Supplementary Table 1). This suggests additional factors contribute to age at autism diagnosis. One of these additional factors could be genetic differences among autistic individuals. Despite the relatively high heritability of autism¹⁸, the role of genetics in age at autism diagnosis has not been studied to date.”

Discussion: “These findings must be interpreted considering several limitations. First, the SNP-based heritability for age at autism diagnosis is only about 11% (Figure 3), and other observed developmental and demographic factors typically explain less than 15% of the variance (Extended Data Figure 2).”

Minor point but in Extended Data Figure 2 I would order (top-to-bottom) the variable sets: sociodemographic, sex, autism severity, clinical factors, socioemotional (probably more appropriately termed socioemotional-behavioral given SDQ subscales). A further point to ensure is emphasized is that these datasets do not capture other family/service access/contextual factors that also impact on age of diagnosis in these different cohorts (particularly the population ones perhaps).

We have re-ordered the variables in Extended Data Figure 2 as requested. We have indicated that family/service access/contextual factors are not adequately captured across these datasets. We have renamed SDQ as socioemotional and behavioural throughout the text and figures.

6. CIs for sex ratios in the cohorts since male:female ratio varies from 3.92 to 2.11 which is substantial (if not significant). Also, the $p > .05$ is less informative than the actual non-significant p-value – please replace.

CIs added. I still find $p > .05$ less informative than knowing the point estimation for a single p-value or minimum p-value for a set of values.

Apologies for the oversight. We have now provided exact p values for the male:female ratios in Supplementary Table 4b. We have masked specific sample sizes where any of the groups had less than 10 participants, given it would be easy to calculate sample sizes using odds ratio. And we have added table note:

“Odds ratios and p-values are presented with masked participant numbers where at least one group had fewer than 10 participants, to protect participant confidentiality.”

7. IQ/ ‘cognitive aptitude’ scores in the different cohorts. Helpful to clarify role of IQ and cognitive ability.

Thank you.

8. Females more likely to be diagnosed with autism later

As above, sex and IQ now more fully considered and (again as above) the addition of Table 1.1 – and Extended Data Figure 2. Is Table 1.1 in the response letter/rebuttal in the SM I cannot find it?

Thank you. We have now provided Table 1.1 in the Supplementary Tables (Supplementary Table 1B).

9. SPARK diagnosed before versus after 5 years.SPARK before vs. after 6 years age of diagnosis helpful. In this response the authors make the point about ‘strength of gradients’ depending on the age of diagnosis – with perhaps an inflection around ~11 years – and I am not sure this comes across from in the main paper and I think it is helpful as any age threshold is arbitrary (or at least chronological age ones are).

Thank you. We have made changes to reflect this in the Results and Discussion sections. We note that we don’t quite find an inflection around ~11 years.

Results: “In Aim 3, we tested this by estimating genetic correlations among the 13 autism GWASs (**Box 1**). We observed genetic correlations ranging from 0.02 (SE = 0.13) to 1 (SE = 0.01) (**Figure 5A, Supplementary Table 13**). We observed a gradient in the genetic correlations related to the similarity in median age at diagnosis between cohorts. Cohorts with the most similar median ages at diagnosis (differing by ≤ 2 years) showed the highest genetic correlations ($r_g = 0.88-1.07$), while correlations progressively decreased as age differences increased, with some dropping as low as $r_g = 0.02$ when comparing cohorts with an median age at diagnosis in early childhood (~ age 3) with cohorts with a median age at diagnosis in adolescence (~ age 16).”

Discussion: “Fourth, we use 'earlier' and 'later' diagnosed autism as relative terms, reflecting that developmental and polygenic differences represent a gradient (Figure 5A, Supplementary Figure 10) rather than discrete categories, particularly given the lack of consensus on age thresholds for early versus late diagnosis.”

10. Formal ID (intellectual disability) is or is not represented in the population and genetic cohorts. Some additional analysis presented in MCS.

Thank you.

11. Contextual and service factors are very influential in the population cohorts where diagnosis relies on parent report of received diagnosis. As above contextual factors now better addressed.

Thank you.

12. Genetic correlation between the factors as 'moderate' (0.38) not 'low' but that might not make sense to geneticists. I am happy with the change (but I am not a geneticist).

Thank you.

Referee #3

This is a revision of “The polygenic architecture of autism varies by age at diagnosis” (previously titled: “An axis of genetic heterogeneity in autism is indexed by age at diagnosis and is associated with varying developmental and mental health profiles). I was one of the original reviewers, and am focusing this review both on the overall manuscript and on responsiveness to Reviews 3 and 4 from the original submission.

I am of two minds regarding this revised submission. On the one hand, while the manuscript has improved since the original submission, it is very dense- reporting a large number of analyses of many different datasets that are difficult to follow (both in terms of the text and in terms of the figures) and thematically connect without a great deal of re-reading and cross-checking information across different parts of the manuscript. Given my expertise in neurodevelopment, complex trait genetics, and statistical methods, I am the exact sort of reader who should find this manuscript tractable. So if I am having difficulty, I fear that the more general reader may be entirely lost. On the other hand, I do think there are some genuine findings and meaningful inferences reported that are important for the field. I think that it is interesting and important that the genetic etiology of early-onset ASD is highly dissociable from that of later diagnosed ASD, with distinct patterns of genetic correlates, and distinct trajectories of symptomology. I also believe that a GWAS of age of onset within ASD cases only case-control GWASes of ASD stratified by age of onset are of potentially high value to the field. I found myself wondering whether the dense nature of manuscript and difficulty to follow was attributable to the strict word/page limits required for a Nature submission, especially given the breadth of analyses reported. However, I also found the responses to reviewers to be difficult to follow and often not directly linked to the issues raised, many times enumerating a slew of tangential points for a given concern, rather than directly successfully addressing the concern. This suggests that perhaps the authors have a more general difficulty clearly and directly articulating their work in an easily digestible way and makes me concerned that the manuscript would require several round of revisions and considerable feedback from reviewers at each juncture before it is ready for publication.

Thank you for this feedback. We have now gone through the manuscript and made extensive changes to the manuscript, figures, and supplementary material to improve readability. Key changes are:

1. We have moved several technical sensitivity analyses to the Supplementary Note. We have further reduced the Discussion section in line with the Editorial Guidelines. This has resulted in reducing the word count from 4,377 to 4,016 words, and making the manuscript more streamlined.
2. We have provided a box with brief descriptions of all 13 autism case-control GWAS used in this study. In addition, in Figure 4, we have provided the median and median absolute deviations of the age at autism diagnosis for each of these GWAS.
3. We have simplified the figures, by providing more labels and splitting complex figures into smaller figures. Now, each figure represents one concept in the manuscript.
4. We have reduced the number of acronyms used.

We respond to specific points below, but hope you find the improved manuscript easier to read.

I am not able to fully detail the many points of difficulty that I had in reading the manuscript, but I will provide a few examples. **One point is that there are many different datasets used, each of which is represented by a different acronym. Some of the datasets are small and used for growth mixture modeling, some are larger and used for case-control GWAS stratified by age of onset, some are larger and used for age of onset GWAS in cases only, others are larger and used for polygenic prediction.** Each dataset has unique features, some of the most salient being the substantial differences in the distribution of ages at which variables are measured and in the distribution of age of ASD onset. These are all important features that must be taken into account, and the relevant information is reported. **However, all too often datasets are simply referred to by their acronym without explicit reference to their sample size, analysis performed, or age (of measurement or onset) distribution, even though these features are key to the interpretation provided.**

We have addressed this as follows:

1. For the autism case-control GWAS, we have now provided Box 1 which introduces all 13 GWAS and provides median age at diagnosis and sample size. Furthermore, in Figures 4 and 5, we provide the median (and median absolute deviation) age at autism diagnosis and sample size of the cases.
2. For the longitudinal trajectory analyses, we have excluded Growing up in Ireland - GUI in the main text, and instead refer to it in the Supplementary Text when we present the findings from the Latent Growth Curve Model analyses. We have also provided the sample sizes and the names of the cohorts in Figure 2. Sample sizes have been provided in text as well.
3. We have removed additional acronyms throughout the manuscript.

For instance, take Figure 3, its associated caption and in-text description. Only after very carefully reading through all of this information am I able to keep in the fore of my mind that the SPARK discovery, SPARK replication, and iPSYCH datasets are used to conduct case-only GWAS of age of onset (which is presented in panel A), and that the covariate control analysis in panel B is conducted specifically with respect to this case-only age-of-onset analysis in the SPARK cohorts only, and that panel C reports genetic correlations between these different case only age-of-onset GWAS with different case-control analyses of ASD (stratified by data source, and in some cases age of onset of cases and sex). Ideally, this information should be so intuitively displayed in the figure that the reader can draw the key inferences without ever reading the caption or main text. Rather, the figures are entirely opaque without very careful readings of such sources.

We have now split Figure 3 into two figures. New Figure 3 presents the results of the heritability analyses. New Figure 4 presents the results of the genetic correlation analyses, and also provides the median and median absolute deviations for the age at autism diagnosis in the case-control autism GWAS used. We also plot the sample sizes (of cases) from these GWAS. More generally, we have simplified the Figures throughout the manuscript, splitting complex figures into smaller, simpler ones. We believe this will make it

easier for readers to obtain the necessary information from the figures.

Further, in the main text, the authors indicate that it is notable that in panel C of Figure 3, “we observed variable genetic correlations between age at autism diagnosis and different GWAS of autism,” however it is unclear why this heterogeneity occurs. **Presumably it is because the different case-control GWAS of AD differ in the typical age of onset of the ascertained cases- but this is not make explicit (except for the iPSYCH after 10 and iPSYCH before 11, which I also not are unintuitive names- “10 and under” and “11 and older” would make more sense).**

Box 1 now provides the median age at diagnosis for all GWAS. In addition, in old Figure 3 (new Figure 4) we also provide the median age at diagnosis for all GWAS.

We choose not to use the terms under and older, as discussions within the team suggest that this can be misinterpreted to mean chronological age of the participants as opposed to age at diagnosis.

This issue carries forward to the analyses presented in Figure 4. **Why is it that PGC 2017 loads on the early onset factor and FinnGen loads on the late onset factor?** This is interesting but the authors do not make clear whether this would be expected based on how ASD was ascertained in those two datasets. They leave it to the reader to put the pieces together.

Apologies that this was not clear in the earlier version. We have addressed this by providing median age at autism diagnosis for all GWAS in Box 1, New Figures 3 and 4. FinnGen has a median age at diagnosis of 22.66. For PGC-2017, although no age at diagnosis has been provided, the majority of the participants were diagnosed as autistic under DSM IV criteria, so must have had clear signs present before three years of age. The majority of the participants were recruited as trios through medical and research centres across the United States, and is most similar in ascertainment to SPARK. For illustrative purposes, we have used the median age at diagnosis from SPARK trios (Matoba et al., 2020), i.e., 3.5 years.

The same reliance on the reader to “read between the lines” is apparent in the Discussion. For instance, the authors write:

“This two-polygenic trait genetic model also explains the often contradictory patterns of genetic correlations between autism and various neurodevelopmental, and psychiatric phenotypes across different autism GWAS. The variation in genetic correlation between ADHD and autism stratified by age of diagnosis is particularly noteworthy. Older GWAS of autism (including the PGC-2017) were not significantly genetically correlated with ADHD45–47 whereas more recent GWAS for autism have moderate genetic correlations with ADHD48. Genetic correlation analyses (Figure 4) indicate that the genetic correlation with ADHD increases with a later diagnosis of autism.”

By older GWAS of autism, I gather that they are referring to GWAS published several years ago. At first I thought they were referring to GWAS of older age of onset- but that is clearly not the case given that PGC-2017 loads on the early onset factor in Figure 4. Even putting aside this point of confusion, the authors indicate that they have explained a contradictory pattern (differing patterns of genetic correlation between ADHD and ASD by GWAS) because

the ADHD-ASD r_G varies by age of onset- but they do not complete the thought for the reader because they never make explicit whether the “older” (in time, not age) GWAS indeed disproportionately included earlier age-of-onset cases. Again, the reader is left to put the pieces together.

Apologies that this was unclear. We have now revised the Discussion, making it substantially shorter and removing any ambiguities. With reference to the point made here, the new revised text reads: “This two-polygenic trait genetic model provides one framework to understand genetic heterogeneity in autism, and the varying patterns of genetic correlations between different GWAS of autism and other phenotypes. For example, previous GWAS of autism (including PGC-2017) found limited genetic correlation with ADHD, contrary to findings from more recent autism GWAS (e.g., Grove et al., 2019). We show that this is explained by the different average age at diagnosis across these GWAS (Box 1, Figure 4), as the genetic correlation between autism and ADHD increases with later age at autism diagnosis (Figure 6). These findings were confirmed using within-family analyses that demonstrated over-transmission of ADHD polygenic scores primarily to individuals with a later autism diagnosis (Supplementary Table 24).”

I would anticipate that the authors may be frustrated by the above- perhaps they feel that the relevant information is indeed in the manuscript and can even point to where it can be found. I do not dispute this. Rather, my comment is that it is the responsibility of the authors to make the information accessible rather than relying on the reader to do the work to search through the manuscript, supplement, and figure captions to find it. Indeed, R4 had also previously written that “It is hard to keep track of the meandering storyline” and “I consider the findings presented in figure 4b as central and very insightful, but the way to this point in the paper is arduous.”

We believe the significantly revised manuscript addresses this comment.

The authors report that “The age at diagnosis in the two cohorts were moderately genetically correlated with each other ($r_G = 0.51$, s.e. = 0.19, $P = 7.56 \times 10^{-3}$, between iPSYCH and meta-analysed SPARK) and had variable genetic correlation with other mental health phenotypes (Supplementary Table 11). This is likely due to the different distribution of age at autism diagnoses in the two cohorts (Extended Data Figure 5) because of differing ascertainment.” The statement that this is likely due to the age distributions is speculative and unsatisfying. For instance, what if the authors were to rerun the age of onset GWAS, attempting to better match the age of onset distributions, e.g. by removing all cases with onset after age 10 where Extended Data Figure 5 indicates they diverge most greatly? Reviewers had raised concern about this genetic correlation being rather low, and the response was that the case-control GWAS of early onset were strongly correlated (~ 0.8) across datasets, as were the case-control GWAS of late onset. However, this is only partially helpful, as the concern about the age of onset GWAS concerns whether the onset phenotype itself in the cases only is a valid and useful analysis.

Thank you for the suggestion. We have calculated genetic correlations between age of diagnoses GWAS in iPSYCH stratified to those diagnosed before and after 9, and the age at diagnosis in SPARK (Supplementary Note 6, Figure 1, provided below). We have additionally

calculated the age of diagnosis GWAS in SPARK using overlapping subsets across the age distribution. We created five subsets of approximately 9,700 autistic individuals each, with increasing mean age at diagnosis and calculated the genetic correlation with the iPSYCH GWAS .

As seen in Figure Supplementary Note 6, Figure 1 the genetic correlation between iPSYCH and SPARK increases as the samples become more similar in their median age at autism diagnosis. The iPSYCH GWAS has larger mean and median age at diagnosis compared to any of the subsets. We have provided these results in **Supplementary Note 6**.

Supplementary Note 6, Figure 1: Genetic correlations among different age at autism diagnosis GWAS. A. Genetic correlations between age at autism diagnosis in iPSYCH (median age = 10, median absolute deviation = 4) and SPARK discovery subsets arranged by median age at diagnosis. B. Genetic correlation between age at autism diagnosis in SPARK (median age = 4, median absolute deviation = 2.71) and iPSYCH subsets. For both A and B, genetic correlations with 95% confidence intervals (CI) are shown as black points with horizontal error bars. For visualisation purposes, error bars are constrained to the range of -1 to 1, while text labels display the actual, unconstrained genetic correlation and 95% CI values. The

vertical dashed line indicates zero correlation. Phenotypes are arranged by median age at diagnosis, with the median and median absolute deviation values in parentheses and sample size (N) below each label.

A technical concern: I was surprised to find the statement “For genomicSEM39 analyses, we first conducted genetic correlation analyses among different autism GWAS using LDSC. This included a multi-ancestry case pseudocontrol GWAS in SPARK93 (6,222 case-pseudocontrol pairs)...” Both LDSC and Genomic SEM can handle multiple ancestries one at a time, but cannot be validly applied to multi-ancestry GWAS. If this was indeed what was done, then the results reported cannot be trusted. Elsewhere the authors indicate that European ancestry LD scores were used, but it appears from the above statement that although European ancestry LD scores were used (this is the independent variable in LD score regression), multi-ancestry GWAS summary data from SPARK were used (this is the dependent variable in LD score regression). This is not appropriate. I do not that in the Discussion, there is the statement “Finally, our genetic analyses focused on genetically inferred European ancestries due to limited GWAS data from other populations.”

We have now removed the analyses of summary statistics from the multi-ancestry GWAS and rerun all genetic correlation and genomicSEM using summary statistics from the same study restricted to individuals with genetically inferred European ancestries (N = 4,535 case/pseudocontrol pairs). We identify consistent results between the multi-ancestry GWAS and the European only subset. All other GWAS used in the study have been restricted to individuals of genetically inferred European ancestry. We have updated the Methods, Results, Figures, and Supplementary Tables to reflect this change.

I will conclude by mentioning a few items that I found valuable and responsive. I found that Extended Data Figure 7 was a very nice way to demonstrate the differential patterns of rG by age of onset are not driven by differences in cohorts. I thought Extended Data Figure 1 and Note 6, Figure 1 were nice ways of illustrating the distinct theoretical models of onset-stratified autism etiology. I thought that the analysis represented by Note 6, Figure 2 was a clever way to pragmatically address the issue of potential differences in the accuracy of differential diagnosis by age of onset.

Thank you!

Changes Requested

Referees' comments:

Referee #1 (Remarks to the Author):

I appreciate the authors' careful and thorough response to the previous set of reviews, both clarifying critical findings as well as conceptual issues. No further comments or suggestions, other than to commend the authors on this important work.

Lonnie Zwaigenbaum MD
University of Alberta

Thank you very much for the very helpful reviews of this manuscript.

Referee #2 (Remarks to the Author):

The authors have been responsive again to Reviews and the manuscript is improved again. It is a fairer and clearer communication of what was done and what was found and their interpretation. I am not going to provide detailed comments on this version. I have reviewed the article and some of the Supplement etc. and will make some minor comments for final consideration.

1. The Abstract correctly describes the 4 independent birth cohort studies but I think it helpful to indicate to the reader here that the genetic analysis is on different samples.

Thank you. We have now said: **In independent cohorts of autistic individuals, common genetic variants account for approximately 11% of the variance in age at autism diagnosis, comparable to the contribution of individual sociodemographic and clinical factors, which typically explain less than 15% of this variance.**

2. Abstract - Is it possible to indicate that in addition to the 2nd (later) polygenic factor having moderate-to-high positive genetic correlations with ADHD and MH conditions that the genetic correlation for the 1st (earlier) polygenic factor is low. Appreciate this might be in part repetition and indirect implicit as stated but that is what I see when I look at Figure 6. Perhaps they can combine the moderate-to-high and the contrast in one sentence?

Thank you. We have made this change in the abstract. The abstract now reads: **One of these factors is associated with an earlier autism diagnosis, and lower social and communication abilities in early childhood but is only modestly genetically correlated with ADHD and mental health conditions. Conversely, the second factor is associated with a later autism diagnosis, increased socioemotional and behavioural difficulties in adolescence, and has moderate to high positive genetic correlations with Attention-Deficit/Hyperactivity Disorder and mental health conditions.**

3. The most important change for me is the contextualisation of the 11% variance with other variance (that they have available) ~15% and this is in the Abstract. I do keep

going back to Extended Data Figure 2 and Suppl Table 1 to look at the data and I would like to see these more prominent though appreciate this might not be possible. Extended Data Figure 2 is quite hard to read but I can see no easy solution...

Thank you. We agree that this is an important piece of information and we have highlighted the ~15% of the variance throughout the text (abstract, results, and discussion). As the reviewer says, it's difficult to make this more prominent in the current manuscript. We have now revised Extended Data Figure 2 (now called Extended Data Figure 3) to change the orientation to make it easier to read.

Referee #3 (Remarks to the Author):

This is my third review of the submission that is now titled "The polygenic and developmental profiles of autism differ by age at diagnosis." The manuscript has evolved considerably since the first submission, and the authors have put a great deal of effort into updating analyses and revising the text and figures based on previous feedback. The updated figures are a major improvement in terms of clarity of communication, and Box 1 is a helpful new addition to aid the reader in accounting for all of the many datasets and their features. I do believe that this paper is innovative and important in establishing that autism diagnoses made very early in child development do not simply differ in their levels of severity, but are considerably etiologically distinct (and therefore probably also taxonomically distinct) from those made later in child development and adolescence/early adulthood. Not only does this result enhance scientific understanding, but it helps to resolve previous discrepancies in the literature, and it informs decisions on how data should be aggregated across individuals and cohorts in future research.

Thank you for the very helpful feedback!

I had previously indicated that the manuscript was a long way from being clear and articulate in a way that did not impose on the reader to assemble the different pieces of information and results together in their own head in order to make sense of the conclusions drawn. I wrote in my last review that I was "concerned that the manuscript would require several rounds of revisions and considerable feedback from reviewers at each juncture before it is ready for publication." In these respects, I think that the manuscript is much improved. I continue to think that the manuscript (text and figures) would benefit from some additional revisions for clarity of communication, but I don't think that additional reviews are appropriate after this. For instance, in Figure 5, panel B it is still not clear without cross-referencing with other panels that PGC (2017) has a very low median age (3.5) at diagnosis and FinnGen has a very high (22.66) age at diagnosis. Yes, the information is there in panel A, but it would be so much more intuitive if the median age for each of the 6 GWAS samples in panel B were also printed right under the sample name.

We have now printed the median age at diagnosis in Panel B.

Similarly, Note 6, Figure 1 is fantastic and the exact sort of analysis that I had suggested. It really drives home the inference that the genetic correlation between different age at diagnosis GWAS depends on the developmental period under

consideration, and directly supports the inference that the moderate genetic correlation between the SPARK and iPSYCH age at autism diagnosis GWASs can be attributed to the differing median age at diagnosis. This is really stunning, and could even be included in the main text. However, The Y axis of the plot in panel A refers to subsets but never makes clear that it is subsets of SPARK (even though the parallel plot in panel B does indicate that the subsets on the Y axis are for iPsych). It is also not clear what the numbers in parentheses on the X and Y axis tick labels in both panels are without digging into the figure note (they are the median age and median absolute deviation in age from the median). I just don't see why the reader should have to do work to make sense of this figure when some clear labelling would make the figure extremely intuitive to understand without even having to read the caption. If the manuscript is accepted, I would hope that copy editors and journal staff would help with these further issues, so as to optimize clarity and readability. Like I indicated in my previous review, it's not that the information is missing. It's about presenting it clearly and accessibly.

We have now clarified the Y axis, indicated the median and MAD and made further changes requested by the Journal team. We prefer to leave Note 6, Figure 1 in the Supplementary Note.

I have been asked to comment on whether the language around describing "autistic individuals" (and similar) is appropriate - or would the use of "individuals with autism" (and similar) be more appropriate? The most appropriate language of course shifts with time, and varies according to specific diagnosis and the preferences of the communities of affected individuals. My understanding is that while there is not a consensus among all individuals within the community, there has been a recent movement toward identity-first language in the case of autism. In other words, many individuals in the autism community prefer the term "autistic person" over "person with autism." However, others still prefer person-first language (i.e. "person with autism"). I do not think that there is a clearly correct or more appropriate term, but I do believe that identify-first is becoming increasingly popular, and I am in support the language as it is currently used in the manuscript, if that is the preference of the authors (who themselves have considerable experience working in this research area).

We agree and prefer to use identity first language.